# Multi-Group Proportional Representation in Retrieval

**Alex Oesterling**[*]
Harvard University

**Claudio Mayrink Verdun**[*]
Harvard University

**Carol Xuan Long**
Harvard University

**Alexander Glynn**
Harvard University

**Lucas Monteiro Paes**
Harvard University

**Sajani Vithana**
Harvard University

**Martina Cardone**
University of Minnesota

**Flavio P. Calmon**
Harvard University

## Abstract

Image search and retrieval tasks can perpetuate harmful stereotypes, erase cultural identities, and amplify social disparities. Current approaches to mitigate these representational harms balance the number of retrieved items across population groups defined by a small number of (often binary) attributes. However, most existing methods overlook intersectional groups determined by combinations of group attributes, such as gender, race, and ethnicity. We introduce Multi-Group Proportional Representation (MPR), a novel metric that measures representation across intersectional groups. We develop practical methods for estimating MPR, provide theoretical guarantees, and propose optimization algorithms to ensure MPR in retrieval. We demonstrate that existing methods optimizing for equal and proportional representation metrics may fail to promote MPR. Crucially, our work shows that optimizing MPR yields more proportional representation across multiple intersectional groups specified by a rich function class, often with minimal compromise in retrieval accuracy. Code is provided at https://github.com/alex-oesterling/multigroup-proportional-representation.

## 1 Introduction

A recognized objective in fair machine learning (ML) is discovering, reporting, and mitigating *representational harms* [1–3]. Representational harms arise when systems reinforce and perpetuate the marginalization of population groups based on characteristics such as race, socioeconomic status, cultural background, and gender [1, 4–6]. These harms often manifest through the biased portrayal or misrepresentation of these groups, stereotype reinforcement, and erasure of cultural identities [7, 8]. Representational harms have been widely documented in ML systems. For instance, many freely available datasets used in ML are not geo-diverse [9], and generative models can output images that under-represent demographic minorities across gender and racial identities [10–12].

We focus on representation in retrieval tasks. Representational harms in retrieval and ranking tasks arise when retrieved items do not accurately reflect the true diversity and proportions present in reality, perpetuating harmful stereotypes and biases [13]. For instance, Kay et al. [2] demonstrated that, in 2015, only $10\%$ of the top 100 results for CEO in Google Image Search were women, despite $28\%$ of CEOs in the US being women, and recent studies [14, 15] have demonstrated that this bias is still present in modern image search engines. Later, Otterbacher et al. [16] showed that the search engine Bing usually produced twice as many men as compared to women when queried with the word "person". Finally, [17] systematically studied gender biases in image representations by auditing the four most important image search engines: Google, Bing, Baidu, and Yandex. In particular, among other findings, this study showed that when a qualifier such as "intelligent" is added to the query "person", the engines still exhibit a significant gender discrepancy; see also [1, Chapter 7]. When

---

[*] Equal contribution, correspondence to `aoesterling@g.harvard.edu`

38th Conference on Neural Information Processing Systems (NeurIPS 2024).

retrieval is based on a search over vector embeddings of images and text, biases in embedding models can propagate to retrieved results, leading to disparate representation of people by demographic groups such as gender or race [18–20], as well as propagating spurious correlations from the dataset more broadly [21] or distorting results due to stereotypes present in the representation [22]. Though correlations between unrelated semantic concepts do not necessarily constitute representational harms, a wealth of research charts undesirable bias in embedding models, including publicly available vision-language embedding models such as CLIP [23] used for retrieval. Bias in embeddings can lead to bias in retrieval [24–26]. In fact, recent work by Srinivasan et al. [27] argues that many image retrieval systems lack "social diversity" as perceived by humans.

Several interventions aim to control gross statistical deviations in group representations in retrieval. A common approach is ensuring equal representation of pre-defined population groups in retrieved items [28, 29]. An alternative goal is proportional representation (PR) [20, 30], where the target representation of different groups is proportional to some reference population or statistic. Various interventions have been proposed to achieve equal or proportional representation, including optimizing for diversity in addition to the similarity in retrieval [27, 31], directly selecting the objects retrieved based on balancing known or predicted sensitive attributes [29] and, for vector databases, modifying embeddings directly to remove information about group-defining attributes [18, 20].

Achieving proportional representation across multiple intersectional groups is challenging, as existing fairness interventions typically consider only a small number of pre-defined groups. Ensuring representation across individual groups (e.g., given by gender **or** race) does not guarantee representation across *intersectional* groups (e.g., given by gender **and** race), as demonstrated in Table 1 (and Table 5 in the Appendix). The study of intersectionality and its consequences is rooted in work in law and the social sciences [32–34] and has been a well-discussed problem for several decades. In machine learning systems, lack of intersectional representation can contribute to the invisibility of historically marginalized groups determined by multiple axes of identity [35], or their mistreatment through "fairness gerrymandering," where interventions on fairness for specific groups may harm fairness on intersectional subgroups [36]. However, as the number of group-denoting attributes increases, the number of intersectional groups grows exponentially, quickly surpassing the number of retrieved items and limiting the achievable proportional representation. In fact, achieving optimal proportional representation across multiple attributes has been shown to be NP-hard [37].

We propose a metric called *Multi-group Proportional Representation* (MPR) to quantify the representation of intersectional groups in retrieval tasks. MPR measures the worst-case deviation between the average values of a collection of *representation statistics* computed over retrieved items relative to a reference population whose representation we aim to match. The set of representation statistics is given by a function class $\mathcal{C}$, where each function $c \in \mathcal{C}$ maps a retrieved item $x$ to a real number. For instance, $c(x) \in \{-1, 1\}$ may denote binary group membership for an input $x$. In theory, $\mathcal{C}$ can be given by bounded complexity function classes, such as linear functions or shallow decision trees. In practice, $\mathcal{C}$ are functions defined over item attributes, such as vector embeddings or group-denoting labels, enabling MPR to measure proportional representation across complex, intersectional subgroups. Compared to naively counting the number of retrieved items across a (potentially exponential) number of pre-defined groups, MPR offers a more flexible, scalable, and theoretically grounded metric for multi-group representation in retrieval.

In addition to **introducing and carefully motivating the MPR metric**, cf. Section 2, we make three contributions. **First**, we show how to compute MPR for several function classes in Section 3. Computing MPR relies on a curated dataset that represents the desired population whose proportional representation we aim to reflect in retrieved items. We derive sample complexity bounds for the size of this curated dataset based on the complexity of the function class $\mathcal{C}$. We also prove that MPR can be computed in closed form for certain function classes or with a regression oracle. **Second**, we propose MOPR, an algorithm that retrieves items from a vector database to maximize their average similarity to a query embedding while satisfying an MPR constraint relative to the curated dataset (Section 4). MOPR achieves this by iteratively calling an oracle that computes MPR violations, yielding both relevant and representative retrieved items. **Third**, we evaluate MOPR on retrieval tasks using datasets such as CelebA [38], the Occupations dataset [31], Fairface [39], and UTKFace [40]. In section 5, we show that MOPR Pareto-dominates competing approaches in balancing retrieval similarity and MPR.

**Related Work.** Broadly speaking, existing work on fair and diverse retrieval with pretrained embeddings either: 1) modifies the embedding space to prioritize downstream fairness, or 2) provides retrieval algorithms optimizing for diversity or fairness.

*Mitigating Bias in Embedding Space.* Various approaches have been proposed to produce vector embeddings of text and images that target diversity or mitigate bias. These include modifying loss functions to encourage group fairness [41], adversarial training [42, 43], disentangling representations [44], and in-processing fair sampling methods for gender imbalance [18]. Many popular embedding models are not trained with such approaches, so post-processing methods for fairness have also been developed for multimodal models such as CLIP [23], including CLIP-clip [18], CLIP-debias [20], and FairCLIP [19]. Recent work by Srinivasan et al. [27] builds upon the pretrained image-text model CoCa and leverages text-guided projections to extract an embedding, called PATHS, that captures complex notions of diversity in people, and then conduct diverse retrieval with a greedy algorithm. However, code for implementing and reproducing PATHS embeddings was not available at the time of publication. In our experiments, we report the MPR achieved by CLIP-clip and CLIP-debias in retrieval tasks and approximate the greedy retrieval algorithm proposed in [27] with vanilla CLIP embeddings. Unlike the aforementioned methods, MOPR is not based on modifying embeddings.

*Retrieval Algorithms with Diversity or Fairness.* Instead of modifying embeddings, several methods aim to promote diversity, given possibly biased embeddings. Given an additional diversity constraint, a popular approach is Maximal Marginal Relevance (MMR) [45], where items are greedily retrieved in order to maximize a linear combination of similarity and diversity metrics. Celis et al. [30] and Celis and Keswani [31] focus on diversity and fairness in image retrieval, using a reweighing method that combines similarity and MMR to select diverse yet relevant images. Alternatively, the Post-Hoc Bias Mitigation [29] method aims to achieve equal representation over a pre-defined attribute (such as gender) through calls to an oracle classifier and selecting a balanced group of images. **We extend this prior work in two key ways:** (i) we aim to achieve proportional representation across intersectional groups, going beyond the representation of two groups defined by binary attributes; (ii) MOPR explicitly balances MPR and retrieval similarity via an optimization that controls for both, offering an efficient alternative to greedy MMR-based methods.

*Diversity Metrics.* A traditional measure of diversity for a set is the pairwise similarity of the retrieved items [46]. However, the general visual (dis)similarity metric is insufficient in optimizing for people-related diversity such as skin tone or gender [30]. When the attribute (e.g., race) along which diversity is optimized is known, or, in other words, we have access to a ground truth attribute label, one can use fairness definition-derived metrics such as proportional representation (see next section and [47]). For more complex notions of diversity that aim to capture intersectional or subtle sociocultural identities (e.g., cultural backgrounds, lifestyles, nationalities, religions), human annotators can be employed to determine if the selected set of images is more diverse than the baseline (retrieval based on similarity to query only) [27]. MPR complements existing metrics by providing a rigorous approach to measuring representation across complex, intersectional groups defined by a rich class of functions. However, as with any fairness metric, MPR has limitations – see Section 6 – and may not always align with human perceptions of representation. In particular, we underscore the importance of involving stakeholders in defining representation goals and auditing retrieval systems.

*Multi-Group Fairness.* Our work is directly informed by the burgeoning literature on multi-group fairness in classification, particularly the work of Hébert-Johnson et al. [48], Kim et al. [49, 50] who proposed rigorous frameworks of multi-group fairness auditing and post-processing to ensure fair predictions across identifiable subgroups. Recent efforts in this field include [51–53], which analyze sample complexity aspects of measuring and reporting multi-group representation, and [54, 55] who develop stronger variants of multi-group fairness notions. Unlike prior work, we focus on multi-group representation in retrieval rather than multi-group fairness in classification and regression.

*Multi-Attribute Proportional Representation.* A recognized challenge in computational social choice research is building committees, groups, or sets of items that ensure proportional representation across several attributes [56–59]. The work most closely related to ours is by Lang and Skowron [37], which introduced the problem of multi-attribute proportional representation, where a set of items must be selected to reflect desired distributions over multiple attributes simultaneously. Their work established fundamental computational complexity results, showing that finding optimal proportional representation is NP-hard, and developed approximation algorithms and integer linear programs for promoting proportional representation across attributes. While our work shares similar

goals, our MPR metric takes a different approach by measuring the worst-case deviation over computationally-identifiable groups defined by a function class $\mathcal{C}$. This formulation connects naturally with statistical learning theory through MMD [60] and sample complexity bounds (Propositions 1 and 2). Though we use the term *multi-group* rather than *multi-attribute* to emphasize connections with the burgeoning literature on multi-group fairness [48, 49], both terms describe related problems of ensuring representation across multiple dimensions. We also highlight that work on multi-attribute representation in computational social choice (including [37]) strives for proportional representation marginally on each attribute, while our primary goal with this work is to achieve intersectional representation. Finally, while [37] similarly seeks to match retrieved candidates to a target distribution, we focus on database retrieval rather than social choice applications. The rich history of proportional representation in social choice theory and political sciences provides, however, an important context for our work. We briefly discuss these connections in Appendix E. There, we elaborate on how our approach relates to and differs in terms of representation formulation from [37].

## 2  A Multi-Group Proportional Representation Metric

**Preliminaries.**   Consider a *retrieval dataset* of items from which we aim to retrieve relevant entries, denoted by $\mathcal{D}_R = \{\boldsymbol{x}_1^r, ..., \boldsymbol{x}_n^r\}$. We assume that $\boldsymbol{x}_i^r \in \mathbb{R}^d \times \mathcal{G}$, where $\mathcal{G}$ is a set of additional labels for each item. For example, items $\boldsymbol{x}_i^r$ can be $d$-dimensional embeddings of images, videos, or textual content, in which case $\mathcal{G} = \emptyset$. Alternatively, $\boldsymbol{x}_i^r = (\mathbf{e}_i, \mathbf{g}_i)$, where $\mathbf{e}_i \in \mathbb{R}^d$ is an embedding and $\mathbf{g}_i$ is a vector of labels indicating group membership (e.g., gender, race, ethnicity).

Given a query embedding $\mathbf{q} \in \mathbb{R}^d$, the goal of retrieval is to search and return the top-$k$ most similar items to $\mathbf{q}$ in $\mathcal{D}_R$. The similarity between an embedding of $\mathbf{q}$ and items in $\mathcal{D}_R$ is measured according to a metric $\kappa : \mathbb{R}^d \times \mathbb{R}^d \to \mathbb{R}$. Throughout the paper, we assume that $\mathcal{D}_R$ is a vector database and $\kappa$ is cosine similarity between embeddings, though this formulation can be generalized. To retrieve items from $\mathcal{D}_R$, users prompt a query $q$ (e.g., "Fortune 500 CEOs") which is then embedded $\mathbb{R}^d$. Ideally, the user receives $k$ retrieved items $\mathcal{R}(\mathbf{q}) = \{\boldsymbol{x}_1^r, ..., \boldsymbol{x}_k^r\} \subset \mathcal{D}_R$ such that $\kappa(\mathbf{q}, \boldsymbol{x}_i^r) \geq \kappa(\mathbf{q}, \boldsymbol{x})$ for all $\boldsymbol{x} \notin \mathcal{R}(\mathbf{q})$ and $i \in [k]$.

The simplest setting for measuring representation is to consider only two population groups denoted by a binary variable – a setting commonly found in the fair retrieval and generation literature [18, 19]. In this case, group membership is determined by a *group-denoting function* $c : \mathbb{R}^d \times \mathcal{G} \to \{-1, 1\}$. For an item $\boldsymbol{x}_i^r$, $c(\boldsymbol{x}_i^r)$ indicates membership to group $-1$ or $1$ (e.g., male/female) based on its embedding $\mathbf{e}_i$ and associated labels $\mathbf{g}_i$. If the retrieval dataset $\mathcal{D}_R$ contains annotations for group membership, $c(\boldsymbol{x}_i^r)$ can simply return the relevant feature from $\mathbf{g}_i$. However, when group labels are not present (i.e., $\mathcal{G} = \emptyset$), $c(\boldsymbol{x}_i^r)$ can be implemented as a classifier that predicts group membership based on the item's embedding $\mathbf{e}_i$.

With a group-denoting function $c(\boldsymbol{x}^r)$ in hand, we can measure the representation of each group in a set of retrieved items $\mathcal{R}(\mathbf{q})$. One popular constraint for representation is *equal representation* [2, 28], which aims to ensure that the number of retrieved items in each group is approximately the same, i.e., $\frac{1}{k} \sum_{\boldsymbol{x}^r \in \mathcal{R}(\mathbf{q})} c(\boldsymbol{x}^r) \approx 0$. An alternative metric is *proportional representation* [20, 61], which quantifies the deviation of group membership from a *reference distribution* $Q$. Methods that promote proportional representation aim to ensure $\frac{1}{k} \sum_{\boldsymbol{x}^r \in \mathcal{R}(\mathbf{q})} c(\boldsymbol{x}^r) \approx \mathbb{E}_Q\left[c(X)\right]$. Here, the measure $Q$ captures the distribution of a reference population whose representation statistics we aim to match. For example, if $Q$ were the distribution of individuals in the US, $\mathbb{E}_Q\left[c(X)\right]$ could measure the proportion of men vs. women in the US and be approximated using Census data.

Proportional representation generalizes equal representation since different groups are rarely uniformly distributed over a given population. Naturally, the choice of distribution $Q$ is application and context-dependent – we revisit the choice of $Q$ below. Importantly, **if $Q$ is biased, then biases will be propagated to the retrieved items.**

**Multi-Group Proportional Representation.**   We aim to ensure that retrieved items represent individuals from diverse and intersectional population groups. Instead of measuring proportional representation in terms of the average of a single group-denoting function $c(\boldsymbol{x}^r)$, we consider a class of $Q$-measurable functions, the *representation statistics class*, $\mathcal{C} \subset \left\{c : \mathbb{R}^d \times \mathcal{G} \to \mathbb{R}\right\}$. This set $\mathcal{C}$ may represent multiple, potentially uncountable overlapping groups. We formally define multi-group proportional representation next.

**Definition 1.** For a reference *representation distribution* $Q$, a set of $Q$-measurable set of *representation statistics* $\mathcal{C}$, and a set of $k$ retrieved items $\mathcal{R}$, we define the *Multi-Group Proportional Representation (MPR)* metric as

$$\text{MPR}(\mathcal{C}, \mathcal{R}, Q) \triangleq \sup_{c \in \mathcal{C}} \left| \frac{1}{k} \sum_{\boldsymbol{x}^r \in \mathcal{R}} c(\boldsymbol{x}^r) - \mathbb{E}_Q[c(X)] \right|. \tag{1}$$

A set of items $\mathcal{R}$ is $(\mathcal{C}, \rho)$-multi-group proportional representative of $Q$ if $\text{MPR}(\mathcal{C}, \mathcal{R}, Q) \leq \rho$.

The MPR metric quantifies the "representativeness" of a rich class of statistics, denoted by functions in $\mathcal{C}$, within a set of retrieved items $\mathcal{R}$. This generalization is crucial for capturing intersectional groups. For example, $\mathcal{C}$ could contain functions that map items to demographic groups based on race, gender, age, or combinations thereof. Alternatively, when labels in $\mathcal{G}$ contain such group-denoting attributes, $\mathcal{C}$ can represent decision trees of a given depth over features (e.g., all combinations of pairs of race, gender, and age). The MPR metric compares the empirical average of each $c \in \mathcal{C}$ over the retrieved set $\mathcal{R}$ to its expectation under the reference distribution $Q$. By defining $Q$ appropriately, we can flexibly specify different statistical representation goals, such as equal representation ($\mathbb{E}_Q[c(X)] = 0$ for binary $c$) or proportional representation w.r.t. a target population. Measuring representation requires defining what constitutes "fair and proportional representation." While equal representation may suffice for binary groups, proportional representation of more complex intersectional groups requires care. The choice of representation reference statistics (given by the distribution Q in the definition of MPR) should be application-dependent, context-aware, and culturally sensitive.

**Remark 1.** MPR is equivalent to the maximum mean discrepancy (MMD) of representation statistics in $\mathcal{C}$ between the empirical distribution over retrieved items $\mathcal{R}$ and the reference distribution $Q$. MMD-based metrics have a long history [60, 62] and are used in hypothesis testing for comparing distributions. MMD is a particular case of the Integral Probability Metric (IPM) [63, 64], allowing us to borrow from the rich literature on IPMs to measure and ensure MPR in practice.

**Remark 2.** MPR can be viewed as the "representation in retrieval" counterpart of multi-group fairness metrics found in classification, such as multiaccuracy [49] and multicalibration [48]. Whereas multiaccuracy and multicalibration measure if classification error residuals are correlated with any group represented within a class $\mathcal{C}$, MPR captures *proportional representation* of groups in $\mathcal{C}$ within a set of retrieved items – a fundamentally different problem. The idea of representing groups in terms of a "computationally identifiable" class of functions $\mathcal{C}$ is directly inspired by the multi-group fairness in classification literature, and we adopt similar notation (i.e., $c$ and $\mathcal{C}$).

**Curated Datasets for Proportional Representation.** A key challenge in computing MPR is selecting the reference representation distribution $Q$ and measuring the expectation of functions in $\mathcal{C}$ against this distribution. In the simple case where $\mathcal{C}$ consists of a small number of functions indicating membership in individual groups, we could potentially set the proportional representation targets $\mathbb{E}_Q[c(X)]$ for each group $c \in \mathcal{C}$ manually, e.g., by defining a target fraction of men/women or individuals from different ethnic backgrounds. This quickly becomes infeasible when there are many intersectional groups – and impossible if $\mathcal{C}$ is uncountable. Moreover, in most practical settings, $Q$ will very likely *not* have a simple closed-form analytical expression.

In practice, i.i.d. samples drawn for $Q$ may be available. For instance, the retrieval dataset $\mathcal{D}_R$ itself may be drawn from the target population for which we aim to preserve proportional representation in retrieved items. Alternatively, we may have access to a dataset that was carefully curated to be representative of a diverse population. Examples include the FairFace dataset [39], which was designed to be balanced across race, gender, and age groups, and the AVFS dataset [65]. More generally, we refer to datasets with samples drawn from the target population $Q$ as a *curated dataset*.

**Definition 2** (Curated Dataset). Let $Q$ be a probability distribution over $\mathbb{R}^d \times \mathcal{G}$ tailored to account for diversity and representation of stakeholders who query the retrieval dataset. The *curated dataset* with $m$ samples drawn the distribution $Q$ is denoted as $\mathcal{D}_C \triangleq \{\boldsymbol{x}_1^c, ..., \boldsymbol{x}_m^c\}$. We denote the MPR of a set of retrieved items $\mathcal{R}$ relative to the *empirical distribution* over $\mathcal{D}_C$ as $\text{MPR}(\mathcal{C}, \mathcal{R}, \mathcal{D}_C)$.

Constructing a proper $\mathcal{D}_C$ is critical to properly measuring bias and preventing downstream harms, and the nature of $\mathcal{D}_C$ is context-dependent and may vary based on the specific application and desired representation goals. We reiterate that if the curated dataset is biased, then these biases will be propagated to downstream usages of MPR. For these reasons, we recommend that curated datasets

used for MPR measurements be developed and verified through participatory design approaches [66, 67] in collaboration with diverse stakeholders. By involving stakeholders in defining representation goals, we can help ensure that the proportional representation in information retrieval systems measured by MPR aligns with the values and needs of the user base these systems serve.

Finally, we note that we can also condition the curated dataset on a given query. Specifically, for a query $\mathbf{q}$, we can retrieve relevant samples from both the curated dataset $\mathcal{D}_C$ and the retrieval dataset $\mathcal{D}_R$. The samples retrieved from $\mathcal{D}_C$ can then serve as a "conditional" curated dataset, denoted as $\mathcal{D}_{C|\mathbf{q}}$, which captures the desired representation target specific to the query $\mathbf{q}$. This approach allows for a more granular and context-aware proportional representation.

In the next two sections, we introduce theoretical guarantees and algorithms that are agnostic to the specific choice of $\mathcal{D}_C$. In other words, our proposed methods for measuring and promoting MPR in retrieval are general and can be applied regardless of how the curated dataset is constructed. In our numerical experiments, presented in Section 5, we use FairFace [39] as the curated dataset.

## 3  Computing Multi-Group Proportional Representation

Computing MPR in Defnition 1 requires approximating two quantities: the representation distribution $Q$ and the supremum $\sup_{c \in \mathcal{C}}$. We first establish generalization bounds for approximating $Q$ using i.i.d. samples. We then show how to compute $\sup_{c \in \mathcal{C}}$ for several classes of representation statistics $\mathcal{C}$.

**Error in approximation $Q$ via a curated dataset.**  Proposition 1, proved in Appendix C, bounds the deviation between the empirical MPR computed over the curated dataset $\mathcal{D}_C$ drawn i.i.d. from reference distribution $Q$ and the true MPR measured over $Q$.

**Proposition 1** (Generalization Gap of MPR). *Let $\mathcal{R}(q) = \{\boldsymbol{x}_i^r\}_{i=1}^k$ be a set of $k$ retrieved samples, $\mathcal{D}_C = \{\boldsymbol{x}_i^c\}_{i=1}^m$ be a curated dataset comprised of $m$ i.i.d. samples from a target representation distribution $Q$, and $\delta > 0$. If $\mathcal{C} = \{c : \mathcal{R}^d \times \mathcal{G} \to \{-1, 1\}\}$ with Rademacher complexity $\mathcal{R}_m(\mathcal{C})$ then, with probability at least $1 - \delta$,*

$$|\mathrm{MPR}(\mathcal{C}, \mathcal{R}, \mathcal{D}_C) - \mathrm{MPR}(\mathcal{C}, \mathcal{R}, Q)| \leq \mathcal{R}_m(\mathcal{C}) + \sqrt{\frac{\log(2/\delta)}{8m}}. \tag{2}$$

We can extend Proposition 1 to any set of bounded functions $\mathcal{C}$ (see, e.g., bounds on empirical MMD estimates in [60]). Note that the guarantee in (2) only holds for a single set of retrieved items $\mathcal{R}$ in response to a query. Proposition 2 provides a bound on the size $m$ of an i.i.d. curated dataset $\mathcal{D}_C$ that ensures an $\epsilon$-accurate estimate of MPR for a set of $M$ queries.

**Proposition 2** (Query Budget Guarantee). *Consider any set of $M$ queries $\mathcal{Q} = \{\mathbf{q}_1, ..., \mathbf{q}_M\}$ where $M \in \mathbb{N}$. Let $\mathsf{VC}(\mathcal{C})$ denote the VC-dimension of the class $\mathcal{C}$ with range in $\{-1, 1\}$. For $\epsilon > 0$, if $\mathcal{D}_C$ consists of $m$ i.i.d. samples drawn from $Q$ where $m$ satisfies*

$$m \geq \frac{32\mathsf{VC}(\mathcal{C})}{\epsilon^2} + \frac{2\log\left(\frac{2M}{\delta}\right)}{2\epsilon^2}, \tag{3}$$

*then, with probability at least $1 - \delta$, $|\mathrm{MPR}(\mathcal{C}, \mathcal{R}, \mathcal{D}_C) - \mathrm{MPR}(\mathcal{C}, \mathcal{R}, Q)| \leq \epsilon$.*

The above results provide guidelines on the size of the curated dataset required to accurately estimate MPR relative to a true target representation distribution $Q$. If the class of representation statistics $\mathcal{C}$ is very complex (in the VC-dimension sense), then the size $m$ of the curated dataset $\mathcal{D}_C$ must be proportionally large to represent $Q$ accurately. Hence, the curation dataset should be designed having in mind (i) the number of diverse queries being asked and (ii) the complexity of the function class $\mathcal{C}$. While Proposition 2 provides a conservative bound, as generalization bounds are often not tight [68], these results nevertheless offer insight into the relationship between the complexity of $\mathcal{C}$ and the accuracy of MPR estimation.

In the remainder of the paper, we assume that MPR is computed against a fixed curated dataset $\mathcal{D}_C$ of size $m$. We consider four specific instantiations of the set of representation statistics $\mathcal{C}$: (i) $\mathcal{C}$ is closed under scalar multiplication, i.e., if $c \in \mathcal{C}$, then $\lambda c \in \mathcal{C}$ for any $\lambda \in \mathbb{R}$, (ii) $\mathcal{C}$ is a set of functions taking values in $\{-1, 1\}$; (iii) $\mathcal{C}$ consists of linear functions; and (iv) $\mathcal{C}$ consists of functions in a Reproducing Kernel Hilbert Space (RKHS). Next, we show that for cases (i) and (ii) MPR can be

computed using calls to an oracle that performs regression over $\mathcal{C}$ and, for linear functions (iii), MPR has a simple closed-form expression. Due to its technical nature, we defer the calculation of MPR for functions in an RKHS to Appendix D.2.

**Computing MPR via Mean Square Error (MSE) Minimization.** When $\mathcal{C}$ is closed under multiplication or consists of binary functions with values $-1, 1$, MPR can be expressed as an MSE minimization. This enables us to compute MPR by leveraging existing black-box oracles that perform regression under quadratic loss, such as regressors implemented in scikit-learn [69].

The key observation for expressing MPR as an MSE minimization is that MPR can be formulated as a maximum correlation problem. Consider the (row-wise) concatenation of the retrieval dataset $\mathcal{D}_R$ and the curated dataset $\mathcal{D}_C$, given by $\mathbf{X} \triangleq [\boldsymbol{x}_1^r, ..., \boldsymbol{x}_n^r, \boldsymbol{x}_1^c, ..., \boldsymbol{x}_m^c]$, where $\boldsymbol{x}_i$ is the $i$-th entry of $\mathbf{X}$. For the remainder of this section, we consider a fixed set of retrieved items $\mathcal{R}$. Let $\mathbf{a} \in \{0, 1\}^n$ be a vector indicating items that are retrieved from $\mathcal{D}_R$ for a given query, i.e., $a_i = 1 \Leftrightarrow \boldsymbol{x}_i^r \in \mathcal{R}$. Under this notation, where retrieved items are indicated by the vector $\mathbf{a}$, MPR can be reformulated as

$$\mathrm{MPR}(\mathcal{C}, \mathcal{R}, \mathcal{D}_C) = \sup_{c \in \mathcal{C}} \left| \frac{1}{k} \sum_{i=1}^{n} a_i c(\boldsymbol{x}_i^r) - \frac{1}{m} \sum_{i=1}^{m} c(\boldsymbol{x}_i^c) \right| = \sup_{c \in \mathcal{C}} \left| \sum_{i=1}^{n+m} c(\boldsymbol{x}_i) \widetilde{a}_i \right|, \quad (4)$$

where $\widetilde{\mathbf{a}} \in \mathbb{R}^{n+m}$ has $i$-th entry given by $\widetilde{a}_i \triangleq \mathbb{1}_{i \leq n} \frac{a_i}{k} - \mathbb{1}_{i > n} \frac{1}{m}$. This reformulation will be useful for explicitly casting MPR-constrained retrieval as an optimization problem.

When $\mathcal{C}$ consists of binary functions in range $\{-1, 1\}$, MPR is equivalent to regression over $\mathcal{C}$ of $\widetilde{\mathbf{a}}$ under quadratic loss, since

$$\inf_{c \in \mathcal{C}} \sum_{i=1}^{n+m} (c(\boldsymbol{x}_i) - \widetilde{a}_i)^2 = n + m + \widetilde{\mathbf{a}}^\intercal \widetilde{\mathbf{a}} + 2 \sup_{c \in \mathcal{C}} \sum_{i=1}^{n+m} c(\boldsymbol{x}_i) \widetilde{a}_i. \quad (5)$$

The absolute value in (4) can be recovered by regressing $-\widetilde{\mathbf{a}}$ instead of $\widetilde{\mathbf{a}}$.

Most classes of regression functions in $\mathbb{R}$ are closed under scalar multiplication (e.g., multiplying the output of a decision tree regressor by a constant still yields a decision tree regressor). However, in this case, it follows from (4) that MPR is unbounded. Even for bounded $\mathcal{C}$, without proper normalization MPR can scale with dataset size, which is undesirable for a representational measure. We constrain $\mathcal{C}$ to a set of normalized functions for MPR to be bounded and independent of dataset size. This constraint also allows us to cast MPR as an MSE minimization over $\mathcal{C}$. We formally state this result next, proven in Appendix C, which will be applied in Section 4 to develop a cutting-plane-based algorithm for ensuring MPR in retrieval called MOPR.

**Proposition 3.** *Let $\mathcal{C}' \triangleq \{c \in \mathcal{C} \mid \sum_{i=1}^{m+n} c(\boldsymbol{x}_i)^2 = \frac{mk}{m+k}\}$ where $\mathcal{C}$ is closed under scalar multiplication. Let $c(\mathbf{X}) = [c(\mathbf{x_1}), ..., c(\mathbf{x_{n+m}})]$. Then $0 \leq \mathrm{MPR}(\mathcal{C}', \mathcal{R}, \mathcal{D}_C) \leq 1$, and for any*

$$c^* \in \arg\inf_{c \in \mathcal{C}} \sum_{i=1}^{n+m} (c(\boldsymbol{x}_i) - \widetilde{a}_i)^2, \quad (6)$$

*let $\hat{c} : \mathcal{X} \to \mathbb{R}$ be defined as $\hat{c}(\mathbf{x}) = \frac{\sqrt{mk/(m+k)}}{\|c^*(\mathbf{X})\|_2} c^*(\mathbf{x})$. Then, we have that*

$$\hat{c} \in \arg\sup_{c \in \mathcal{C}'} \left| \sum_{i=1}^{n+m} c(\boldsymbol{x}_i) \widetilde{a}_i \right|. \quad (7)$$

**Computing MPR for Bounded-Norm Linear Regression.** Recall that each item in the retrieval or curated datasets is of the form $x_i = (e_i, g_i)$, where $e_i$ is an embedding and $g_i$ are labels. When $x_i \in \mathbb{R}^l$ (i.e., the labels have numerical values), we can consider $\mathcal{C}$ as a linear set of functions over retrieved items. In this case, MPR enjoys a closed-form expression, as stated in the next proposition.

**Proposition 4.** *Let items in the retrieval and curated datasets be vectors in $\mathbb{R}^l$ for $l < m + n$ and $\mathcal{C} = \{x \mapsto w^\intercal x \mid w \in \mathbb{R}^k\}$. Moreover, let $\mathbf{X} \in \mathbb{R}^{(m+n) \times l}$ be the matrix formed by concatenating items in the retrieval and curated dataset $\mathcal{D}_R$ and $\mathcal{D}_C$, respectively. For $\mathcal{C}'$ in Proposition 3, we have*

$$\mathrm{MPR}(\mathcal{C}', \mathcal{R}, \mathcal{D}_C) = \sqrt{\frac{mk}{m+k}} \left\| \mathbf{U}_l^\intercal \widetilde{\mathbf{a}} \right\|_2, \quad (8)$$

---

**Algorithm 1** `MOPR` (Multi-group Optimized Proportional Retrieval)

---

**Input**: $\mathcal{D}_R$, query $\mathbf{q}$, number of iterations $T$, number of items to retrieve $k$, MPR constraint $\rho$, $\texttt{Oracle}(\mathbf{a})$ that computes solves (4) and returns $c \in \mathcal{C}$ that approximates the sup.

1: $\widetilde{\mathcal{C}} \leftarrow \emptyset$, $s_i = \kappa(\boldsymbol{x}_i^r, q)$
2: **for** $\texttt{iter} \in \{1, 2, ..., T\}$ **do**
3:     $\mathbf{a} \leftarrow \arg\max_{\mathbf{a} \in [0,1]^n} \mathbf{a}^\intercal \mathbf{s}$ subject to $\left| \frac{1}{k} \sum_{i=1}^n a_i c(\boldsymbol{x}_i^r) - \frac{1}{m} \sum_{i=1}^m c(\boldsymbol{x}_i^c) \right| \le \rho$ for $c \in \widetilde{\mathcal{C}}$
4:     $(\texttt{mpr\_violation}, c) \leftarrow \texttt{Oracle}(\mathbf{a})$                                                   ▷ Solve (4)
5:     **if** $\texttt{mpr\_violation} \le \rho$ or $\texttt{iter} = T$ **then**
6:         return $\mathcal{R} \leftarrow \{\boldsymbol{x}_i^r \in \mathcal{D}_R$ corresponding to $k$ largest entries of $a_i\}$
7:     **else**
8:         $\widetilde{\mathcal{C}} \leftarrow \widetilde{\mathcal{C}} \cup \{c\}$
9:     **end if**
10: **end for**

---

*where* $\mathbf{X} = \mathbf{U}\mathbf{\Sigma}\mathbf{V}^\intercal$ *is the SVD of* $\mathbf{X}$ *and* $\mathbf{U}_l \in \mathbb{R}^{(m+n) \times l}$ *are the left singular vectors in* $\mathbf{U}$ *corresponding to the top-$l$ largest singular values.*[1]

Proposition 4 can be directly adapted to linear functions defined only over embeddings or a subset of features of retrieved items. The closed-form expression for MPR in (8) also allows MPR-constrained retrieval to be computed by a quadratic program, as discussed in Appendix D.1.

## 4 Promoting Multi-Group Proportional Representation in Retrieval Tasks

We develop an optimization framework for retrieving the $k$ most similar items in a vector database to a given query $\mathbf{q}$ while satisfying a target MPR threshold of $\rho$. We formulate the retrieval goal as maximizing the average similarity between a query and retrieved items, given by $s_i = \kappa(\boldsymbol{x}_i^r, \mathbf{q})$ (recall Section 2 for notation), where $\kappa(\boldsymbol{x}_i^r, \mathbf{q})$ is given by cosine similarity in our experiments. The problem of maximizing utility in retrieval while satisfying an MPR constraint (expressed as Eq. (4)) can be formulated as the integer program:

$$\max_{\mathbf{a} \in \{0,1\}^n} \mathbf{a}^\intercal \mathbf{s} \text{ subject to } \sup_{c \in \mathcal{C}} \left| \frac{1}{k} \sum_{i=1}^n a_i c(\boldsymbol{x}_i^r) - \frac{1}{m} \sum_{i=1}^m c(\boldsymbol{x}_i^c) \right| \le \rho, \quad \sum_{i=1}^k a_i = k. \tag{9}$$

When $\mathcal{C}$ is given by normalized linear functions as in Proposition 4, the integer constraints can be relaxed to $\mathbf{a} \in [0,1]^n$ and the optimization can be approximated via standard quadratic solvers; see Appendix D.1.

**Multi-Group Optimized Proportional Retrieval.** We approximate (9) via the Multi-Group Optimized Proportional Retrieval Algorithm (`MOPR`), described in Algorithm 1. `MOPR` is essentially a cutting-plane method that aims to find a set of $k$ items that maximize utility while approximately satisfying an MPR constraint of $\rho$. The algorithm iterates between (i) a call to an oracle that computes MPR and returns a function $c \in \mathcal{C}$ that achieves the supremum in (4) and (ii) a call to a linear program (LP) solver that approximates the top-$k$ most similar items to a query subject to linear constraints on the violation measured by $c$. Each oracle call adds an additional constraint to the LP solver which, in turn, approximates the top-$k$ items under an increasing set of constraints. The solution of the LP is rounded to satisfy the integer constraint $\mathbf{a} \in \{0,1\}^n$. In our implementation, we assume that MPR can be computed via MSE minimization (cf. Section 3) – in which case we consider the normalized class $\mathcal{C}'$. The oracle call consists of running a black-box quadratic loss minimization over $\mathcal{C}$ and normalizing.

Interestingly, we observe that, despite relaxing the integer constraints in each LP call, the solution to the relaxed problem is very sparse and (after rounding) approximates well the solution of the ideal integer program (9) for moderate values of $\rho$. However, the method can fail to accurately approximate the IP solution when $\rho$ is small. We present a more detailed analysis of `MOPR` in Appendix D, which discusses potential convergence issues with the cutting plane method as well as stopping conditions.

---

[1] For analysis of functions in an RKHS, see Appendix D.2.

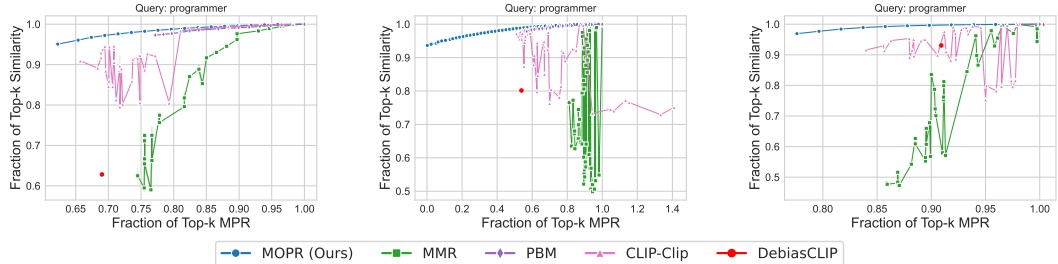

**Figure 1:** Cosine similarity vs MPR for query "A photo of a programmer" with $k = 50$ images retrieved. From left to right: CelebA, UTKFaces, Occupations. Values are normalized so MPR and similarity when retrieving the top-$k$ items is the point (1,1). MOPR Pareto-dominates baselines and significantly closes the MPR gap.

## 5 Numerical Experiments

In this section, we show that MOPR is effective in promoting more proportional representation across intersectional groups while preserving similarity between retrieved items and a given query. Notably, MOPR Pareto-dominates competing benchmarks in terms of achieved MPR gap and utility.

**Datasets.** We conduct retrieval over three image datasets of faces: **CelebA** [38], which includes labels for gender, age, and various other attributes, **UTKFace** [40], which contains gender, age, and race attributes, and **Occupations** [2], which contains gender attributes. We compute MPR for equal representation by constructing a synthetic dataset $\mathcal{D}_C$ balanced over all attributes and for proportional representation using **FairFace** [39] as the curated dataset $\mathcal{D}_C$, since it is a carefully designed dataset of faces with subgroup attributes for race, gender, and age. In all cases, each dataset entry consists of an image (CLIP) embedding $\mathbf{e}_i$ and a set of labels $\mathbf{g}_i$.

**Benchmarks.** We compare our method with four baselines. DebiasCLIP [20] modifies text queries with a prepended learned embedding for debiased retrieval but does not allow for tunable control of the representation-similarity trade-off, so it only results in a single retrieval set over which we compute similarity and MPR. CLIP-Clip [18] trims features from CLIP embeddings that have high "mutual information" with gender, allowing for control of how many features are clipped from the embedding. PBM [29] post-processes CLIP embeddings to mitigate bias by predicting gender attributes and subsampling from each gender equally. PBM also allows tuning the representation-similarity tradeoff by controlling the likelihood of sampling at random or in a balanced manner from the retrieval dataset. Finally, the state-of-the-art for fair retrieval first constructs PATHS embeddings [27], which use a set of adjective-noun keywords to capture a broad set of identity vectors in vision-language models, and then greedily samples using the MMR [45] algorithm to navigate the representation-similarity tradeoff. However, the authors' method of computing PATHS embeddings is not public, so we used CLIP embeddings to replicate their algorithm. We refer to this method as MMR. For a detailed discussion of these algorithms and their approaches to representation, see Appendix E.

**Experimental Setup.** We consider three classes of representation statistics $\mathcal{C}$: linear regression, decision trees, and MLPs (the last two are presented in Appendix G). In both cases, we compute MPR over $\mathcal{C}'$, i.e., $\mathcal{C}$ normalized over the retrieval and curated datasets (see Prop. 3). We report results over 10 queries for occupations suggested by ChatGPT 3.5 (Appendix H) and an additional 45 occupations for the the Occupations dataset. To accelerate retrieval, we query each of our retrieval datasets for the top 10k items according to the prompt ``A photo of a {query}" in terms of raw similarity using FAISS [70]. We similarly retrieve the top 10k items from FairFace as the query-conditioned curated dataset $\mathcal{D}_C$. One reason we first filter for the top 10k items is the runtime of MMR, which is a greedy algorithm that traverses the full dataset for each retrieval $k$ (even for 10k entries and $k = 50$ retrieved items, MMR takes hours to run for 10 queries).

In order to evaluate representation over interpretable population groups, we consider that representation statistics $\mathcal{C}$ are computed only over labels $\mathbf{g}_i$ given in FairFace. These labels are not present in all retrieval datasets. Thus, for CelebA and Occupations, we train linear probes on FairFace's CLIP embeddings to predict their race and age attributes, and when using UTKFace, we map from FairFace's race labels to UTKFace's race labels by mapping `Southeast Asian` and `East Asian`

**Table 1:** Retrieval averaged over 10 queries with 50 retrievals on UTKFace dataset [40] for `MOPR`, Top-$k$, and MMR [27]. Entries indicate average percentage representation in retrieved items. `MOPR` is able to balance representation across classes, whereas Top-$k$ and MMR miss intersectional groups (highlighted in red).

| | | White | Black | Asian | Indian | Others | Sum |
|---|---|---|---|---|---|---|---|
| **Top-$k$** | Male | $21.2 \pm 11.84$ | $13.2 \pm 7.70$ | $10.8 \pm 7.54$ | $21.2 \pm 20.70$ | $1.4 \pm 1.56$ | $\mathbf{67.8 \pm 23.72}$ |
| | Female | $17.2 \pm 21.04$ | $6.6 \pm 6.14$ | $4.0 \pm 2.82$ | $2.8 \pm 2.56$ | $1.6 \pm 1.50$ | $\mathbf{32.2 \pm 23.72}$ |
| | Sum | $\mathbf{38.4 \pm 17.18}$ | $\mathbf{19.8 \pm 12.56}$ | $\mathbf{14.8 \pm 7.38}$ | $\mathbf{24 \pm 19.38}$ | $\mathbf{3.0 \pm 2.04}$ | |
| **MMR** | Male | $28.8 \pm 8.96$ | $11.0 \pm 4.58$ | $10.2 \pm 4.04$ | $10.8 \pm 2.40$ | $3.8 \pm 1.06$ | $\mathbf{64.6 \pm 7.74}$ |
| | Female | $16.2 \pm 9.06$ | $7.6 \pm 3.66$ | $5.0 \pm 3.00$ | $4.0 \pm 1.26$ | $2.6 \pm 1.56$ | $\mathbf{35.4 \pm 7.74}$ |
| | Sum | $\mathbf{45.0 \pm 7.76}$ | $\mathbf{18.6 \pm 5.38}$ | $\mathbf{15.2 \pm 5.82}$ | $\mathbf{14.8 \pm 3.12}$ | $\mathbf{6.4 \pm 1.96}$ | |
| `MOPR` (**Ours**) | Male | $7.8 \pm 3.74$ | $10.2 \pm 2.90$ | $13.0 \pm 2.56$ | $11.8 \pm 2.44$ | $7.2 \pm 3.70$ | $\mathbf{50 \pm 0}$ |
| | Female | $12.2 \pm 3.74$ | $9.8 \pm 2.90$ | $7.0 \pm 2.56$ | $8.2 \pm 2.44$ | $12.8 \pm 3.70$ | $\mathbf{50 \pm 0}$ |
| | Sum | $\mathbf{20 \pm 0}$ | $\mathbf{20 \pm 0}$ | $\mathbf{20 \pm 0}$ | $\mathbf{20 \pm 0}$ | $\mathbf{20 \pm 0}$ | |

to `Asian` and `Middle Eastern` and `Latino_Hispanic` to `Other`. While the fair ML community should engage in broader discussions addressing the ethics of predicting sensitive attributes from CLIP embeddings [71] (especially given that CLIP itself has been found to be biased) and the issues surrounding the grouping and re-grouping of diverse racial identities, we acknowledge that these are larger-scale issues that our work does not presume to address.

We retrieve $k = 50$ items for each of the above queries. For a given function class and query, we compute the baseline MPR and average cosine similarity given by the top 50 most similar items. Then, we conduct a parameter sweep over $\rho$ starting from this max-MPR value, inputting each value to `MOPR` in Algorithm 1. We normalize our results such that the Top-$k$ MPR and similarity are the point (1,1) on each graph, and each point measures the fraction of Top-$k$ MPR and similarity.

**Results.** In Fig. 1 we report results for proportional representation for the query ''A photo of a programmer'' with respect to FairFace for linear regression models, with additional results for other queries, as well as decision trees and MLPs in Appendix G (Figs. 8-19). We observe that `MOPR` Pareto-dominates benchmark methods, and is able to preserve similarity while reaching lower levels of MPR. Methods based on directly modifying embeddings or queries for a single group attribute such as CLIP-Clip and DebiasCLIP only partially reduce MPR at a high utility cost. This is because these methods do not modify the retrieval algorithm but use an unbiased embedding and hope by chance that the retrieved items will be representational. The most competitive method to ours is the retrieval algorithm MMR, which is our attempt to replicate PATHS. Though MMR is competitive to ours for large MPR, it fails to achieve small values of MPR relative to `MOPR`. It is also notable that `MOPR` can drive MPR to near zero for UTKFace, yet gaps remain in CelebA and Occupations, indicating this gap may be caused by the use of probes to estimate group attribute labels.

In Table 1, we construct a synthetic curation set with an equal number of each group attribute. This allows us to evaluate the performance of `MOPR` in achieving equal representation. We conduct a similar hyperparameter sweep as above, and take the retrieval set with the minimum MPR averaged over all 10 queries to fairly compare the best possible representation of each method.[2] Our results demonstrate that `MOPR` can exactly achieve our equal representation goals when provided with an equally-balanced curation dataset, whereas prior baselines (see DebiasCLIP, CLIP-Clip, and PBM in Appendix Table 5) underrepresent several intersectional groups.

## 6   Concluding Remarks

In this work, we introduced a novel retrieval metric called Multi-group Proportional Representation (MPR) and developed algorithms to ensure MPR in retrieval tasks. By measuring deviations in a set of representation statistics, MPR provides a scalable approach to quantifying and enforcing proportional representation for complex, intersectional groups. We analyzed the generalization properties and realizable regimes of MPR and, through experiments in image retrieval, demonstrated its favorable utility-fairness trade-off compared to existing fair retrieval algorithms.

---

[2]Aggregate results should be handled with care. As each query-conditioned retrieval setting results in a different distribution, average-case performance does not preclude the existence of a query where baseline methods achieve superior results. See Limitations (Appendix B) for further discussion.

# 7 Acknowledgements

The authors would like to thank the discussions with Filipe Goulart Cabral. This material is based upon work supported by the National Science Foundation under awards CAREER-1845852, CIF-1900750, CIF-2231707, and CIF-2312667, FAI-2040880, and also by the National Science Foundation Graduate Research Fellowship under Grant No. DGE-2140743. The views expressed here are those of the authors and do not reflect the official policy or position of the funding agencies.

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

**Supplementary material to the paper** *Multi-Group Proportional Representation in Retrieval*

In this supplement to the paper, we begin in Section A and Section B by noting some ethical considerations and limitations for our work, respectively. Then, we present our proofs of Propositions 1, 3, 2, and 4 in Appendix C. In Appendix D, we discuss the MOPR algorithm, its convergence aspects, and MOPR for linear functions and functions belonging to an RKHS. We also compare the linear relaxation to the integer program optimization problem in Equation 9 empirically and similarly compare the result of MOPR to directly solving the closed form linear program in the case of linear models. In Appendix E, we describe the baselines reported in this paper and provide further retrieval statistics. In particular, we describe the algorithms CLIP-clip Wang et al. [18], CLIP-debias Berg et al. [20], Post-Hoc Bias Mitigation ([29] and PATHS [27]. In F, we provide a brief discussion about proportional representation in political sciences, give a few historical comments about social choice theory, and discuss the contribution given by [37]. Then, in Appendix G, we include an additional table of results for equal representation with the other baseline methods (Table 5). We also include plots for additional queries for the linear regression model (Figs. 8, 9, and 10) for as well as two additional models, 1) decision tree models in Figs. 11, 12, and 13 and for 2) multilayer perceptrons in Figs. 14, 15, and 16. We provide additional details on the experiment setting in Appendix H. Finally, we include some example sets of retrieved images for qualitative comparison.

## A  Ethics considerations

It is important to consider the long-term societal impact of enforcing MPR in real-world retrieval systems to understand its potential benefits and risks. First, applying MPR, such as with MOPR in deployment settings, could result in "ethics-washing"–where a company claims their system is fair because they measured MPR when, in fact, there are still representational harms–or could provide users and developers with a false sense of fairness. Second, MPR is just one of many ways of analyzing representation, and as a statistical metric, it fails to consider aspects such as human perceptions of representation. There are legal and regulatory risks with overreliance on a single metric for fairness, especially if this metric is used to inform policy and decision-making. As mentioned previously, any biases present in the curation dataset will propagate to the results and downstream uses of MPR; furthermore, the choice of what dataset and what attributes to consider is an inherently social and political choice made by humans, subject to its own limitations and biases. Finally, the introduction of MPR in real-world retrieval systems could introduce new stereotypes or erase identities if not conducted properly.

## B  Limitations

Our aggregate results (e.g., Table 1) should be approached with caution, as each MPR-similarity curve is sampled from a different distribution of data conditioned on a query, limiting our ability to draw statistical conclusions. This does not preclude the existence of a query where a competing method may achieve an MPR-similarity trade-off point not achieved by MOPR. However, we note that MOPR is specifically optimized to promote MPR while preserving similarity, explaining its favorable performance. In all instances observed by the authors, MOPR Pareto dominates competing benchmarks and was the most successful method in achieving low values of MPR while maintaining high levels of similarity with a given query. Second, while MPR is designed to handle a large number of groups, the computational complexity of the proposed cutting-plane algorithm in the case of a general function class $\mathcal{C}$ can be a concern for high-dimensional feature spaces. Developing more efficient algorithms or approximation techniques could help address this issue. Finally, MPR is inherently limited by the curated dataset. If this dataset is biased against a given group, this bias will be propagated by MOPR. The creation and selection of curated datasets must carefully account for stakeholder interest and target representation statistics. Additionally, we note that not every combination of intersectional identities (such as "white male" as opposed to "black female") may be considered intersectional from a sociological perspective, and furthermore, these datasets do not contain multiracial labels. We do not presume to address representational issues in ML datasets, and instead, our experiments aim to demonstrate the flexibility and scalability of the MPR metric in handling complex intersectional groups. We hope that MPR will only prove more useful as datasets become more representative.

# C Proofs from Section 3

We start by proving Proposition 1 from Section 3 which bounds the error in MPR estimates when $Q$ is approximated via a curated dataset $\mathcal{D}_C$. We do so by using tools from standard Rademacher complexity results in statistical learning theory, e.g., [72]. See also [73, Chapter 4]. We restate the theorems here to facilitate readability. We start with a lemma about the difference of suprema.

**Lemma 1.** *Let $P$ and $P'$ be two distributions over $\mathcal{X}$ and $\mathcal{C} = \{c : \mathcal{X} \to [0, 1]\}$. Then*

$$\sup_{c \in \mathcal{C}} \mathbb{E}_P[c(X)] - \sup_{c \in \mathcal{C}} \mathbb{E}_{P'}[c(X)] \leq \sup_{c \in \mathcal{C}} \left\{ \mathbb{E}_P[c(X)] - \mathbb{E}_{P'}[c(X)] \right\}. \tag{10}$$

*Proof.*

$$\begin{aligned}
&\sup_{c \in \mathcal{C}} \mathbb{E}_P[c(X)] - \sup_{c \in \mathcal{C}} \mathbb{E}_{P'}[c(X)], \\
&= \sup_{c \in \mathcal{C}} \left\{ \mathbb{E}_P[c(X)] - \mathbb{E}_{P'}[c(X)] + \mathbb{E}_{P'}[c(X)] \right\} - \sup_{c \in \mathcal{C}} \mathbb{E}_{P'}[c(X)], \\
&\leq \sup_{c \in \mathcal{C}} \left\{ \mathbb{E}_P[c(X)] - \mathbb{E}_{P'}[c(X)] \right\} + \sup_{c \in \mathcal{C}} \mathbb{E}_{P'}[c(X)] - \sup_{c \in \mathcal{C}} \mathbb{E}_{P'}[c(X)], \\
&\leq \sup_{c \in \mathcal{C}} \left\{ \mathbb{E}_P[c(X)] - \mathbb{E}_{P'}[c(X)] \right\}.
\end{aligned}$$

$\square$

**Proposition 1** (Generalization Gap of MPR). *Let $\mathcal{R}(q) = \{\boldsymbol{x}_i^r\}_{i=1}^k$ be a set of $k$ retrieved samples, $\mathcal{D}_C = \{\boldsymbol{x}_i^c\}_{i=1}^m$ be a curated dataset comprised of $m$ i.i.d. samples from a target representation distribution $Q$, and $\delta > 0$. If $\mathcal{C} = \{c : \mathcal{R}^d \times \mathcal{G} \to \{-1, 1\}\}$ with Rademacher complexity $\mathcal{R}_m(\mathcal{C})$ then, with probability at least $1 - \delta$,*

$$|\text{MPR}(\mathcal{C}, \mathcal{R}, \mathcal{D}_C) - \text{MPR}(\mathcal{C}, \mathcal{R}, Q)| \leq \mathcal{R}_m(\mathcal{C}) + \sqrt{\frac{\log(2/\delta)}{8m}}. \tag{2}$$

*Proof.* First, note that

$$\text{MPR}(\mathcal{C}, \mathcal{R}, \mathcal{D}_C) - \text{MPR}(\mathcal{C}, \mathcal{R}, Q) \tag{11}$$

$$= \sup_{c \in \mathcal{C}} \left| \frac{1}{k} \sum_{i=1}^k c(\boldsymbol{x}_i^r) - \frac{1}{m} \sum_{i=1}^m c(\boldsymbol{x}_i^c) \right| - \sup_{c \in \mathcal{C}} \left| \frac{1}{k} \sum_{i=1}^k c(\boldsymbol{x}_i^r) - \mathbb{E}_Q[c(X)] \right|, \tag{12}$$

$$\leq \sup_{c \in \mathcal{C}} \left\{ \left| \frac{1}{k} \sum_{i=1}^k c(\boldsymbol{x}_i^r) - \frac{1}{m} \sum_{i=1}^m c(\boldsymbol{x}_i^c) \right| - \left| \frac{1}{k} \sum_{i=1}^k c(\boldsymbol{x}_i^r) - \mathbb{E}_Q[c(X)] \right| \right\}, \tag{13}$$

$$\leq \sup_{c \in \mathcal{C}} \left| \left| \frac{1}{k} \sum_{i=1}^k c(\boldsymbol{x}_i^r) - \frac{1}{m} \sum_{i=1}^m c(\boldsymbol{x}_i^c) \right| - \left| \frac{1}{k} \sum_{i=1}^k c(\boldsymbol{x}_i^r) - \mathbb{E}_Q[c(X)] \right| \right|, \tag{14}$$

$$\leq \sup_{c \in \mathcal{C}} \left| \frac{1}{m} \sum_{i=1}^m c(\boldsymbol{x}_i^c) - \mathbb{E}_Q[c(X)] \right|. \tag{15}$$

Where (13) comes from Lemma 1 and (15) comes from the triangular inequality.

Additionally, by switching $\text{MPR}(\mathcal{C}, \mathcal{R}, Q)$ by $\sup_{c \in \mathcal{C}} \left| \frac{1}{k} \sum_{i=1}^k c(x_i) - \mathbb{E}_Q[c(X)] \right|$ in (11) and repeating the previous argument, we conclude that:

$$\left| \text{MPR}(\mathcal{C}, \mathcal{R}, \mathcal{D}_C) - \sup_{c \in \mathcal{C}} \left| \frac{1}{k} \sum_{i=1}^k c(x_i) - \mathbb{E}_Q[c(X)] \right| \right| \leq \sup_{c \in \mathcal{C}} \left| \frac{1}{m} \sum_{i=1}^m c(\boldsymbol{x}_i^c) - \mathbb{E}_Q[c(X)] \right|. \tag{16}$$

Therefore, from (16) we show

$$\Pr\left[\left|\left|\mathrm{MPR}(\mathcal{C},\mathcal{R},\mathcal{D}_C) - \sup_{c\in\mathcal{C}}\left|\frac{1}{k}\sum_{i=1}^{k}c(x_i) - \mathbb{E}_Q[c(X)]\right|\right|\right| \geq 2\mathcal{R}_m(\mathcal{C}) + \sqrt{\frac{\log\left(\frac{2}{\delta}\right)}{2m}}\right], \qquad (17)$$

$$\leq \Pr\left[\sup_{c\in\mathcal{C}}\left|\frac{1}{m}\sum_{i=1}^{m}c(\boldsymbol{x}_i^c) - \mathbb{E}_Q[c(X)]\right| \geq 2\mathcal{R}_m(\mathcal{C}) + \sqrt{\frac{\log\left(\frac{2}{\delta}\right)}{2m}}\right], \qquad (18)$$

$$\leq \Pr\left[\sup_{c\in\mathcal{C}}\left|\frac{1}{m}\sum_{i=1}^{m}\frac{c(\boldsymbol{x}_i^c)}{2} - \mathbb{E}_Q\left[\frac{c(X)}{2}\right]\right| \geq \mathcal{R}_m(\mathcal{C}) + \sqrt{\frac{\log\left(\frac{2}{\delta}\right)}{8m}}\right] \leq \delta, \qquad (19)$$

here (19) comes from the generalization bound using Rademacher complexity from [72] and the fact that $\frac{c(x)}{2}\in[-\frac{1}{2},\frac{1}{2}]$ hence the image of $\frac{c}{2}$ is in an interval of size 1. $\qquad\square$

We now prove Proposition 2, which provides a bound on the size $m$ of an i.i.d. curated dataset $\mathcal{D}_C$ that ensures an $\epsilon$-accurate estimate of MPR for a set of $M$ queries.

**Proposition 2** (Query Budget Guarantee). *Consider any set of $M$ queries $\mathcal{Q} = \{\mathbf{q}_1, ..., \mathbf{q}_M\}$ where $M\in\mathbb{N}$. Let $\mathsf{VC}(\mathcal{C})$ denote the VC-dimension of the class $\mathcal{C}$ with range in $\{-1,1\}$. For $\epsilon > 0$, if $\mathcal{D}_C$ consists of $m$ i.i.d. samples drawn from $Q$ where $m$ satisfies*

$$m \geq \frac{32\mathsf{VC}(\mathcal{C})}{\epsilon^2} + \frac{2\log\left(\frac{2M}{\delta}\right)}{2\epsilon^2}, \qquad (3)$$

*then, with probability at least $1 - \delta$, $|\mathrm{MPR}(\mathcal{C},\mathcal{R},\mathcal{D}_C) - \mathrm{MPR}(\mathcal{C},\mathcal{R},Q)|\leq \epsilon$.*

*Proof.* First, recall that the Rademacher complexity $\mathcal{R}_m(\mathcal{C})$ is upper bounded by the VC-dimension of $\mathcal{C}$ by the relation

$$\mathcal{R}_m(\mathcal{C}) \leq \sqrt{\frac{2\mathsf{VC}(\mathcal{C})\log\left(\frac{em}{\mathsf{VC}(\mathcal{C})}\right)}{m}}, \qquad (20)$$

where $e$ is Euler's number.

Denote the samples retrieved for a given query $q$ by $\mathcal{R}(q) = \{\boldsymbol{x}_i^r(q)\}_{i=1}^{k}$. From Proposition 1 we have that for all $\delta^*$

$$\Pr\left[|\mathrm{MPR}(\mathcal{C},\mathcal{R},\mathcal{D}_C) - \mathrm{MPR}(\mathcal{C},\mathcal{R},Q)| \geq \mathcal{R}_m(\mathcal{C}) + \sqrt{\frac{\log\left(\frac{2}{\delta^*}\right)}{8m}}\right] \leq \delta^*.$$

Therefore, using the previous equation if we take $\delta^* = \frac{\delta}{M}$ and denote $\mathcal{R}(q) = \mathcal{R}$ we have that

$$\Pr\left[\sup_{q\in\mathcal{Q}}|\mathrm{MPR}(\mathcal{C},\mathcal{R},\mathcal{D}_C) - \mathrm{MPR}(\mathcal{C},\mathcal{R},Q)| \geq \mathcal{R}_m(\mathcal{C}) + \sqrt{\frac{\log\left(\frac{2}{\delta^*}\right)}{8m}}\right],$$

$$\leq \sum_{q\in\mathcal{Q}}\Pr\left[|\mathrm{MPR}(\mathcal{C},\mathcal{R},\mathcal{D}_C) - \mathrm{MPR}(\mathcal{C},\mathcal{R},Q)| \geq \mathcal{R}_m(\mathcal{C}) + \sqrt{\frac{\log\left(\frac{2}{\delta^*}\right)}{8m}}\right],$$

$$\leq \sum_{q\in\mathcal{Q}}\delta^* = \frac{\delta M}{M} = \delta. \qquad (21)$$

Hence, we have that, with probability at least $1 - \delta$ for all $q \in \mathcal{Q}$

$$\sup_{c\in\mathcal{C}}\left|\frac{1}{k}\sum_{i=1}^{k}c(\boldsymbol{x}_i^r(q)) - \mathbb{E}_Q[c(X)]\right| \leq \mathrm{MPR}(\mathcal{C},\mathcal{R}(q),\mathcal{D}_C) + \mathcal{R}_m(\mathcal{C}) + \sqrt{\frac{\log\left(\frac{2M}{\delta}\right)}{8m}}, \qquad (22)$$

$$\leq \mathrm{MPR}(\mathcal{C}, \mathcal{R}(q), \mathcal{D}_C) + \sqrt{\frac{2\mathrm{VC}(\mathcal{C}) \log\left(\frac{em}{\mathrm{VC}(\mathcal{C})}\right)}{m}} + \sqrt{\frac{\log\left(\frac{2M}{\delta}\right)}{8m}}.$$
(23)

Where the inequality in (23) comes from the upper bound in (20).

Moreover, if $m \geq \frac{32\mathrm{VC}(\mathcal{C})}{\epsilon^2}$ then $\sqrt{\frac{2\mathrm{VC}(\mathcal{C}) \log\left(\frac{em}{\mathrm{VC}(\mathcal{C})}\right)}{m}} \leq \frac{\epsilon}{2}$ and if $m \geq \frac{\log\left(\frac{2M}{\delta}\right)}{2\epsilon^2}$ then $\sqrt{\frac{\log\left(\frac{2M}{\delta}\right)}{8m}} \leq \frac{\epsilon}{2}$. Therefore, we conclude that if $m$ is such that

$$m \geq \frac{32\mathrm{VC}(\mathcal{C})}{\epsilon^2} + \frac{\log\left(\frac{2M}{\delta}\right)}{2\epsilon^2},$$
(24)

then

$$\sup_{c\in\mathcal{C}} \left| \frac{1}{k} \sum_{i=1}^{k} c(\boldsymbol{x}_i^r(q)) - \mathbb{E}_Q[c(X)] \right| \leq \mathrm{MPR}(\mathcal{C}, \mathcal{R}(q), \mathcal{D}_C) + \epsilon.$$
(25)

$\square$

As described in Section 3, such bounds tend to be conservative and these bounds can be improved by doing a sharper analysis with PAC-Bayesian [74] or mutual information bounds [75]. See also [76] and [77].

We now prove Proposition 3. This proposition states that for specific function classes, computing MPR is equivalent to solving an MSE regression problem. This proposition allows us to efficiently estimate MPR with standard regression models from packages such as scikit-learn [69].

**Proposition 3.** *Let $\mathcal{C}' \triangleq \{c \in \mathcal{C} \mid \sum_{i=1}^{m+n} c(\boldsymbol{x}_i)^2 = \frac{mk}{m+k}\}$ where $\mathcal{C}$ is closed under scalar multiplication. Let $c(\mathbf{X}) = [c(\mathbf{x_1}), ..., c(\mathbf{x_{n+m}})]$. Then $0 \leq \widehat{\mathrm{MPR}}(\mathcal{C}', \mathcal{R}, \mathcal{D}_C) \leq 1$, and for any*

$$c^* \in \arg\inf_{c\in\mathcal{C}} \sum_{i=1}^{n+m} (c(\boldsymbol{x}_i) - \widetilde{a}_i)^2,$$
(6)

*let $\hat{c} : \mathcal{X} \to \mathbb{R}$ be defined as $\hat{c}(\mathbf{x}) = \frac{\sqrt{mk/(m+k)}}{\|c^*(\mathbf{X})\|_2} c^*(\mathbf{x})$. Then, we have that*

$$\hat{c} \in \arg\sup_{c\in\mathcal{C}'} \left| \sum_{i=1}^{n+m} c(\boldsymbol{x}_i)\widetilde{a}_i \right|.$$
(7)

*Proof.* We begin proving $0 \leq \mathrm{MPR}(\mathcal{C}', \mathcal{R}, \mathcal{D}_C) \leq 1,$. As a shorthand, let $c(\mathbf{X}) = [c(\mathbf{x_1}), ..., c(\mathbf{x_{n+m}})]$. The Cauchy-Schwarz inequality gives,

$$\left| \sum_{i=1}^{n+m} c(\boldsymbol{x}_i)\widetilde{a}_i \right| \leq \|c(\mathbf{X})\|_2 \|\widetilde{\mathbf{a}}\|_2,$$
(26)

where $\mathbf{X} \in \mathbb{R}^{(m+n)\times l}$ is the matrix formed by the row-wise concatenation of $\mathcal{D}_R$ and $\mathcal{D}_C$ and $\widetilde{\mathbf{a}}$ is the vector $[\widetilde{a}_0, ..., \widetilde{a}_{m+n}]$. $\mathcal{C}'$ gives us a constraint on the norm of $c(\mathbf{X})$, and we can compute $\widetilde{\mathbf{a}}$ given there are $k$ terms with value $1/k$, $m$ terms with value $1/m$, and $n - k$ terms with value $0$. Also note that $\frac{mk}{m+k} = (\frac{1}{k} + \frac{1}{m})^{-1}$

$$\left| \sum_{i=1}^{n+m} c(\boldsymbol{x}_i)\widetilde{a}_i \right| \leq (\frac{1}{k} + \frac{1}{m})^{-\frac{1}{2}}(\frac{1}{k} + \frac{1}{m})^{\frac{1}{2}} = 1.$$
(27)

Next, we prove the second statement. Consider the minimizer of the MSE problem,

$$\arg\inf_{c\in\mathcal{C}} \sum_{i=1}^{n+m} (c(\boldsymbol{x}_i) - \widetilde{a}_i)^2,$$
(28)

As $\mathcal{C}$ is closed under scalar multiplication, we can equivalently write this a two-part optimization problem

$$\arg\inf_{c\in\mathcal{C}} \left( \inf_\lambda \sum_{i=1}^{n+m} (\lambda c(\boldsymbol{x}_i) - \widetilde{a}_i)^2 \right), \tag{29}$$

where the inner optimization is quadratic and yields a solution

$$\lambda^* = \frac{\sum_{i=1}^{n+m} c(\mathbf{x}_i)\widetilde{a}_i}{\|c(\mathbf{X})\|_2^2}, \tag{30}$$

as $\|c(\mathbf{X})\|_2 < \infty$. Plugging this back into Equation (29), the MSE problem becomes

$$\arg\inf_{c\in\mathcal{C}} \sum_{i=1}^{n+m} \left( \frac{\sum_{i=1}^{n+m} c(\mathbf{x}_i)\widetilde{a}_i}{\|c(\mathbf{X})\|_2^2} c(\boldsymbol{x}_i) - \widetilde{a}_i \right)^2,$$

$$= \arg\inf_{c\in\mathcal{C}} \sum_{i=1}^{n+m} \left( \frac{\langle c(\mathbf{X}), \widetilde{\mathbf{a}}\rangle^2}{\|c(\mathbf{X})\|_2^4} c(\boldsymbol{x}_i)^2 - 2\frac{\langle c(\mathbf{X}), \widetilde{\mathbf{a}}\rangle}{\|c(\mathbf{X})\|_2^2} c(\boldsymbol{x}_i)\widetilde{a}_i + \widetilde{a}_i^2 \right),$$

$$= \arg\inf_{c\in\mathcal{C}} \left( \frac{\langle c(\mathbf{X}), \widetilde{\mathbf{a}}\rangle^2}{\|c(\mathbf{X})\|_2^4} \|c(\mathbf{X})\|_2^2 - 2\frac{\langle c(\mathbf{X}), \widetilde{\mathbf{a}}\rangle}{\|c(\mathbf{X})\|_2^2} \langle c(\mathbf{X}), \widetilde{\mathbf{a}}\rangle + \sum_{i=1}^{n+m} \widetilde{a}_i^2 \right),$$

$$= \arg\inf_{c\in\mathcal{C}} -\frac{\langle c(\mathbf{X}), \widetilde{\mathbf{a}}\rangle^2}{\|c(\mathbf{X})\|_2^2} + \sum_{i=1}^{n+m} \widetilde{a}_i^2.$$

As $\widetilde{a}_i$ does not depend on $c$, this optimization is equivalent to

$$\arg\sup_{c\in\mathcal{C}} \frac{\langle c(\mathbf{X}), \widetilde{\mathbf{a}}\rangle^2}{\|c(\mathbf{X})\|_2^2} = \arg\sup_{c\in\mathcal{C}} \frac{|\langle c(\mathbf{X}), \widetilde{\mathbf{a}}\rangle|^2}{\|c(\mathbf{X})\|_2^2}.$$

Note that the supremum can be multiplied by a constant. Then we can multiply by some $\Delta > 0$ and get

$$\arg\sup_{c\in\mathcal{C}} \frac{\Delta|\langle c(\mathbf{X}), \widetilde{\mathbf{a}}\rangle|^2}{\|c(\mathbf{X})\|_2^2} = \arg\sup_{\substack{c\in\mathcal{C} \\ \|c(\mathbf{X})\|_2=\sqrt{\Delta}}} |\langle c(\mathbf{X}), \widetilde{\mathbf{a}}\rangle|^2. \tag{31}$$

In other words, for $\Delta > 0$ and

$$c^* \in \arg\sup_{c\in\mathcal{C}} \frac{|\langle c(\mathbf{X}), \widetilde{\mathbf{a}}\rangle|^2}{\|c(\mathbf{X})\|_2^2},$$

and letting $\hat{c} : \mathcal{X} \to \mathbb{R}$ be defined as $\hat{c}(\mathbf{x}) = \frac{\sqrt{\Delta}}{\|c^*(\mathbf{X})\|_2} c^*(\mathbf{x})$, we have that

$$\hat{c} \in \arg\sup_{\substack{c\in\mathcal{C}, \\ \|c(\mathbf{X})\|_2=\sqrt{\Delta}}} |\langle c(\mathbf{X}), \widetilde{\mathbf{a}}\rangle|^2 = \arg\sup_{\substack{c\in\mathcal{C}, \\ \|c(\mathbf{X})\|_2=\sqrt{\Delta}}} \left|\sum_{i=1}^{n+m} c(\boldsymbol{x}_i)\widetilde{a}_i\right|^2 = \arg\sup_{\substack{c\in\mathcal{C}, \\ \|c(\mathbf{X})\|_2=\sqrt{\Delta}}} \left|\sum_{i=1}^{n+m} c(\boldsymbol{x}_i)\widetilde{a}_i\right|. \tag{32}$$

Letting $\Delta = \frac{mk}{m+k}$, we can conclude that solving an MSE regression problem is equivalent to finding the representation statistic for MPR. $\qquad\square$

Now, we establish Proposition 4, restated below for convenience. It says that MPR enjoys a closed-form expression whenever the class $\mathcal{C}$ is given by linear functions. This proposition allows us to simplify MOPR since, in this case, the optimization problem can be cast as a quadratic program.

**Proposition 4.** *Let items in the retrieval and curated datasets be vectors in $\mathbb{R}^l$ for $l < m + n$ and $\mathcal{C} = \{x \mapsto w^\intercal x \mid w \in \mathbb{R}^k\}$. Moreover, let $\mathbf{X} \in \mathbb{R}^{(m+n)\times l}$ be the matrix formed by concatenating items in the retrieval and curated dataset $\mathcal{D}_R$ and $\mathcal{D}_C$, respectively. For $\mathcal{C}'$ in Proposition 3, we have*

$$\mathrm{MPR}(\mathcal{C}', \mathcal{R}, \mathcal{D}_C) = \sqrt{\frac{mk}{m+k}} \|\mathbf{U}_l^\intercal \widetilde{\mathbf{a}}\|_2, \tag{8}$$

*where $\mathbf{X} = \mathbf{U}\mathbf{\Sigma}\mathbf{V}^\intercal$ is the SVD of $\mathbf{X}$ and $\mathbf{U}_l \in \mathbb{R}^{(m+n)\times l}$ are the left singular vectors in $\mathbf{U}$ corresponding to the top-$l$ largest singular values.*[3]

---

[3]For analysis of functions in an RKHS, see Appendix D.2

The closed-form expression for MPR in (8) allows MPR-constrained retrieval to be approximated by a quadratic program, whose proof can be found in Appendix D. Moreover, the above proposition can be directly adapted to linear functions defined only for embeddings or a subset of features of retrieved items.

*Proof.* Since we are considering the class $\mathcal{C}' \triangleq \{c \in \mathcal{C} \mid \sum_{i=1}^{m+n} c(\boldsymbol{x}_i)^2 = \frac{mk}{m+k}\}$, where $\mathcal{C} = \{x \mapsto w^\intercal x \mid w \in \mathbb{R}^d\}$, the computation of the Multi-Group Proportional Representation metric

$$\mathrm{MPR}(\mathcal{C}', \mathcal{R}, \mathcal{D}_C) = \arg\sup_{c \in \mathcal{C}} \left| \sum_{i=1}^{n+m} c(x_i)\widetilde{a}_i \right| \tag{33}$$

boils down to the following optimization problem

$$\max_{c \in \mathcal{C}} \quad \sum_{i=1}^{n+m} (w^\intercal x)\widetilde{a}_i \tag{34}$$

$$\text{subject to} \quad \sum_{i=1}^{m+n} [w^\intercal x]^2 = \frac{mk}{m+k},$$

which can be written as

$$\max_{w \in \mathbb{R}^{n+m}} \quad \widetilde{\mathbf{a}}^T \mathbf{X} \mathbf{w} \tag{35}$$

$$\text{subject to} \quad \|\mathbf{X}\mathbf{w}\|_2^2 = \frac{mk}{m+k},$$

where $\mathbf{X} \in \mathbb{R}^{(n+m) \times l}$ is the matrix formed by concatenating items in the retrieval and curated dataset $\mathcal{D}_R$ and $\mathcal{D}_C$, respectively. Let $\mathbf{X} = \mathbf{U}\boldsymbol{\Sigma}\mathbf{V}^\intercal$ be the SVD of $\mathbf{X}$ and $\mathbf{U}_l \in \mathbb{R}^{(m+n) \times l}$ are the left singular vectors in $\mathbf{U}$ corresponding to the top-$l$ largest singular values. Then, the problem becomes

$$\max_{w \in \mathbb{R}^{m+n}} \quad \widetilde{\mathbf{a}}^T \mathbf{U}\boldsymbol{\Sigma}\mathbf{V}^\intercal \mathbf{w} \tag{36}$$

$$\text{subject to} \quad \|\boldsymbol{\Sigma}\mathbf{V}^\intercal\mathbf{w}\|_2^2 = \frac{mk}{m+k}.$$

Now, by a change of variables $\boldsymbol{\Sigma}\mathbf{V}^\intercal\mathbf{w} = \widetilde{\mathbf{w}}$ and $\mathbf{U}_l^\intercal \widetilde{\mathbf{a}} = \mathbf{z}$

$$\max_{\widetilde{\mathbf{w}}, \mathbf{w} \in \mathbb{R}^{m+n}} \quad \mathbf{z}^T \widetilde{\mathbf{w}} \tag{37}$$

$$\text{subject to} \quad \|\widetilde{\mathbf{w}}\|_2^2 = \frac{mk}{m+k},$$

$$\boldsymbol{\Sigma}\mathbf{V}^\intercal\mathbf{w} = \widetilde{\mathbf{w}}.$$

We can use the method of Lagrange multipliers. Let us define the Lagrangian function:

$$L(\widetilde{\mathbf{w}}, \mathbf{w}, \lambda, \boldsymbol{\mu}) = \mathbf{z}^T \widetilde{\mathbf{w}} - \frac{\lambda}{2}\left(\|\widetilde{\mathbf{w}}\|_2^2 - \frac{mk}{m+k}\right) - \boldsymbol{\mu}^T(\boldsymbol{\Sigma}\mathbf{V}^\intercal\mathbf{w} - \widetilde{\mathbf{w}}). \tag{38}$$

To find the optimal solution, we set the partial derivatives of the Lagrangian with respect to $\widetilde{\mathbf{w}}$, $\mathbf{w}$, $\lambda$, and $\boldsymbol{\mu}$ to zero:

$$\frac{\partial L}{\partial \widetilde{\mathbf{w}}} = \mathbf{z} - \lambda\widetilde{\mathbf{w}} + \boldsymbol{\mu} = 0, \tag{39}$$

$$\frac{\partial L}{\partial \mathbf{w}} = -\mathbf{V}\boldsymbol{\Sigma}\boldsymbol{\mu} = 0, \tag{40}$$

$$\frac{\partial L}{\partial \lambda} = \frac{1}{2}\left(\|\widetilde{\mathbf{w}}\|_2^2 - \frac{mk}{m+k}\right) = 0, \tag{41}$$

$$\frac{\partial L}{\partial \boldsymbol{\mu}} = \boldsymbol{\Sigma}\mathbf{V}^\intercal\mathbf{w} - \widetilde{\mathbf{w}} = 0. \tag{42}$$

From (42), we have $\mathbf{V}\boldsymbol{\Sigma}\boldsymbol{\mu} = 0$. Assuming $\mathbf{V}$ and $\boldsymbol{\Sigma}$ are full rank, this implies $\boldsymbol{\mu} = 0$. Substituting this into (39), we get:

$$\mathbf{z} - \lambda\widetilde{\mathbf{w}} = 0 \implies \widetilde{\mathbf{w}} = \frac{1}{\lambda}\mathbf{z}. \tag{43}$$

Using (43) and the constraint $\|\widetilde{\mathbf{w}}\|_2^2 = \frac{mk}{m+k}$, we can solve for $\lambda$:

$$\|\widetilde{\mathbf{w}}\|_2^2 = \frac{1}{\lambda^2}\|\mathbf{z}\|_2^2 = \frac{mk}{m+k} \implies \lambda = \pm\|\mathbf{z}\|_2\sqrt{\frac{m+k}{mk}} \tag{44}$$

Since we are maximizing $\mathbf{z}^T\widetilde{\mathbf{w}}$, we choose the positive value of $\lambda$. Thus, the optimal $\widetilde{\mathbf{w}}$ is:

$$\widetilde{\mathbf{w}}^* = \frac{\sqrt{\frac{mk}{m+k}}}{\|\mathbf{z}\|_2}\mathbf{z}. \tag{45}$$

Finally, using (45), we can find the optimal $\mathbf{w}$:

$$\boldsymbol{\Sigma}\mathbf{V}^{\mathsf{T}}\mathbf{w}^* = \widetilde{\mathbf{w}}^* \implies \mathbf{w}^* = (\boldsymbol{\Sigma}\mathbf{V}^{\mathsf{T}})^{-1}\widetilde{\mathbf{w}}^* = \frac{\sqrt{\frac{mk}{m+k}}}{\|\mathbf{z}\|_2}(\boldsymbol{\Sigma}\mathbf{V}^{\mathsf{T}})^{-1}\mathbf{z} = \frac{\sqrt{\frac{mk}{m+k}}}{\|\mathbf{z}\|_2}(\boldsymbol{\Sigma}\mathbf{V}^{\mathsf{T}})^{-1}\mathbf{U}_l^{\mathsf{T}}\widetilde{\mathbf{a}}. \tag{46}$$

The optimal $\mathbf{w}$ is then obtained by solving the linear system $\boldsymbol{\Sigma}\mathbf{V}^{\mathsf{T}}\mathbf{w}^* = \widetilde{\mathbf{w}}^*$. Finally, plugging the solution back in to our objective and canceling terms, we arrive at our solution. Note that we can replace $\mathbf{U}$ with $\mathbf{U}_l$ in our original SVD as $\mathbf{U} \in \mathbb{R}^{(m+n)\times l}$, so we can truncate the vectors in the null space of $\mathbf{U}$.

$$\max_{w\in\mathbb{R}^{m+n}} \widetilde{\mathbf{a}}^T\mathbf{U}\boldsymbol{\Sigma}\mathbf{V}^{\mathsf{T}}\mathbf{w}$$

$$= \widetilde{\mathbf{a}}^T\mathbf{U}_l\boldsymbol{\Sigma}\mathbf{V}^{\mathsf{T}}\frac{\sqrt{\frac{mk}{m+k}}}{\|\mathbf{z}\|_2}(\boldsymbol{\Sigma}\mathbf{V}^{\mathsf{T}})^{-1}\mathbf{U}_l^{\mathsf{T}}\widetilde{\mathbf{a}}$$

$$= \frac{\sqrt{\frac{mk}{m+k}}}{\|\mathbf{U}_l^{\mathsf{T}}\widetilde{a}\|_2}\widetilde{\mathbf{a}}^T\mathbf{U}_l\mathbf{U}_l^{\mathsf{T}}\widetilde{\mathbf{a}}$$

$$= \sqrt{\frac{mk}{m+k}}\|\mathbf{U}_l^{\mathsf{T}}\widetilde{a}\|_2$$

$$\square$$

## D  The Multi-group Optimized Retrieval Algorithm

In this section, we discuss the convergence of `MOPR`, the Multi-group Optimized Proportional Retrieval algorithm. As described in Section 4, the `MOPR` is essentially a cutting plane method (also known as the Kelley-Cheney-Goldstein method [78, 79]) applied to the linear function $\mathbf{a}^T\mathbf{s}$; see also [80].

The algorithm begins by identifying the $k$-most similar items to a given query in $\mathcal{D}_R$. Then, an MSE-minimizing oracle is called and returns the statistic (or group-denoting function) $c \in \mathcal{C}$ with the most disproportionate representation for the set of retrieved items. The function $c$ is then added as a constraint to a linear program that outputs a vector $\mathbf{a} \in [0,1]^n$ maximizing average similarity as in (9) subject to a single constraint $\left|\frac{1}{k}\sum i = 1^n a_i c(\boldsymbol{x}_i^r) - \frac{1}{m}\sum_{i=1}^m c(\boldsymbol{x}_i^c)\right| \leq \rho$, where integer constraints are relaxed. The optimal $\mathbf{a}$ is rounded so only the largest $k$ entries are marked as 1, corresponding to a new set of retrieved items. The process repeats with an increasing number of constraints to the linear program and halts when the oracle does not return a function that violates the target MPR constraint $\rho$. In mathematical terms, we start by solving the problem

$$\max_{\mathbf{a}\in\{0,1\}^n} \mathbf{a}^{\mathsf{T}}\mathbf{s} \tag{47}$$

$$\text{subject to} \quad \sum_{i=1}^{k} \mathbf{a}_i = k.$$

Note that the solution to this problem is given by the vector $\mathbf{a}$ that finds the $k$th largest entries of $\mathbf{s}$, i.e., it identifies the $k$-most similar items to a given query in $\mathcal{D}_R$. Then, Proposition 3 provides a way to generate cuts given the vector $\mathbf{a}_k \in \mathbb{R}^n$ found as a solution of the previous iteration. These cuts are given by the function $c \in \mathcal{C}$ with the most disproportionate representation for the set of retrieved items.

At each iteration, we test if the MPR constraint, less or equal to $\rho$, is satisfied. If not, this gives us a new inequality to be incorporated into the linear problem (D) that outputs a vector $\mathbf{a} \in [0,1]^n$ maximizing average similarity as in (9) subject to a single constraint $\left| \frac{1}{k} \sum_{i=1}^{n} a_i c(\boldsymbol{x}_i^r) - \frac{1}{m} \sum_{i=1}^{m} c(\boldsymbol{x}_i^c) \right| \leq \rho$, where integer constraints are relaxed. Therefore, the algorithm consists of iteratively adding a new constraint (cut) of the form

$$G(a) =: \sup_{c \in C} \left| \frac{1}{k} \sum_{i=1}^{n} a_i c(\boldsymbol{x}_i^r) - \frac{1}{m} \sum_{i=1}^{m} c(\boldsymbol{x}_i^c) \right| \tag{48}$$

and solving the problem again until all the inequalities are fulfilled, i.e., until the oracle does not return a function that violates the target MPR constraint $\rho$. At this point, the algorithm halts. It remains to be shown that the description above indeed corresponds to a traditional cutting plane method for convex functions. For the subsequent discussion, we will denote the term inside the supremum in (48) by $g(\mathbf{a}, c)$, i.e.,

$$g(\mathbf{a}, c) = \frac{1}{k} \sum_{i=1}^{n} a_i c(\boldsymbol{x}_i^r) - \frac{1}{m} \sum_{i=1}^{m} c(\boldsymbol{x}_i^c), \quad \text{for } c \in \widetilde{\mathcal{C}}. \tag{49}$$

The next proposition links the oracle via MSE, described in Proposition 3, with the subdifferential of the function $G(\mathbf{a})$.

**Proposition 5.** *Let $\mathcal{C}$ be in infinite set and let $G(\mathbf{a}) = \sup_{c \in \mathcal{C}} |g(\mathbf{a}, c)|$ be the convex restriction of the optimization problem (9) given by the supremum, i.e., $G(\mathbf{a}) \leq \rho, \forall \mathbf{a} \in \mathbb{R}^n$. If $c^* \in \arg\sup_{c \in \mathcal{C}} |g(\mathbf{a}^*, c)|$ and $\mathbf{c}$ denote the function $c^*$ evaluated at the points $x_1, \ldots, x_n$, i.e., $\mathbf{c} = [c_*(x_1), \ldots, c_*(x_m)]$, then $g^* := \text{sign}(g(\mathbf{a}^*, c^*)) \cdot \mathbf{c} \in \partial G(\mathbf{a}^*)$, i.e., $g^*$ is a subgradient of $G$ at a point $\mathbf{a}^*$.*

*Proof.* We start by noting that $G(\mathbf{a})$ is a convex function since it is a supremum of a composition of an affine function with the absolute value. Also, note that $G(\mathbf{a})$ can be written as $G(\mathbf{a}) = \sup_{c \in \mathcal{C}} |g(\mathbf{a}, c)| = \sup_{c \in \mathcal{C}} |\mathbf{c}^{\mathsf{T}} \mathbf{a} + b_c|$, where $b : \mathcal{C} \to \mathbb{R}$. We need to prove that

$$G(\mathbf{a}^*) + g^{*\mathsf{T}}(\mathbf{a} - \mathbf{a}^*) \leq G(\mathbf{a}), \forall \mathbf{a} \in \mathbb{R}^n \tag{50}$$

In fact, starting from the left-hand side leads to

$$|g(\mathbf{a}^*, c^*)| + \text{sign}(g(\mathbf{a}^*, c^*)) \mathbf{c}^{\mathsf{T}}(\mathbf{a} - \mathbf{a}^*)$$
$$= |\mathbf{c}^{\mathsf{T}} \mathbf{a} + b_c| + \frac{\mathbf{c}^{\mathsf{T}} \mathbf{a} + b_c}{|\mathbf{c}^{\mathsf{T}} \mathbf{a} + b_c|} \mathbf{c}^{\mathsf{T}}(\mathbf{a} - \mathbf{a}^*)$$
$$= |\mathbf{c}^{\mathsf{T}} \mathbf{a} + b_c| + \frac{\mathbf{c}^{\mathsf{T}} \mathbf{a} + b_c}{|\mathbf{c}^{\mathsf{T}} \mathbf{a} + b_c|} - \mathbf{c}^{\mathsf{T}} \mathbf{a}^* - b_{c^*} + \mathbf{c}^{\mathsf{T}} \mathbf{a} + b_{c^*}$$
$$= |\mathbf{c}^{\mathsf{T}} \mathbf{a} + b_c| - \frac{|\mathbf{c}^{\mathsf{T}} \mathbf{a} + b_c|^2}{|\mathbf{c}^{\mathsf{T}} \mathbf{a} + b_c|} + \text{sign}(g(\mathbf{a}^*, c^*))(\mathbf{c}^{\mathsf{T}} \mathbf{a} + b_{c^*})$$
$$\leq |\mathbf{c}^{\mathsf{T}} \mathbf{a} + b_{c^*}| \leq G(\mathbf{a}), \forall \mathbf{a} \in \mathbb{R}^n$$

This shows that $\mathbf{c} = [c_*(x_1), \ldots, c_*(x_m)]$ is a subgradient of $G(\mathbf{a})$. $\qquad \square$

The cutting plane strategy described in Algorithm 1 addresses the constraints of the form $G(\mathbf{a}) \leq \rho$ that are not fulfilled. Since (50) holds, the algorithm adds linear approximations (or cutting planes) of the form

$$G(\mathbf{a}^*) + g^{*\mathsf{T}}(\mathbf{a} - \mathbf{a}^*) \leq \rho \tag{51}$$

to the problem at each iteration until the solution satisfies all the inequalities. This reformulation demonstrates that the MOPR algorithm can be viewed as a traditional cutting plane method for minimizing a linear function over the polyhedron $P = \{a_i \in [0,1] \cap \sum_{i=1}^n a_i = k\}$. If, at a certain iteration, the algorithm finds an iterate that satisfies all the inequalities, it terminates since the lower bound provided by the cutting plane method matches the upper bound. In the case where the algorithm continues adding constraints, the accumulation point (guaranteed to exist due to the compactness of the set $P$) will converge to an optimal solution, as proven in [81, Proposition 4.1.2] or the main theorem in [78, Section 2].

However, it is important to note that even when the algorithm terminates in a finite number of steps, the cutting plane method often requires a substantial number of iterations to converge and may exhibit numerical instabilities. Addressing these instabilities or exploring techniques to discard certain cuts for improved convergence, such as partial cutting plane methods, is beyond the scope of this paper. For further information on these topics, refer to [82, 83] and the references therein.

The integer program may not be as scalable as its convex relaxation. If we relax it to $a_i \in [0,1]$, this becomes a convex optimization problem solvable by standard methods (even for very large $\mathbf{C}$) and provides an upper bound to the original program for reasonable values of the MPR violation (and for which Proposition 5 applies). However, the optimal solution of the relaxed problem might not be feasible for the original problem, because it might have fractional values for some $a_i$, while the original problem requires these to be binary. There are certain special cases where the optimal solution of the relaxed problem is guaranteed to be integral (i.e., all $a_i$ are 0 or 1), in which case it is also an optimal solution for the original problem. This is the case, for example, when the constraint matrix is totally unimodular and the right-hand sides of the constraints are integral [84]. Still, we observe from empirical experiments that selecting the top k $a_i$ of the relaxed problem is a computationally efficient alternative to solving the integer program in Equation (9), since results from rounding produce a negligible difference to the upper bound (see Figures 2 and 3).

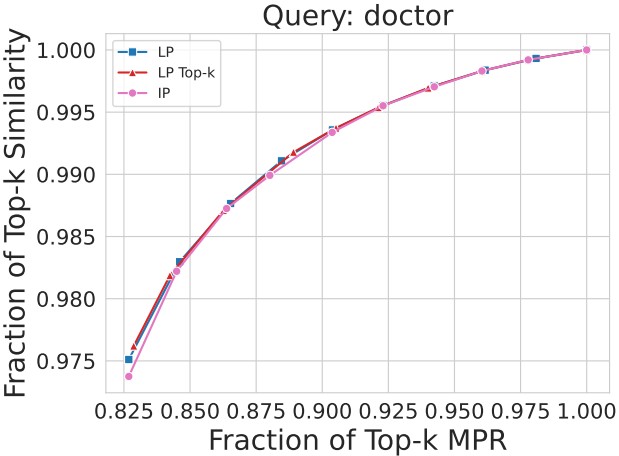

**Figure 2:** Comparison of the linear program with and without taking the top-$k$ to the integer program. Solving the relaxed program is much more computationally efficient and achieves similar performance after rounding to solving the integer program.

### D.1 Multi-group Optimized Retrieval via Quadratic Programming

Proposition 4 allowed us to have a simple closed-form representation of MPR in the case of bounded-norm linear regression functions, i.e., when $\mathcal{C}' \triangleq \{c \in \mathcal{C} \mid \sqrt{\frac{mk}{m+k}} \sum_{i=1}^{m+n} c(x_i)^2 = 1\}$, and $\mathcal{C} = \{x \mapsto w^\mathsf{T} x \mid w \in \mathbb{R}^d\}$. In this case, MOPR becomes

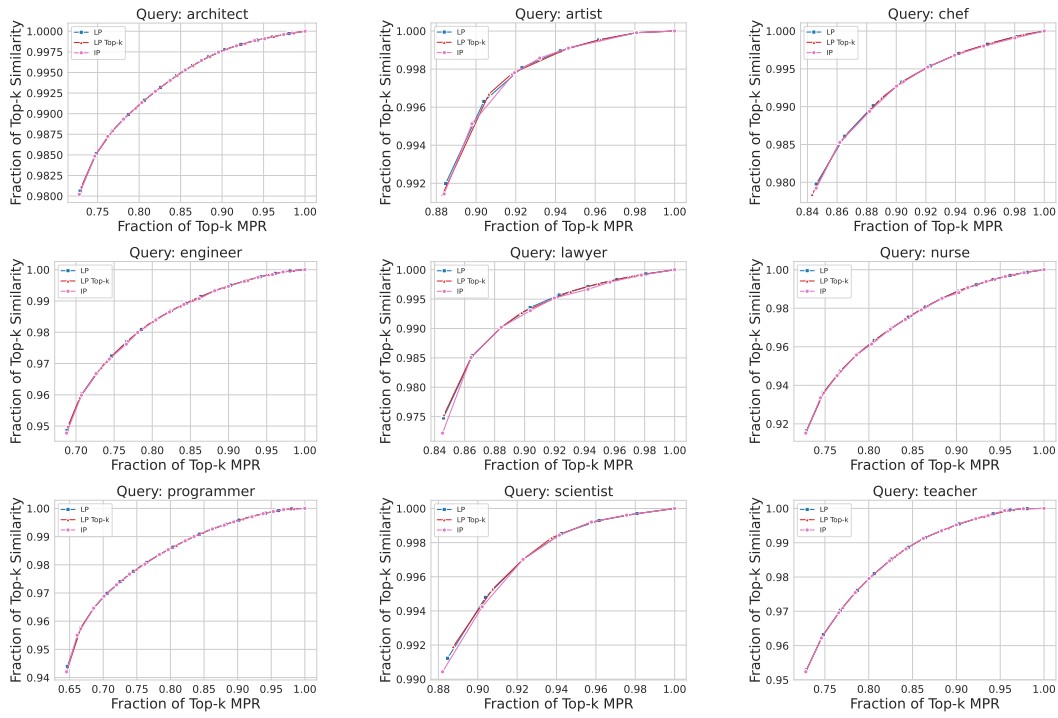

**Figure 3:** Additional comparisons of relaxed problem and top-$k$ selection to integer program. "chef", "nurse", "artist", "lawyer", "teacher", "engineer", "architect", "scientist", and "programmer".

$$\max_{\mathbf{a},\mathbf{y}} \quad \mathbf{a}^\mathsf{T}\mathbf{s} \tag{52}$$

$$\text{subject to} \quad \sqrt{\frac{mk}{m+k}} \, \|\mathbf{U}_l^\mathsf{T}\widetilde{\mathbf{a}}\|_2 \leq \rho,$$

$$\mathbf{a}^\mathsf{T}\mathbf{1} = k,$$

$$a_i \in \{0,1\},$$

where $\widetilde{\mathbf{a}} \in \mathbb{R}^{n+m}$ has $i$-th entry given by $\widetilde{a}_i \triangleq \mathbb{1}_{i \leq n} \frac{a_i}{k} - \mathbb{1}_{i > n} \frac{1}{m}$.

This result allows us to explore the efficiency of the cutting plane algorithm in MOPR by comparing it to the solution to (52), which is a closed-form, convex quadratic program that can be easily optimized with existing solvers. Note that in these experiments, top-$k$ refers to the process of solving the relaxed linear program and then rounding to get a solution in $\{0,1\}^n$ by selecting the $k$ items with the highest score given by $\mathbf{a}$. This is different than the Top-$k$ described elsewhere in the paper, which denotes conducting vanilla retrieval without fairness considerations. In Figures 4 and 6, we compare the quadratic optimization problem (Eqn. 52) to MOPR (a linear program, Algorithm 1) for linear regression and measure the MPR of our solutions with the closed form value from Proposition 4. As we can see, while the quadratic program traces a smoother curve, MOPR's cutting plane approach well-approximates the closed form solution. In these experiments, we retrieve 50 items from CelebA, without a curation set. Finally, in Figures 5 and 7, we run MOPR with the quadratic program oracle and compare our measurement of MPR with the closed form solution and linear regressor-approximated measurement of MPR. In other words, where Figures 4 and 6 measure the quality of our *retrieval algorithm* given a perfect (closed form) MPR oracle, Figures 5 and 7 measure the quality of our MSE estimator of MPR in comparison to the perfect oracle. We observe that the MSE oracle is a perfect approximator of the closed form for MPR, as expected.

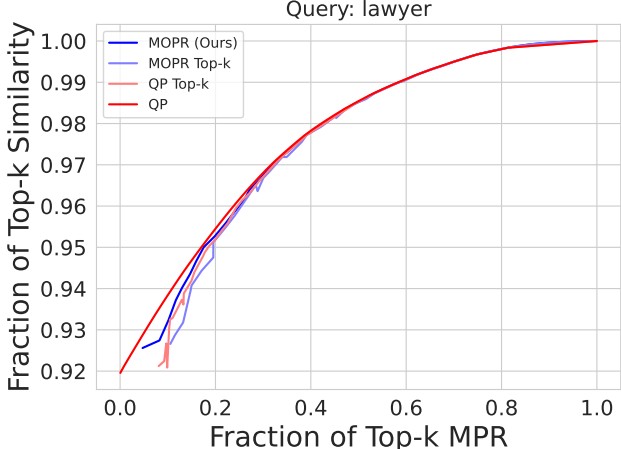

**Figure 4:** Similarity vs MPR for Quadratic Program (Eqn. 52) and `MOPR`. `MOPR` well approximates the quadratic program along the Pareto frontier. Measured over a single query "A photo of a lawyer" for 50 retrieved samples on CelebA.

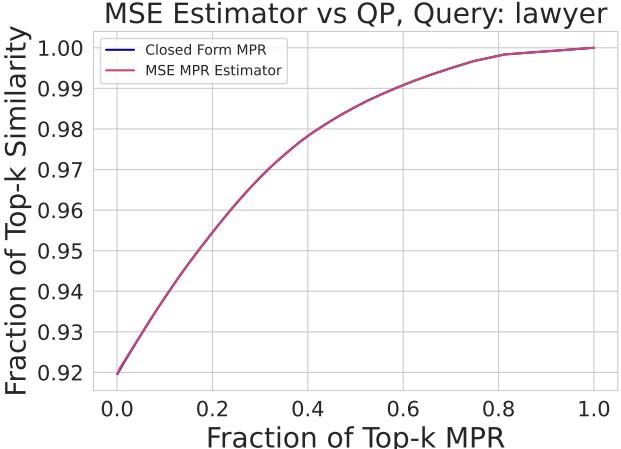

**Figure 5:** Similarity vs MPR for MSE estimated (Prop 3 and closed form (Prop 4) measures of MPR. For the class of linear models, a linear regression oracle perfectly achieves the analytical solution for MPR. Measured over a single query "A photo of a lawyer" for 50 retrieved samples on CelebA.

### D.2 Computing MPR for Functions in an RKHS

Kernel methods are a generalized but practical, non-linear, and easy technique to implement algorithms for a plethora of important problems in machine learning [85, 86]. As MPR is closely related to the maximum mean discrepancy problem, a closed-form expression of MPR can be easily deduced by using results from [60, Lemma 6] when $\mathcal{C}$ is a reproducing kernel Hilbert space, which includes, for example, linear regression, logistic regression, Gaussian kernel ridge regression, and hard-margin SVMs.

**Proposition 6** (MPR for RKHS [60]). *Let* $\{\boldsymbol{x}_i^r\}_{i=1}^k$ *be retrieved samples* $\mathcal{R}$ *and* $\{\boldsymbol{x}_i^c\}_{i=1}^m$ *the samples from the curation dataset* $\mathcal{D}_C$. *Let* $\mathcal{C}$ *be a reproducing kernel hilbert space with kernel* $\mathcal{K}(.,.)$, *then*

$$\mathrm{MPR}(\mathcal{C}, \mathcal{R}, Q) = \left[ \frac{1}{k^2} \sum_{i,j=1}^{k} \mathcal{K}(\boldsymbol{x}_i^r, \boldsymbol{x}_j^r) - \frac{2}{mk} \sum_{i,j=1}^{k,m} \mathcal{K}(\boldsymbol{x}_i^r, \boldsymbol{x}_j^c) + \frac{1}{n^2} \sum_{i,j=1}^{m} \mathcal{K}(\boldsymbol{x}_i^c, \boldsymbol{x}_j^c) \right]^{1/2}. \quad (53)$$

Therefore, given a kernel $\mathcal{K}(.,.)$, we can easily compute $\mathrm{MPR}(\mathcal{C}, \mathcal{R}, Q)$ by evaluating the kernel at the points $\boldsymbol{x}_i^r, \boldsymbol{x}_j^c$ and use this in Algorithm 1.

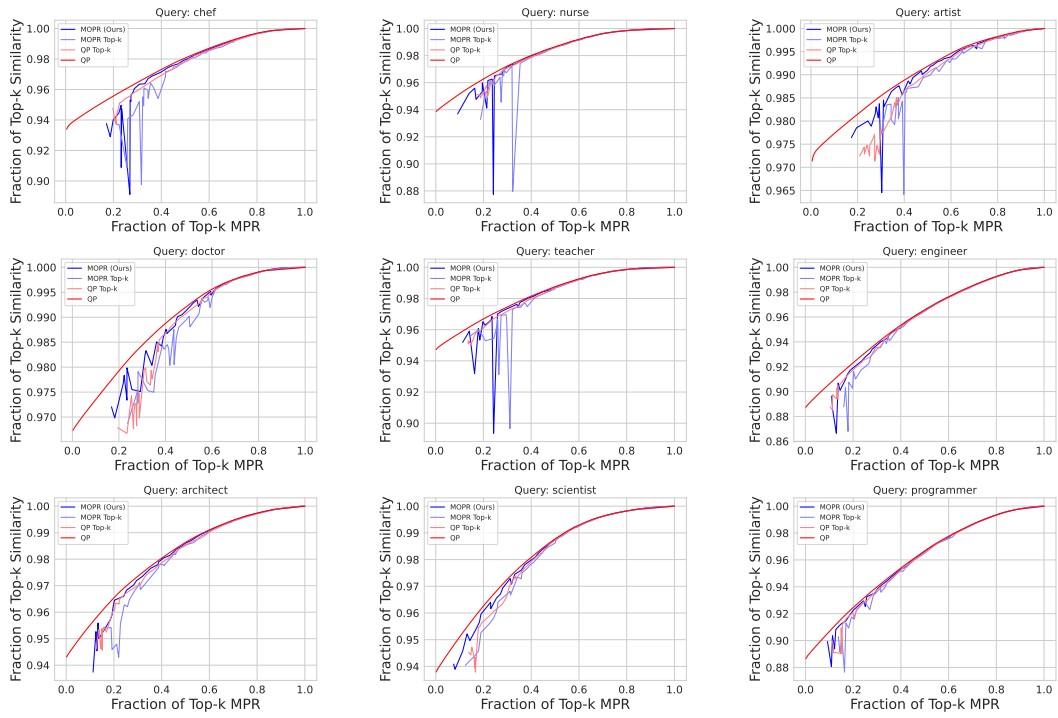

**Figure 6:** Additional comparisons of Quadratic Program (Eqn. 52) and `MOPR` over queries "chef", "nurse", "artist", "doctor", "teacher", "engineer", "architect", "scientist", and "programmer".

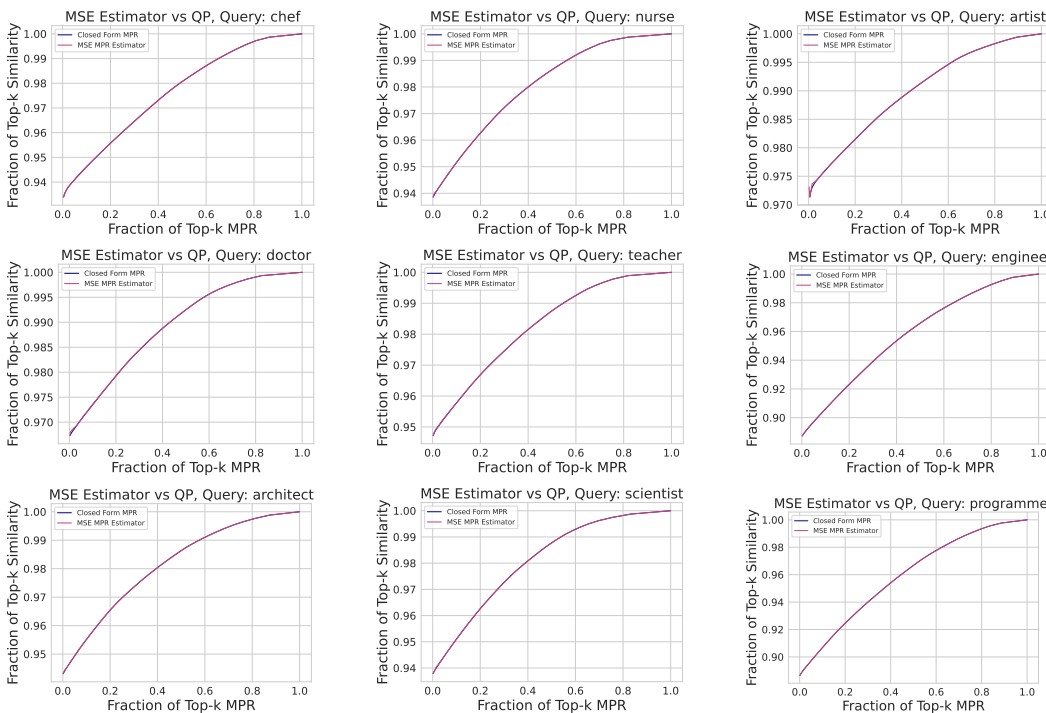

**Figure 7:** Additional comparisons of MSE to closed form MPR estimator for queries "chef", "nurse", "artist", "doctor", "teacher", "engineer", "architect", "scientist", and "programmer".

| | Embedding | Change Embedding | Change Retrieval | Pre-defined features | Approach to Diversity |
|---|---|---|---|---|---|
| **DebiasCLIP** | Custom | ✔ | ✗ | ✔ | Ignore/Decorrelate Attributes |
| **PBM** | Any VLM | ✗ | ✔ | ✔ | Equal Representation over Attributes |
| **PATHS** | Custom | ✔ | ✔ | ✔ | High Variance Representation over Attributes |
| **CLIP-Clip** | Any VLM | ✔ | ✗ | ✔ | Ignore/Decorrelate Attributes |
| MOPR **(ours)** | Any | ✗ | ✔ | ✗ | Multigroup Proportional Representation over Attributes |

**Table 2:** Axis of comparison for ours and existing fair retrieval methods. The blue color indicates a positive feature of the method, while the red color indicates a negative one.

# E    Description of Fair Retrieval Algorithms and Retrieval Statistics

In this section, we describe the methods we benchmark against that aim to promote fairness multimodal models like CLIP [23]. We also briefly overview proportional representation in political sciences.

- CLIP-clip (Wang et al. [18]): CLIP-clip is a post-processing algorithm with two stages. First, the mutual information between each positional feature of an embedding and some sensitive attributes is estimated (treating each sensitive attribute as a discrete random variable and each position as a continuous random variable [87]). Then, in order of greatest to lowest mutual information, at test-time some number of CLIP features are dropped from both the text and image embedding, resulting in less bias (at cost to the fidelity of the embedding). By varying the number of features dropped, a recall-bias trade-off is induced.

- CLIP-debias (Berg et al. [20]): The algorithm learns a series of tokens that are pre-pended to text queries into CLIP via adversarial training to minimize the ability to predict a sensitive attribute of the resulting embedding (in the default case, gender). Then, retrieval is conducted with the debiased embeddings.

- Post-Hoc Bias Mitigation ([29]): Post-Hoc Bias Mitigation is a post-processing method that relies on an oracle classifier to label elements that might be retrieved with either one of several sensitive attributes or a neutral label. Then, iteratively, Post-Hoc Bias Mitigation adds either a set of elements (one of each predicted sensitive attribute) or a predicted neutral element to target equal representation, based on which option has higher average cosine similarity. The oracle may be a pretrained classifier, a zero-shot method using CLIP embeddings, or the true labels.

- PATHS (Srinivasan et al. [27]): PATHS is a new embedding space built upon the pretrained image-text model CoCa and leveraged text-guided projections to extract an embedding that captures complex notions of diversity in people, upon which a diverse retrieval method (such as the MMR method detailed above, or even MOPR) can be run to effectively increase the diversity of the returned samples.

# F    Connection to Social Choice Theory

In this section, we comment on the role of proportional representation in social sciences. As described in the *related works* in Section 1, proportional representation (broadly construed) has a long history in social choice theory and political sciences [88–92] and has been a guiding principle in the design of political systems, aiming to ensure representation reflects underlying population preferences and demographics. The foundations of social choice theory can be traced back to the late 18th century, with the seminal works of Jean-Charles de Borda and the Marquis de Condorcet, who developed fundamental voting methods and paradoxes that continue to influence modern computational approaches. During this same period, early proponents of proportional representation emerged, such as Honoré Gabriel Riqueti, Comte de Mirabeau, who discussed the idea in 1780.

The 19th century saw significant contributions to the development of proportional representation, including the works of Thomas Wright Hill, Thomas Hare, and Charles Dodgson (better known as Lewis Carroll), who made substantial contributions to voting theory and proportional representation systems. Hare's 1859 publication, "A Treatise on the Election of Representatives, Parliamentary and Municipal" [93], outlined a system of single transferable vote for proportional representation. In the late 19th and early 20th centuries, the works of Carl Andrae, Victor D'Hondt, and John Stuart Mill further explored and contributed to the discussion of proportional representation in the political

science literature. The field was revolutionized in the 20th century by Kenneth Arrow's seminal work, whose impossibility theorem fundamentally shaped our understanding of social choice mechanisms. See [88–92].

Similarly, in ML, our aim is proportional representation in data and outputs, ensuring equitable performance across groups and mirroring representation goals in political systems. PR electoral systems [94], used by countries like Uruguay, Sweden, South Africa, and New Zealand, allocate legislative seats proportionally to votes received by parties. Challenges include, e.g., gerrymandering [94] and the complexity of apportionment and seat allocation [95, 96].

The emergence of computational social choice theory [97] in recent decades has built upon these classical foundations while introducing new algorithmic perspectives to address modern challenges. Researchers began studying algorithmic aspects of proportional representation, particularly focusing on the complexity of achieving different notions of proportionality in committee selection [95, 96]. The Hamilton and d'Hondt methods, initially developed for parliamentary seat allocation, have been extended and generalized to handle more complex scenarios. As mentioned at the end of Section 1, a notable contribution in this direction is the work of Lang and Skowron [37], who introduced multi-attribute proportional representation. Their framework generalizes classical apportionment methods to scenarios where candidates possess multiple attributes (such as gender, profession, and age), and representation goals must be satisfied across all these dimensions simultaneously. Their work provides both theoretical guarantees through approximation algorithms and practical methods for achieving proportional representation in modern committee selection problems, bridging the gap between classical political science approaches and contemporary computational challenges. In particular, they introduced the problem of multi-attribute proportional representation in committee selection. Their framework can be viewed as solving an $(m, t)$-filling problem: given $m$ buckets (representing intersectional groups) with target capacities $\{t_1, ..., t_m\}$ (derived from marginal probabilities), the goal is to optimally fill these buckets with appropriately labeled balls (committee members).

While groundbreaking in extending classical apportionment methods to multiple attributes, their approach differs from ours in several key aspects. First, their work considers multiple attributes, but they primarily focus on matching target proportions derived from marginal distributions for each attribute independently rather than truly addressing intersectional representation. That is, their approach aims to satisfy target proportions for each individual attribute (like gender or profession separately) but does not explicitly handle the complex interactions between attributes (like representation of specific gender-profession combinations). This is in contrast to our framework, where the function class $\mathcal{C}$ can directly capture and measure such intersectional relationships. Moreover, their formulation focuses solely on matching target distributions without additional objectives, whereas our work explicitly balances representation with retrieval utility through cosine similarity maximization. Second, while they work with pre-specified target probabilities, we estimate the desired representation from a reference dataset, making our approach more flexible and data-driven. Third, and perhaps most importantly, our MPR metric measures representation through the lens of computationally identifiable groups defined by the aforementioned function class $\mathcal{C}$ rather than explicitly enumerating all possible intersectional groups. Through $\mathcal{C}$, our framework provides a unified way to handle both simple and complex representations: it can capture discrete indicators for each attribute-value pair (like the Hamilton method; see Definition 5 in [37]) but also complex nonlinear relationships (like d'Hondt method; see Definition 6 in [37]), and goes beyond both to measure intersectional representation via decision trees, linear functions, neural networks, or more generally, any function in a reproducing kernel Hilbert space (RKHS). This allows us to handle more complex notions of representation while maintaining computational tractability. In essence, while [37] provides fundamental insights for multi-attribute committee selection, our framework offers a more general approach suitable for modern retrieval systems where both representation and relevance matter.

## G  Additional Results

In this section, we include additional experimental results. First, we include a comparison of runtimes for each method in Table 3. We find that MOPR provides up to a 100x speedup when compared with the most competitive retrieval algorithm, MMR, while remaining competitive to methods such as CLIP-Clip and DebiasCLIP that do not modify the retrieval algorithm at all and just use embedding similarity search. We additionally provide the similarity score for the minimum MPR achieved by each algorithm, which was used to report the results in Tables 1 and 5. These similarity scores can be

found in Table 4. Even when significantly closing the MPR gap and achieving more representation than other methods, MOPR also preserves high cosine similarity with respect to the query.

Second, we include full tables of intersectional retrieval performance on UTKFace for our method, Top-$k$, MMR, CLIP-Clip, PBM, and DebiasCLIP. We report averaged results for each of the 10 queries while retrieving 50 images. In Table 5, we compute MPR with a linear regression oracle and optimize our method with the same linear regression oracle. We are able to perfectly balance across intersectional groups while other methods often fail to account for the 'Others' race category and the 'Female' gender category. Some methods, such as PBM, which optimize for equal representation, are able to achieve balanced representation across the gender axis but fail to consider diversity in race. In Table 6, we see a similar story for MPR measured and optimized with a decision tree.

Next, we include additional similarity-MPR tradeoff curves for the other 9 queries not reported in the main paper (Figs. 8,9,10), as well as for all 10 queries on two additional models: a decision tree (Figs. 11, 12, 13) and with an MLP with 1 hidden layer of dimension 64 (Figs. 14, 15, 16). We observe a similar Pareto-domination to that observed in the main paper, as our method is able to lower MPR the most while preserving similarity even with more nonlinear oracle estimators of MPR. In the MLP setting, while PBM preserves high similarity on the whole, it is still unable to decrease MPR as low as MOPRis able to at ≈ 95% of top-k similarity.

We also include results for an additional 45 queries for the Occupations dataset (with FairFace as the curation dataset), as this dataset comes with 45 ground-truth occupations labels. These results again highlight the Pareto-dominance of MOPR, as shown in Figures 17, 18, and 19.

In Figure 20, we explore the performance of MOPR in terms of precision, a common retrieval metric. We can do this using the Occupations dataset, as it has ground truth occupations labels with which we can benchmark retrieval precision. We see that MOPR performs competitively in terms of precision while also traversing more of the precision-MPR Pareto frontier. In Figure 21, we include nine additional precision-representation curves. Similar to the result for the query "chief executive officer", MOPR maintains similar precision-representation performance to baseline methods.

Finally, we include example sets of retrievals for 'architect' and 'teacher' queries over UTKFace with each of our baselines. Similar to the tables, we select each algorithm's solution with the lowest MPR after doing a hyperparameter sweep for each respective algorithm and plot the corresponding images below. The rest of the Appendix is solely tables and figures.

|  | k=10 | k=50 | k=150 |
|---|---|---|---|
| MOPR(LP, 10 iterations) | $1.434 \pm 0.304$ | $1.6 \pm 0.54$ | $1.371 \pm 0.361$ |
| MOPR(LP, 50 iterations) | $5.14 \pm 0.352$ | $4.199 \pm 1.74$ | $4.342 \pm 1.803$ |
| MOPR(QP, linear regression) | $0.528 \pm 0.061$ | $0.533 \pm 0.066$ | $0.584 \pm 0.089$ |
| MMR | $151.961 \pm 1.126$ | $267.412 \pm 3.863$ | $556.184 \pm 6.324$ |
| PBM | $0.275 \pm 0.162$ | $0.241 \pm 0.013$ | $0.276 \pm 0.151$ |
| CLIP-Clip | $21.303 \pm 0.361$ | $21.907 \pm 0.525$ | $21.34 \pm 0.353$ |
| DebiasCLIP | 2e-4 $\pm$ 1e-6 | 2e-4 $\pm$ 1e-6 | 2e-4 $\pm$ 1e-6 |

**Table 3:** Average running time (in seconds) over ten queries for k=10, 50, and 150 items for the different methods. MOPR is able to achieve 100x speedup over the SOTA retrieval algorithm, MMR, and maintains competitive performance with vanilla KNN-based methods such as CLIP-Clip and DebiasCLIP. For more discussion, see the general comment in the rebuttal.

|  | Normalized Mean Cosine Similarity |
|---|---|
| Top-$k$ | $1.0000 \pm 0.0000$ |
| MOPR | $0.9049 \pm 0.0692$ |
| MMR | $0.8047 \pm 0.1477$ |
| PBM | $0.9416 \pm 0.0802$ |
| CLIP-Clip | $0.7758 \pm 0.1433$ |
| DebiasCLIP | $0.8497 \pm 0.0985$ |

**Table 4:** Normalized mean cosine similarity between the query embedding and retrieved image embeddings for retrieved images from Tables 1 and 5. The normalization is performed by dividing the mean similarity by the mean similarity given by the Top-$k$ nearest neighbors of a query.

**Table 5:** Retrieval averaged over 10 queries with 50 retrievals on UTKFace dataset [40] for `MOPR`, $k$-NN, MMR [27], CLIP-Clip, PBM, and DebiasCLIP with MPR optimized and measured with a linear regression oracle. Entries indicate the average percentage representation in retrieved items. `MOPR` is able to balance representation across classes, whereas other baselines miss intersectional groups (highlighted in red).

| | | White | Black | Asian | Indian | Others | Sum |
|---|---|---|---|---|---|---|---|
| `MOPR` **(Ours)** | Male | $7.8 \pm 3.74$ | $10.2 \pm 2.90$ | $13.0 \pm 2.56$ | $11.8 \pm 2.44$ | $7.2 \pm 3.70$ | $\mathbf{50 \pm 0}$ |
| | Female | $12.2 \pm 3.74$ | $9.8 \pm 2.90$ | $7.0 \pm 2.56$ | $8.2 \pm 2.44$ | $12.8 \pm 3.70$ | $\mathbf{50 \pm 0}$ |
| | Sum | $\mathbf{20 \pm 0}$ | $\mathbf{20 \pm 0}$ | $\mathbf{20 \pm 0}$ | $\mathbf{20 \pm 0}$ | $\mathbf{20 \pm 0}$ | |
| **Top-K** | Male | $21.2 \pm 11.84$ | $13.2 \pm 7.70$ | $10.8 \pm 7.54$ | $21.2 \pm 20.70$ | $1.4 \pm 1.56$ | $\mathbf{67.8 \pm 23.72}$ |
| | Female | $17.2 \pm 21.04$ | $6.6 \pm 6.14$ | $4.0 \pm 2.82$ | $2.8 \pm 2.56$ | $1.6 \pm 1.50$ | $\mathbf{32.2 \pm 23.72}$ |
| | Sum | $\mathbf{38.4 \pm 17.18}$ | $\mathbf{19.8 \pm 12.56}$ | $\mathbf{14.8 \pm 7.38}$ | $\mathbf{24 \pm 19.38}$ | $\mathbf{3.0 \pm 2.04}$ | |
| **MMR** | Male | $28.8 \pm 8.96$ | $11.0 \pm 4.58$ | $10.2 \pm 4.04$ | $10.8 \pm 2.40$ | $3.8 \pm 1.06$ | $\mathbf{64.6 \pm 7.74}$ |
| | Female | $16.2 \pm 9.06$ | $7.6 \pm 3.66$ | $5.0 \pm 3.00$ | $4.0 \pm 1.26$ | $2.6 \pm 1.56$ | $\mathbf{35.4 \pm 7.74}$ |
| | Sum | $\mathbf{45.0 \pm 7.76}$ | $\mathbf{18.6 \pm 5.38}$ | $\mathbf{15.2 \pm 5.82}$ | $\mathbf{14.8 \pm 3.12}$ | $\mathbf{6.4 \pm 1.96}$ | |
| **CLIP-Clip** | Male | $23.2 \pm 9.96$ | $11.0 \pm 5.16$ | $10.8 \pm 5.88$ | $14.2 \pm 8.70$ | $1.8 \pm 2.08$ | $\mathbf{61.0 \pm 17.08}$ |
| | Female | $15.2 \pm 9.44$ | $7.0 \pm 4.32$ | $7.4 \pm 5.30$ | $6.6 \pm 5.52$ | $2.8 \pm 1.84$ | $\mathbf{39.0 \pm 17.08}$ |
| | Sum | $\mathbf{38.4 \pm 11.42}$ | $\mathbf{18.0 \pm 8.20}$ | $\mathbf{18.2 \pm 5.68}$ | $\mathbf{20.8 \pm 9.52}$ | $\mathbf{4.6 \pm 2.2}$ | |
| **PBM** | Male | $19.8 \pm 7.82$ | $13.0 \pm 7.12$ | $9.4 \pm 5.66$ | $15.2 \pm 8.72$ | $1.2 \pm 1.6$ | $\mathbf{58.6 \pm 11.62}$ |
| | Female | $18.6 \pm 12.96$ | $8.4 \pm 5.92$ | $5.6 \pm 3.88$ | $7.2 \pm 5.16$ | $1.6 \pm 1.50$ | $\mathbf{41.4 \pm 11.62}$ |
| | Sum | $\mathbf{38.4 \pm 13.50}$ | $\mathbf{21.4 \pm 11.38}$ | $\mathbf{15.0 \pm 6.70}$ | $\mathbf{22.4 \pm 13.44}$ | $\mathbf{2.8 \pm 2.04}$ | |
| **DebiasCLIP** | Male | $20.4 \pm 7.48$ | $8.2 \pm 3.94$ | $5.4 \pm 2.54$ | $11.0 \pm 3.50$ | $3.8 \pm 2.6$ | $\mathbf{48.8 \pm 5.52}$ |
| | Female | $19.4 \pm 6.98$ | $9.4 \pm 5.14$ | $8.4 \pm 4.28$ | $10.0 \pm 4.64$ | $4.0 \pm 1.78$ | $\mathbf{51.2 \pm 5.52}$ |
| | Sum | $\mathbf{39.8 \pm 9.78}$ | $\mathbf{17.6 \pm 3.66}$ | $\mathbf{13.8 \pm 5.62}$ | $\mathbf{21.0 \pm 6.52}$ | $\mathbf{7.8 \pm 3.16}$ | |

**Table 6:** Retrieval averaged over 10 queries with 50 retrievals on UTKFace dataset [40] for `MOPR`, $k$-NN, MMR [27], PBM, and DebiasCLIP with MPR optimized and measured with a decision tree oracle. Entries indicate the average percentage representation in retrieved items. `MOPR` is able to balance representation across classes, whereas other baselines miss intersectional groups (highlighted in red).

| | | White | Black | Asian | Indian | Others | Sum |
|---|---|---|---|---|---|---|---|
| **Ours** | Male | $11.0 \pm 1.34$ | $10.4 \pm 0.8$ | $10.0 \pm 0$ | $10.2 \pm 0.6$ | $8.8 \pm 0.98$ | $\mathbf{50.4 \pm 1.64}$ |
| | Female | $10.6 \pm 1.28$ | $9.6 \pm 0.80$ | $9.6 \pm 1.20$ | $10.2 \pm 1.08$ | $9.6 \pm 1.2$ | $\mathbf{49.6 \pm 1.64}$ |
| | Sum | $\mathbf{21.6 \pm 1.64}$ | $\mathbf{20 \pm 1.26}$ | $\mathbf{20.4 \pm 1.2}$ | $\mathbf{20.4 \pm 1.2}$ | $\mathbf{18.4 \pm 1.2}$ | |
| **Top-K** | Male | $21.2 \pm 11.84$ | $13.2 \pm 7.70$ | $10.8 \pm 7.54$ | $21.2 \pm 20.70$ | $1.4 \pm 1.56$ | $\mathbf{67.8 \pm 23.72}$ |
| | Female | $17.2 \pm 21.04$ | $6.6 \pm 6.14$ | $4.0 \pm 2.82$ | $2.8 \pm 2.56$ | $1.6 \pm 1.50$ | $\mathbf{32.2 \pm 23.72}$ |
| | Sum | $\mathbf{38.4 \pm 17.18}$ | $\mathbf{19.8 \pm 12.56}$ | $\mathbf{14.8 \pm 7.38}$ | $\mathbf{24 \pm 19.38}$ | $\mathbf{3.0 \pm 2.04}$ | |
| **MMR** | Male | $27.4 \pm 9.13$ | $11.8 \pm 5.02$ | $10.2 \pm 4.33$ | $10.2 \pm 3.03$ | $3.2 \pm 2.4$ | $\mathbf{62.8 \pm 11.77}$ |
| | Female | $17 \pm 9.13$ | $9 \pm 5.67$ | $5.4 \pm 3.47$ | $3.6 \pm 1.5$ | $2.2 \pm 1.08$ | $\mathbf{37.2 \pm 11.77}$ |
| | Sum | $\mathbf{44.4 \pm 6.62}$ | $\mathbf{20.8 \pm 9.39}$ | $\mathbf{15.6 \pm 5.64}$ | $\mathbf{13.8 \pm 3.84}$ | $\mathbf{5.4 \pm 2.37}$ | |
| **PBM** | Male | $19.6 \pm 13.11$ | $12.4 \pm 6.56$ | $9.4 \pm 6$ | $15 \pm 8.91$ | $1.2 \pm 1.6$ | $\mathbf{57.6 \pm 12.74}$ |
| | Female | $19.6 \pm 13.11$ | $8.4 \pm 5.92$ | $5.4 \pm 3.69$ | $7.2 \pm 5.15$ | $1.8 \pm 1.66$ | $\mathbf{42.4 \pm 12.74}$ |
| | Sum | $\mathbf{39.2 \pm 13.45}$ | $\mathbf{20.8 \pm 11.07}$ | $\mathbf{14.8 \pm 6.46}$ | $\mathbf{22.2 \pm 13.61}$ | $\mathbf{3 \pm 2.05}$ | |
| **DebiasCLIP** | Male | $20.4 \pm 6.99$ | $8.2 \pm 3.94$ | $5.4 \pm 2.54$ | $11 \pm 3.49$ | $3.8 \pm 2.6$ | $\mathbf{48.8 \pm 5.53}$ |
| | Female | $19.4 \pm 6.99$ | $9.4 \pm 5.14$ | $8.4 \pm 4.27$ | $10 \pm 4.65$ | $4 \pm 1.79$ | $\mathbf{51.2 \pm 5.53}$ |
| | Sum | $\mathbf{39.8 \pm 9.78}$ | $\mathbf{17.6 \pm 3.67}$ | $\mathbf{13.8 \pm 5.62}$ | $\mathbf{21 \pm 6.53}$ | $\mathbf{7.8 \pm 3.16}$ | |

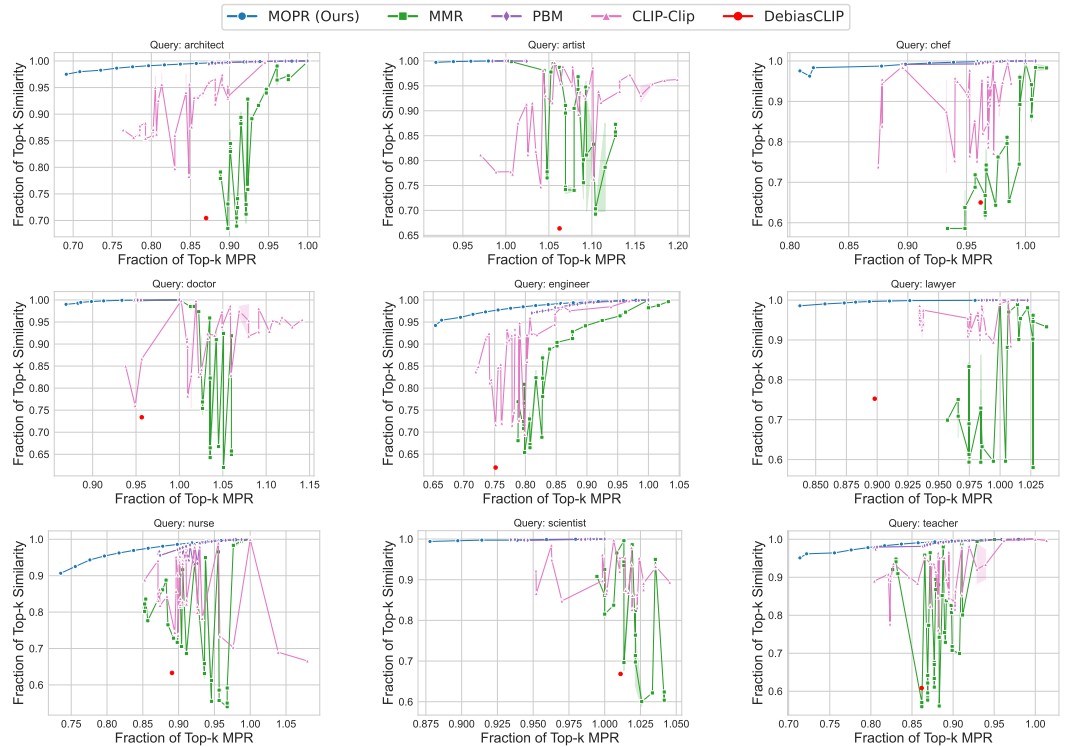

**Figure 8:** Fraction of Top-$k$ cosine similarity vs Fraction of Top-$k$ MPR over linear regression models for 9 additional queries for the CelebA dataset. Values are normalized so Top-$k$ achieves point (1,1) in each case. MOPR Pareto-dominates baselines and significantly closes the MPR gap.

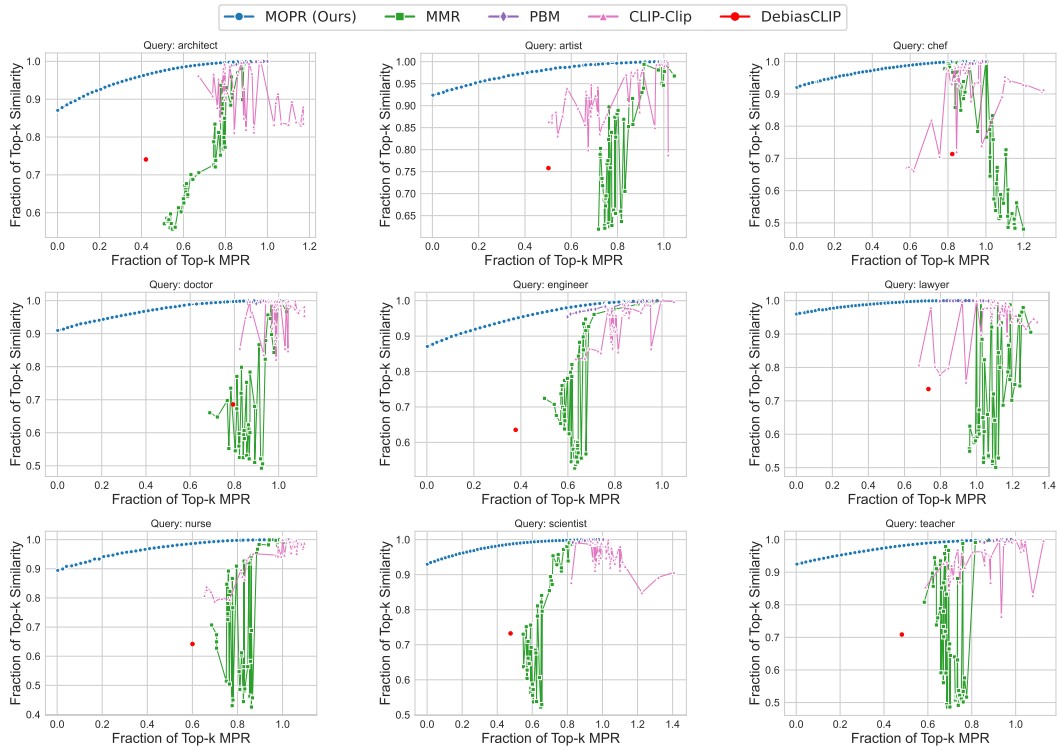

**Figure 9:** Fraction of Top-$k$ cosine similarity vs Fraction of Top-$k$ MPR over linear regression models for 9 additional queries for the UTKFace dataset. Values are normalized so Top-$k$ achieves point (1,1) in each case. `MOPR` Pareto-dominates baselines and significantly closes the MPR gap.

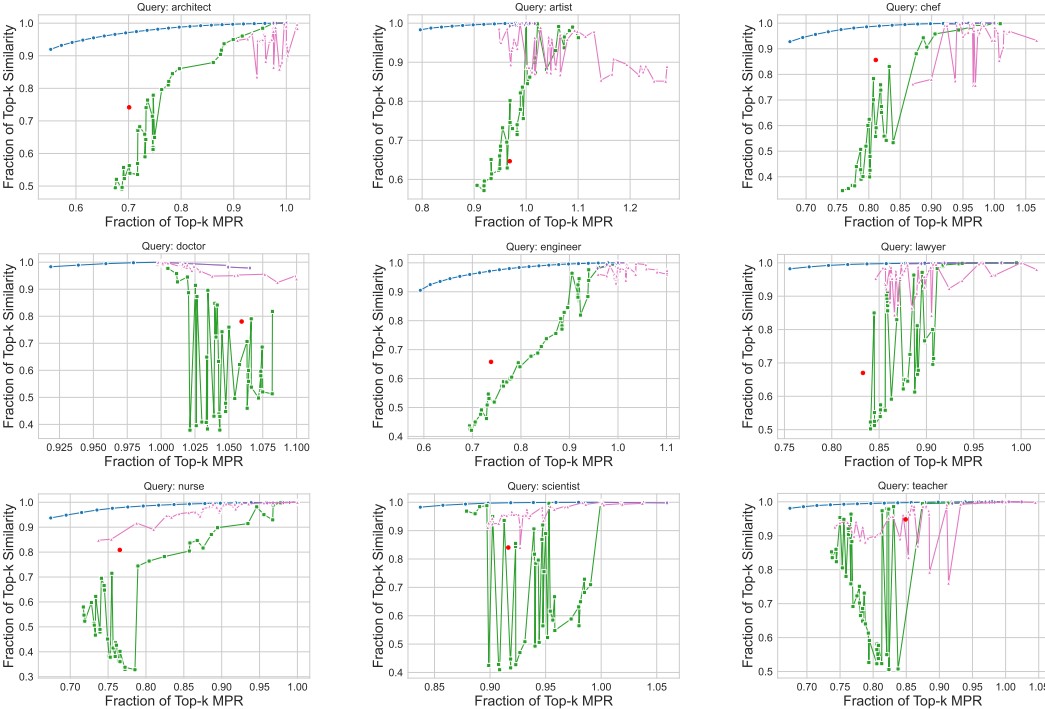

**Figure 10:** Fraction of Top-$k$ cosine similarity vs Fraction of Top-$k$ MPR over linear regression models for 9 additional queries for the Occupations dataset. Values are normalized so Top-$k$ achieves point (1,1) in each case. `MOPR` Pareto-dominates baselines and significantly closes the MPR gap.

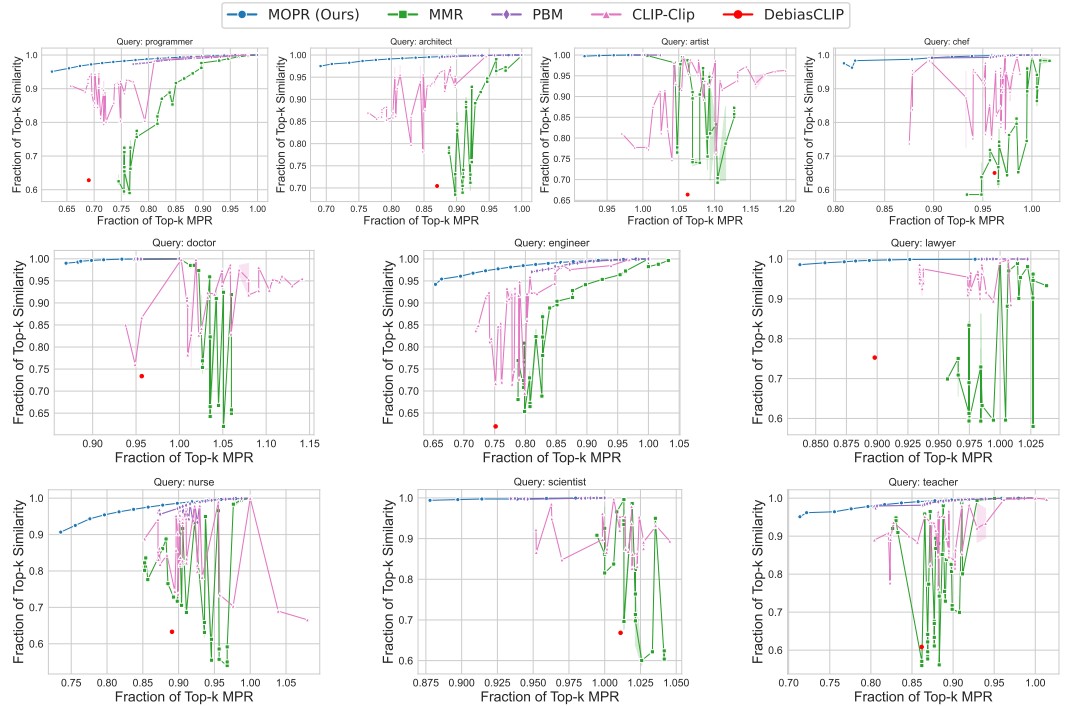

**Figure 11:** Fraction of Top-$k$ cosine similarity vs. Fraction of Top-$k$ MPR over decision trees for 10 queries for the CelebA dataset. Values are normalized so Top-$k$ achieves point (1,1) in each case. MOPR Pareto-dominates baselines and significantly closes the MPR gap.

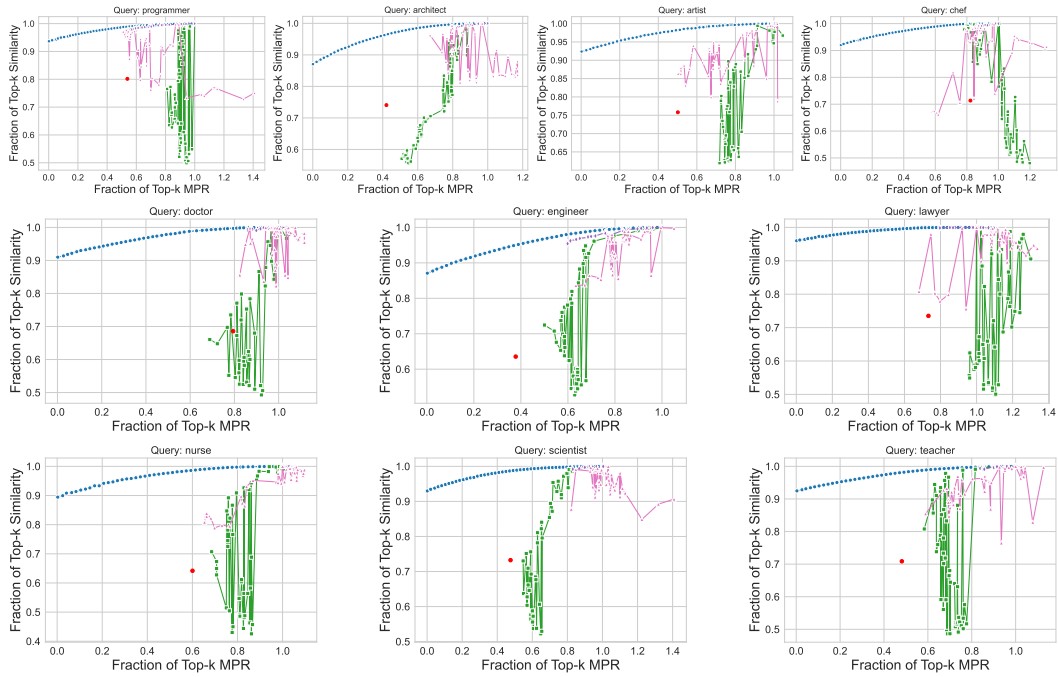

**Figure 12:** Fraction of Top-$k$ cosine similarity vs Fraction of Top-$k$ MPR over decision trees for 10 queries for the UTKFace dataset. Values are normalized so Top-$k$ achieves point (1,1) in each case. MOPR Pareto-dominates baselines and significantly closes the MPR gap.

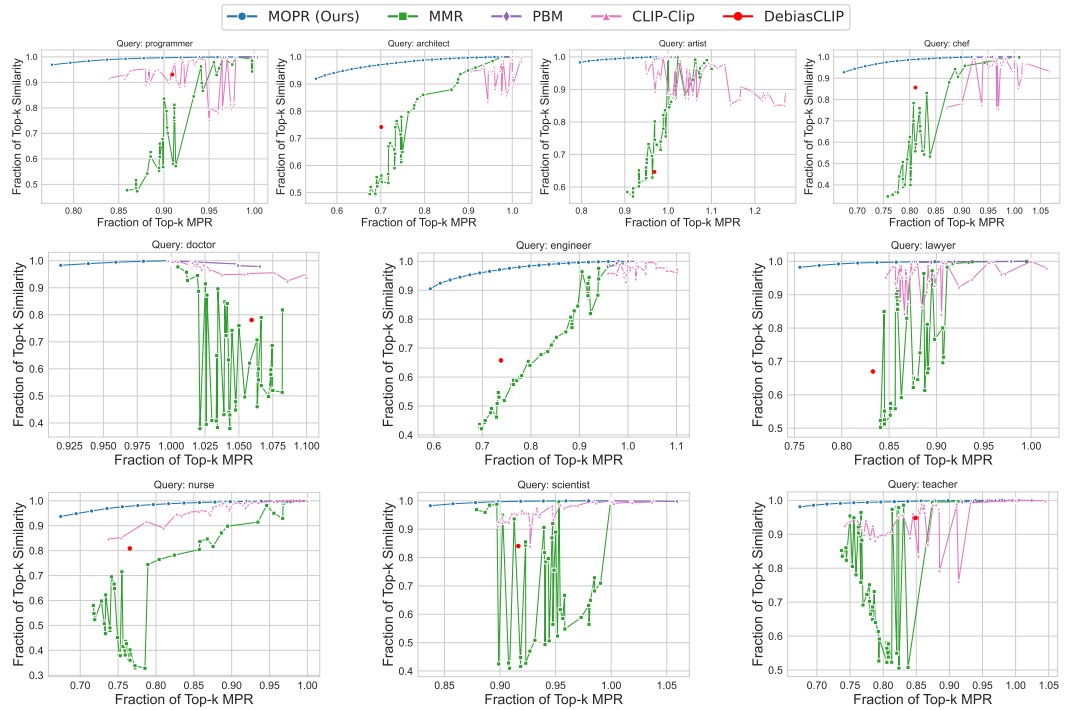

**Figure 13:** Fraction of Top-$k$ cosine similarity vs. Fraction of Top-$k$ MPR over decision trees for 10 queries for the Occupations dataset. Values are normalized so Top-$k$ achieves point (1,1) in each case. `MOPR` Pareto-dominates baselines and significantly closes the MPR gap.

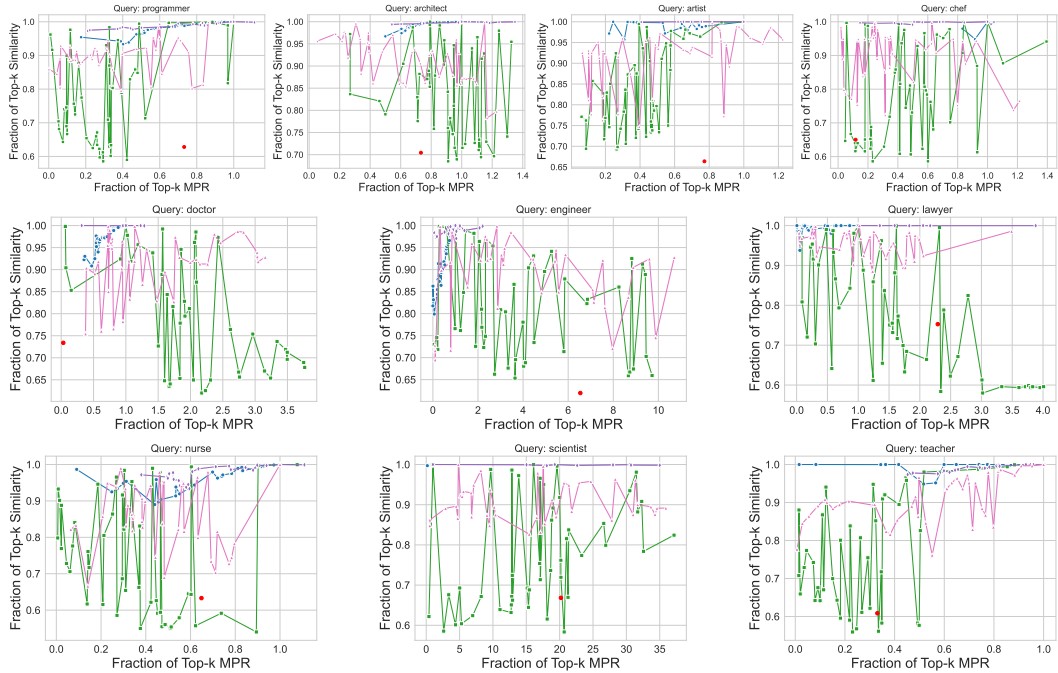

**Figure 14:** Fraction of Top-$k$ cosine similarity vs Fraction of Top-$k$ MPR over 2-layer MLPs (hidden dimension 64) for 10 queries for the CelebA dataset. Values are normalized so Top-$k$ achieves point (1,1) in each case. `MOPR` Pareto-dominates baselines and significantly closes the MPR gap.

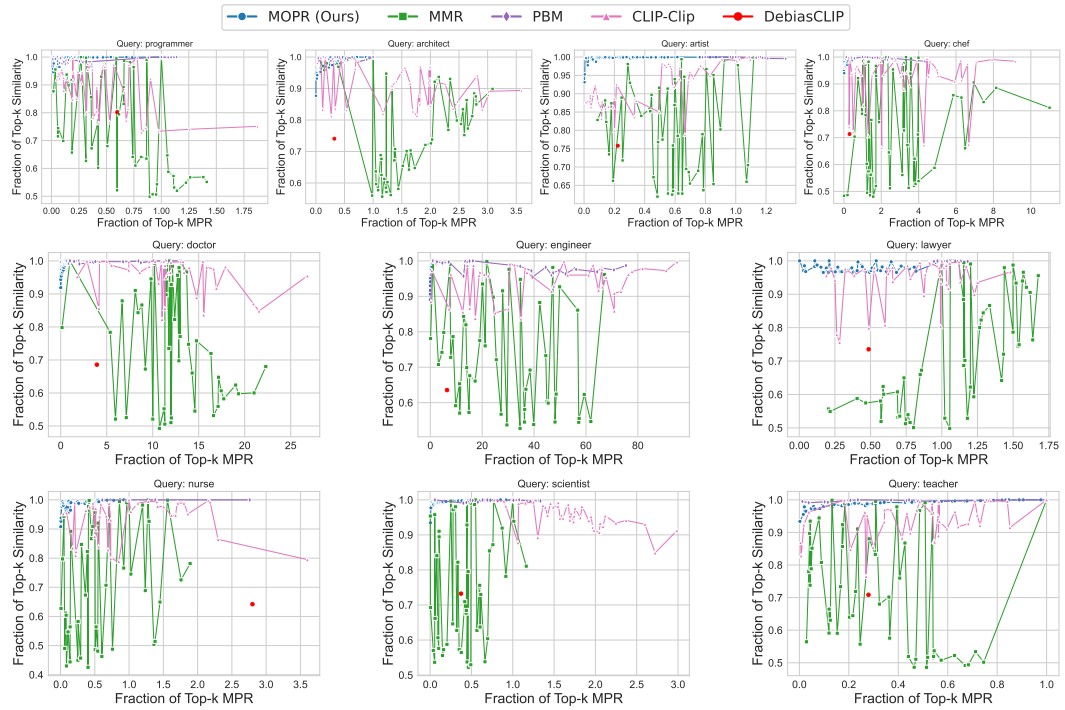

**Figure 15:** Fraction of Top-$k$ cosine similarity vs Fraction of Top-$k$ MPR over 2-layer MLPs (hidden dimension 64) for 10 queries for the UTKFace dataset. Values are normalized so Top-$k$ achieves point (1,1) in each case. `MOPR` Pareto-dominates baselines and significantly closes the MPR gap.

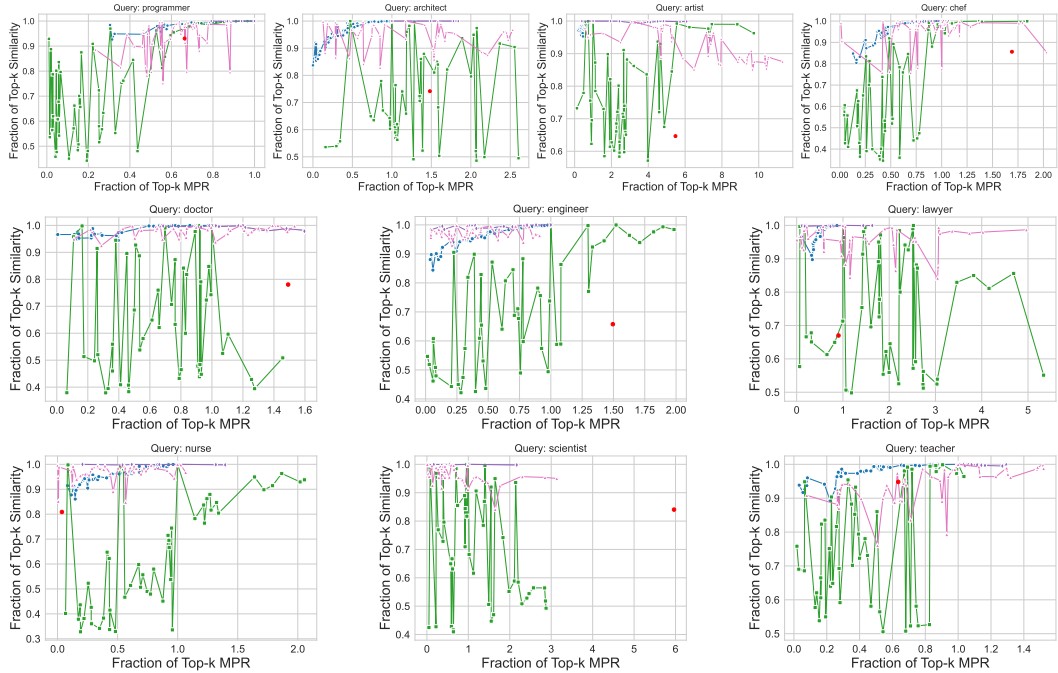

**Figure 16:** Fraction of Top-$k$ cosine similarity vs Fraction of Top-$k$ MPR over 2-layer MLPs (hidden dimension 64) for 10 queries for the Occupations dataset. Values are normalized so Top-$k$ achieves point (1,1) in each case. `MOPR` Pareto-dominates baselines and significantly closes the MPR gap.

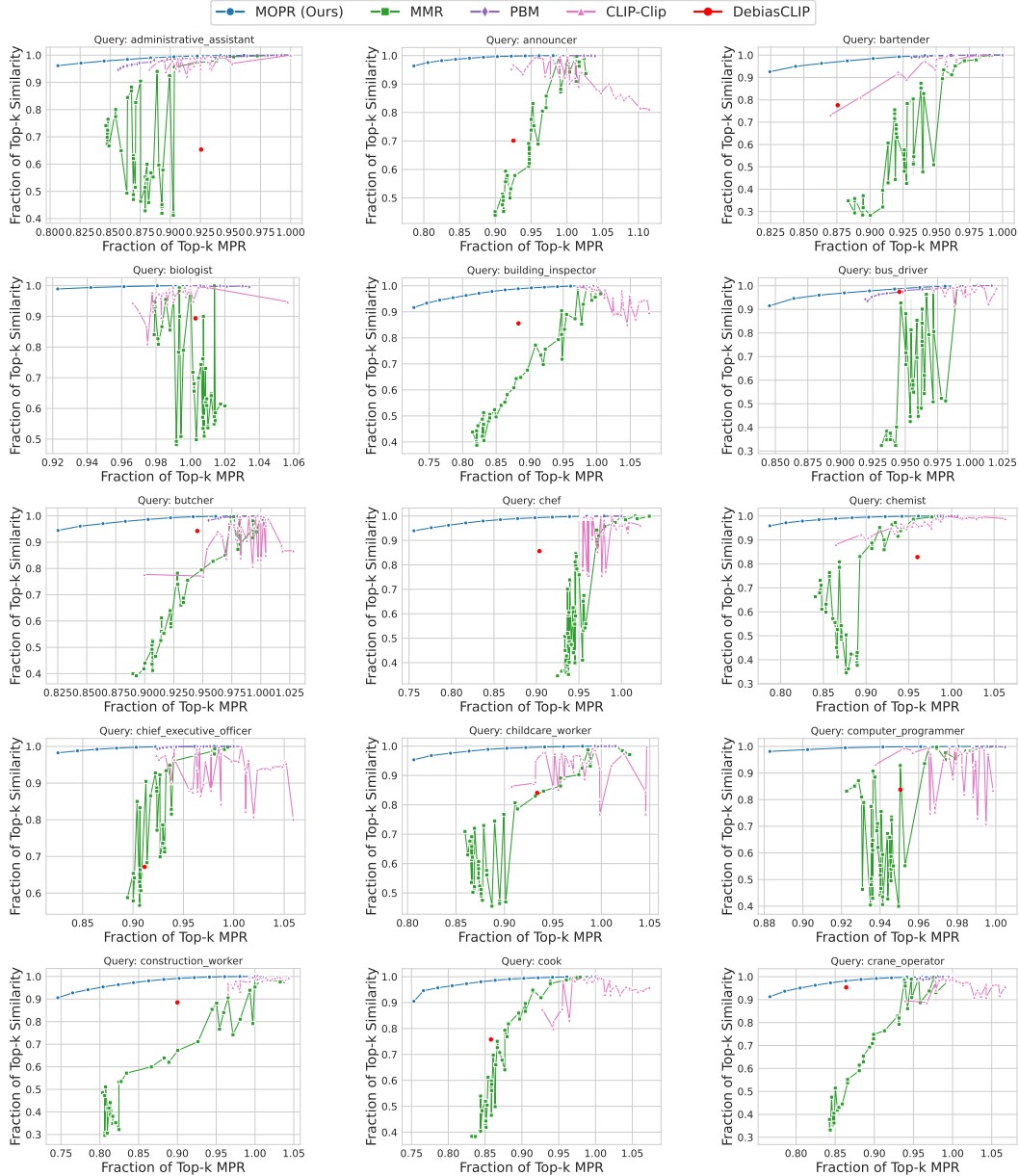

**Figure 17:** Fraction of Top-$k$ cosine similarity vs Fraction of Top-$k$ MPR over linear regression models for 45 (First 15, see Figures 18 and 19) occupations present in the Occupations dataset. Values are normalized so Top-$k$ achieves point (1,1) in each case. MOPR Pareto-dominates baselines and significantly closes the MPR gap.

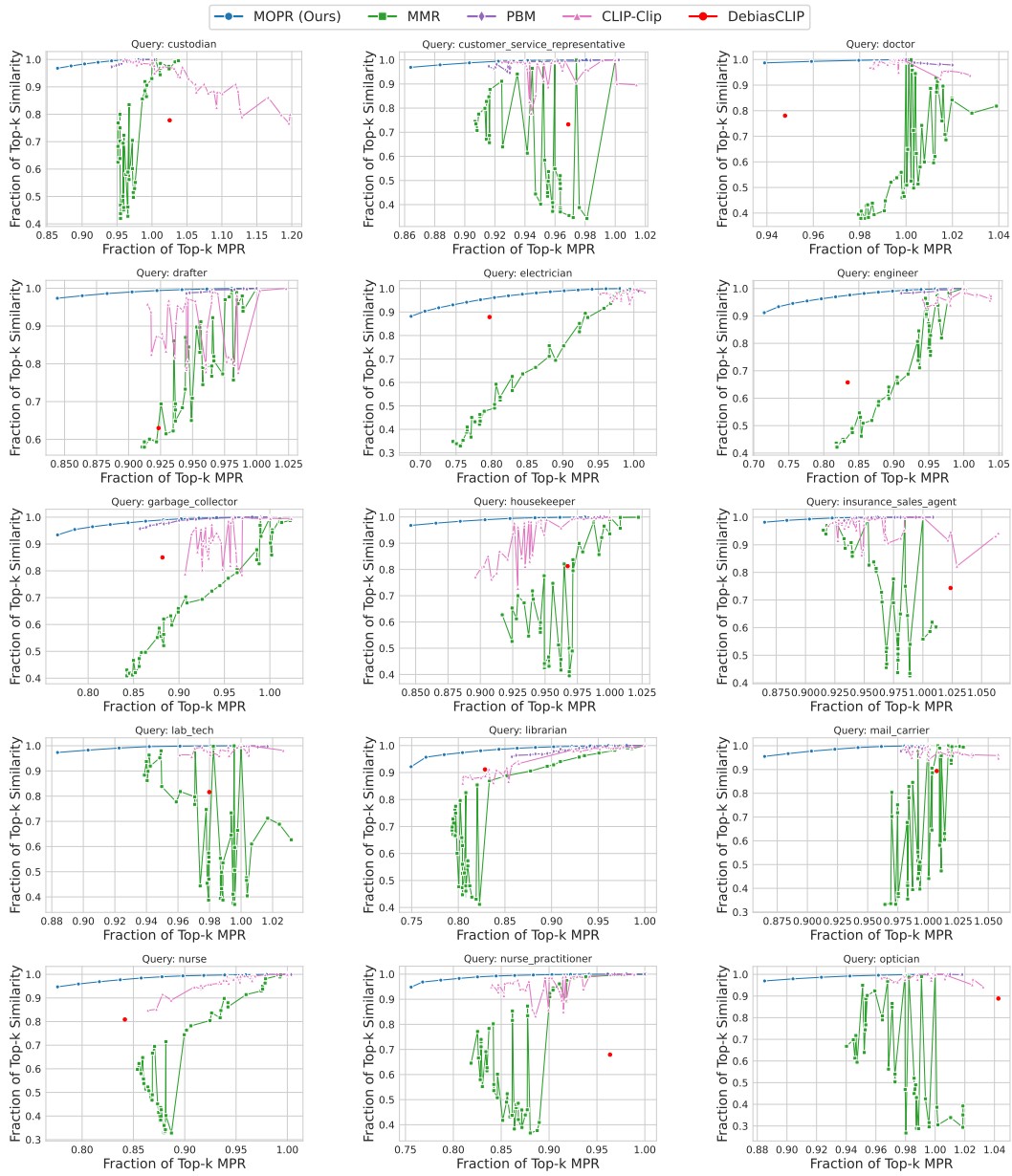

**Figure 18:** Fraction of Top-$k$ cosine similarity vs Fraction of Top-$k$ MPR over linear regression models for 45 (Second 15, see Figures 17 and 19) occupations present in the Occupations dataset. Values are normalized so Top-$k$ achieves point (1,1) in each case. `MOPR` Pareto-dominates baselines and significantly closes the MPR gap.

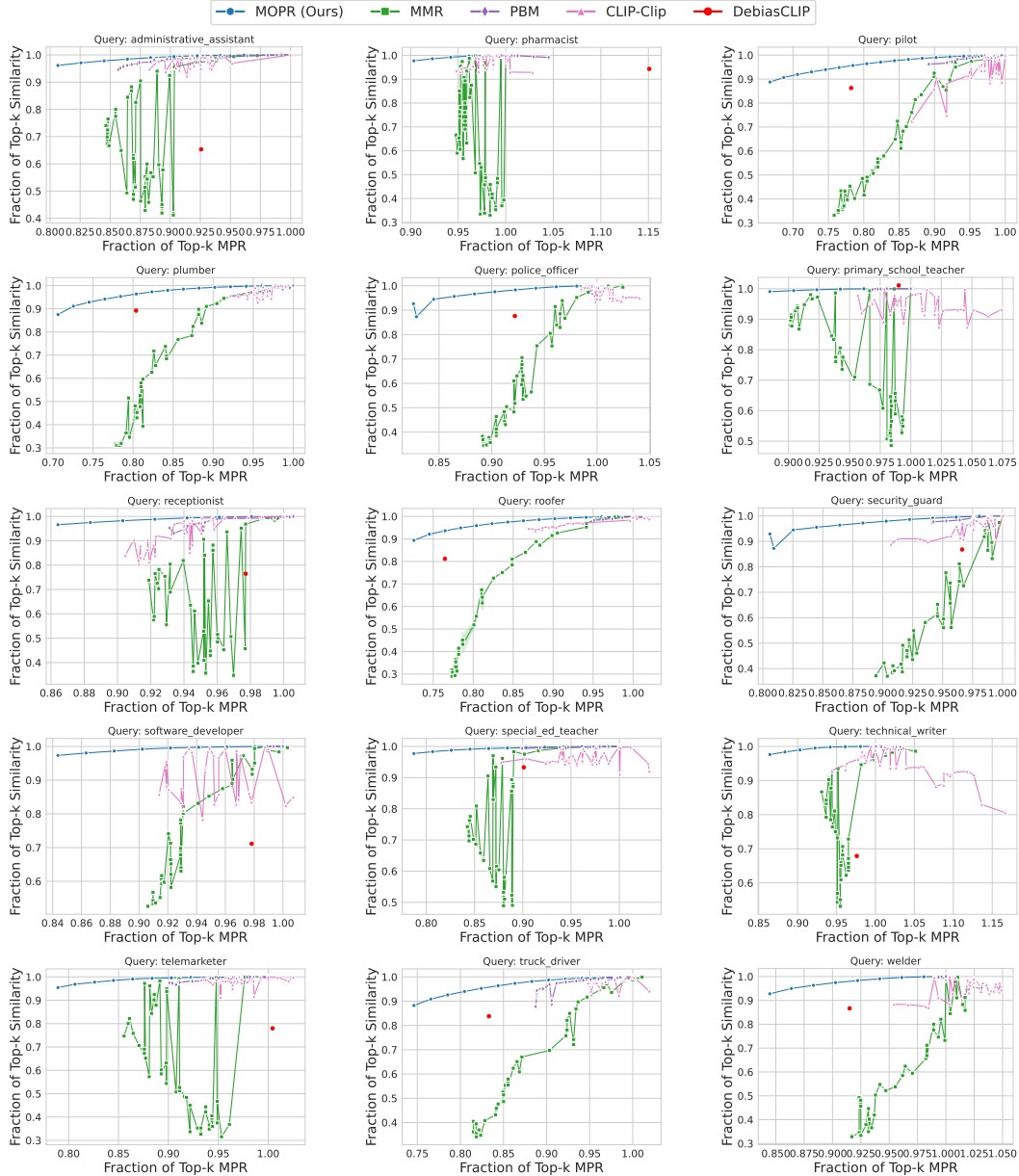

**Figure 19:** Fraction of Top-$k$ cosine similarity vs Fraction of Top-$k$ MPR over linear regression models for 45 (Third 15, see Figures 17 and 18) occupations present in the Occupations dataset. Values are normalized so Top-$k$ achieves point (1,1) in each case. `MOPR` Pareto-dominates baselines and significantly closes the MPR gap.

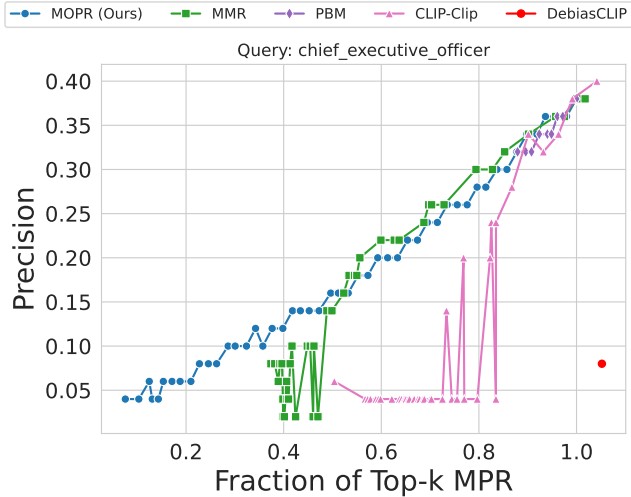

**Figure 20:** Precision-MPR curve on Occupations dataset over linear regression. Retrieving 50 items for the query "A photo of a chief executive officer." MPR remains competitive in terms of retrieval performance.

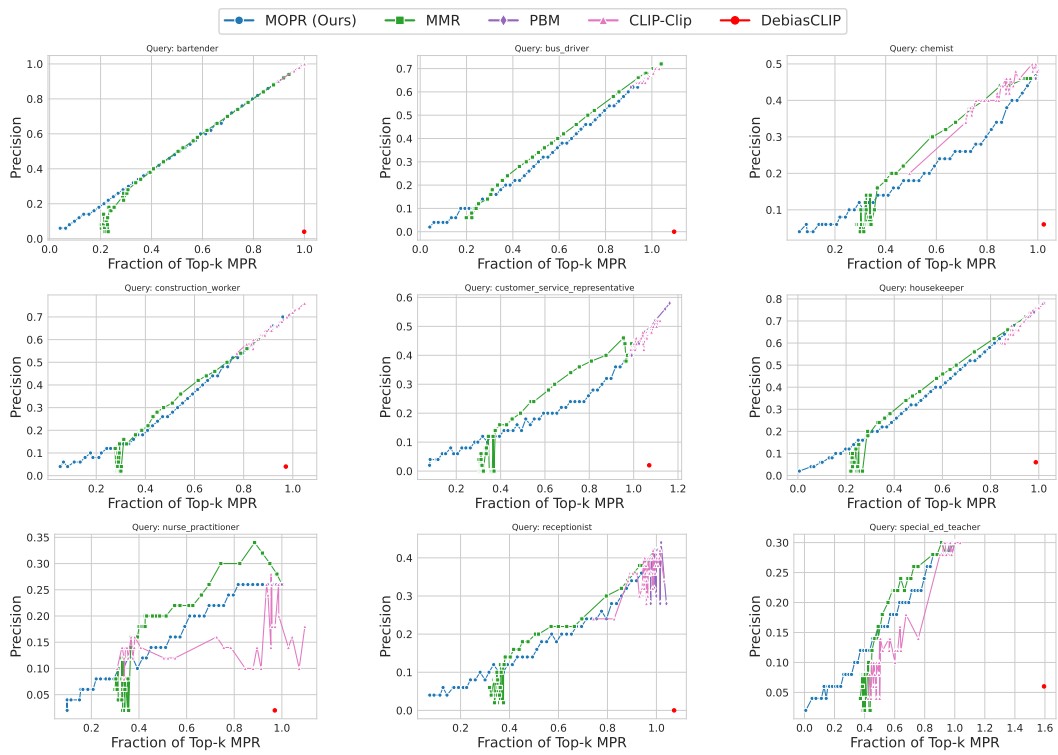

**Figure 21:** Additional precision-representation plots over queries "bartender", "bus driver", "chemist", "construction worker", "customer service representative", "housekeeper", "nurse practitioner", "receptionist", and "sepecial ed teacher".

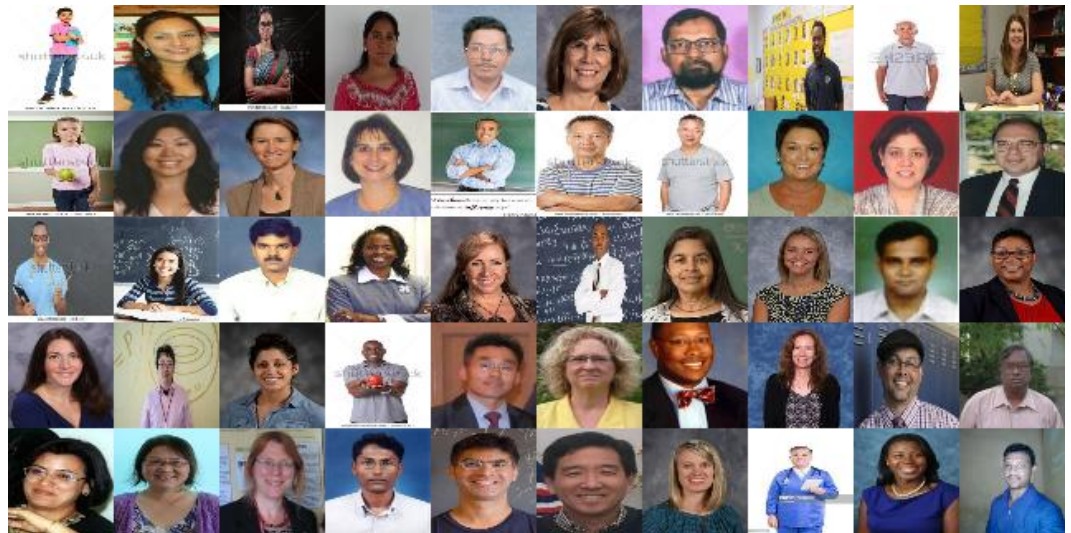

**Figure 22:** Retrievals for MOPR with query "A photo of a teacher"

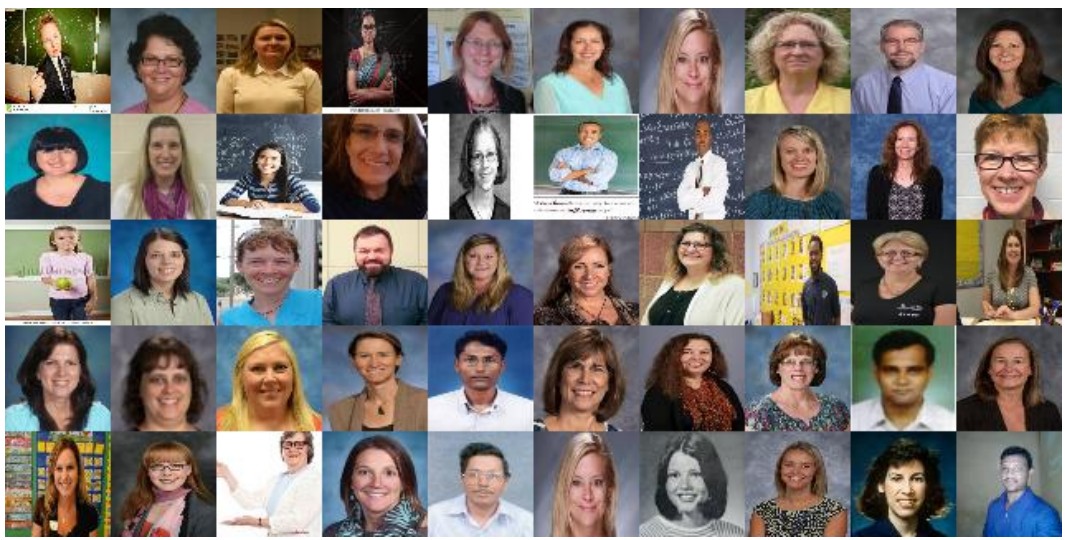

**Figure 23:** Retrievals for $k$-NN with query "A photo of a teacher"

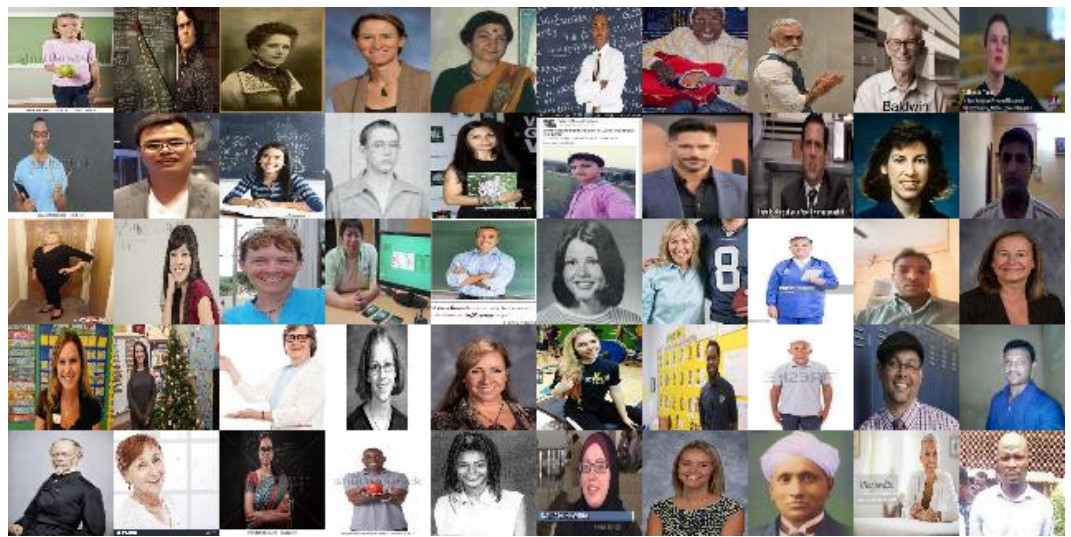

**Figure 24:** Retrievals for MMR with query "A photo of a teacher"

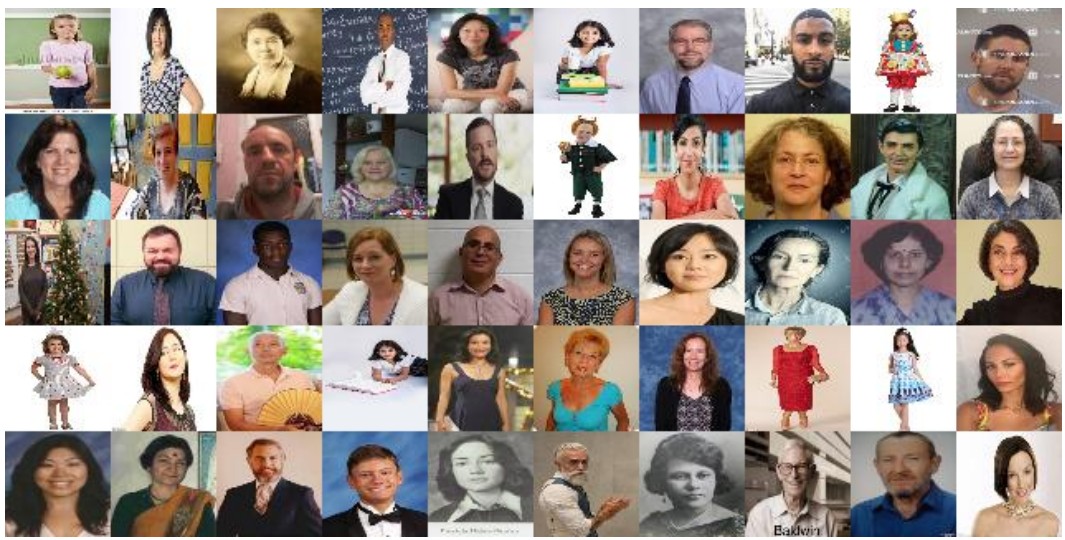

**Figure 25:** Retrievals for CLIP-Clip with query "A photo of a teacher"

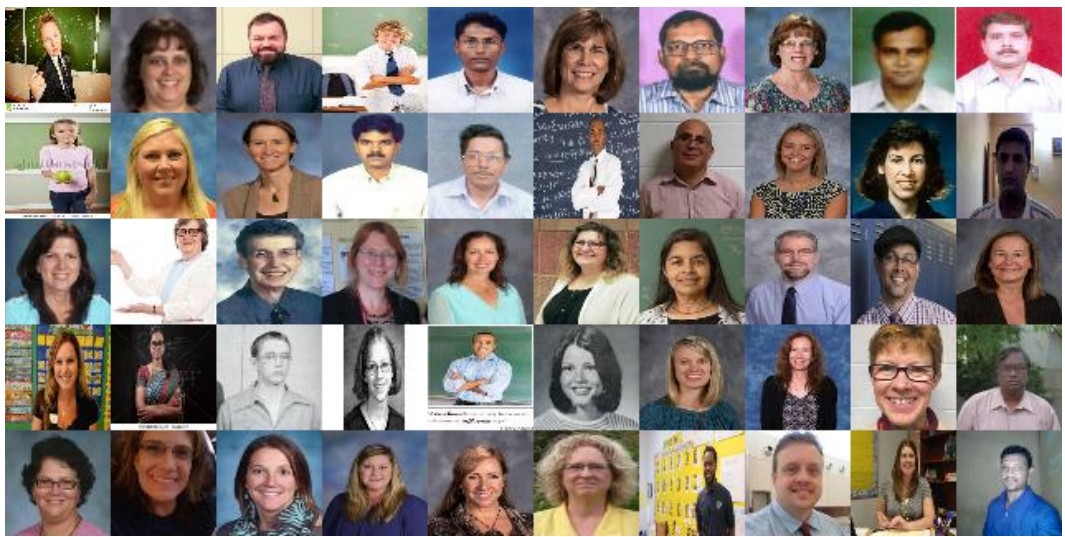

**Figure 26:** Retrievals for PBM with query "A photo of a teacher"

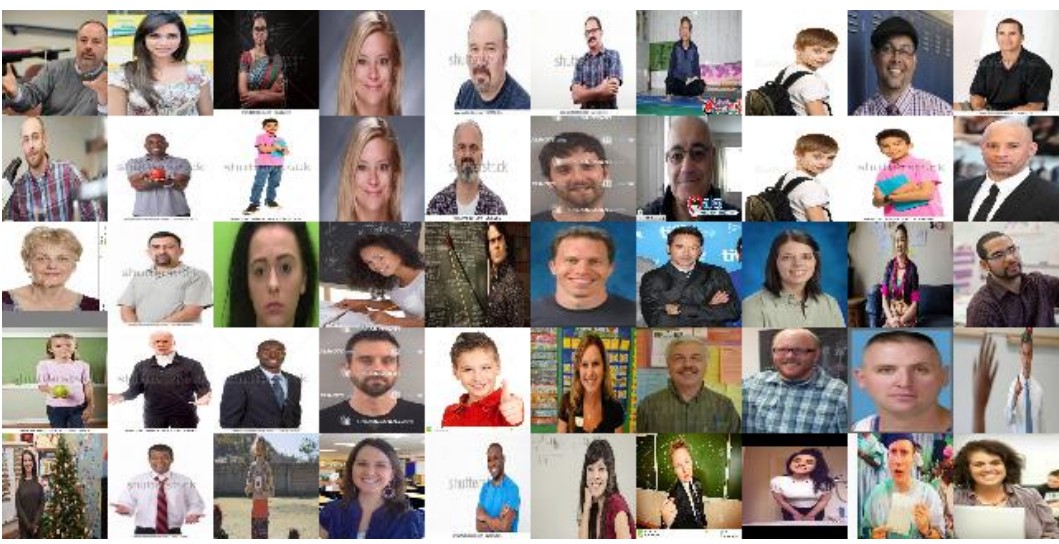

**Figure 27:** Retrievals for DebiasCLIP with query "A photo of a teacher"

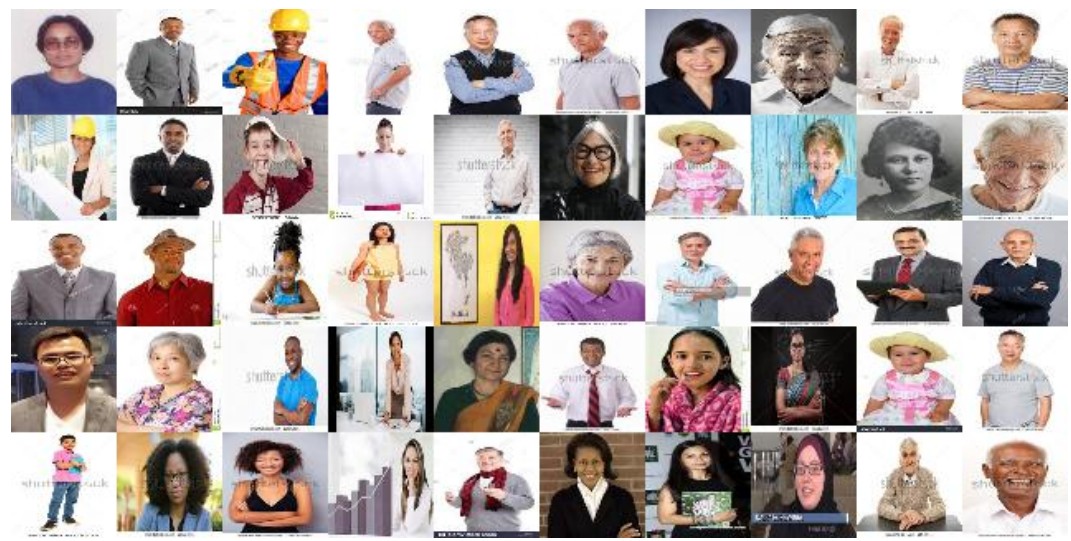

**Figure 28:** Retrievals for MOPR with query "A photo of an architect"

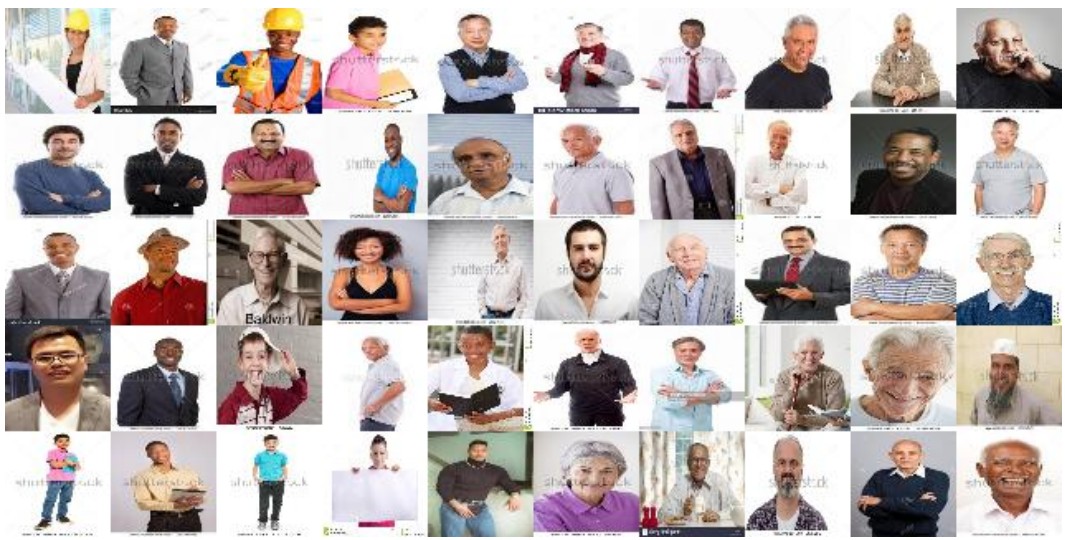

**Figure 29:** Retrievals for $k$-NN with query "A photo of an architect"

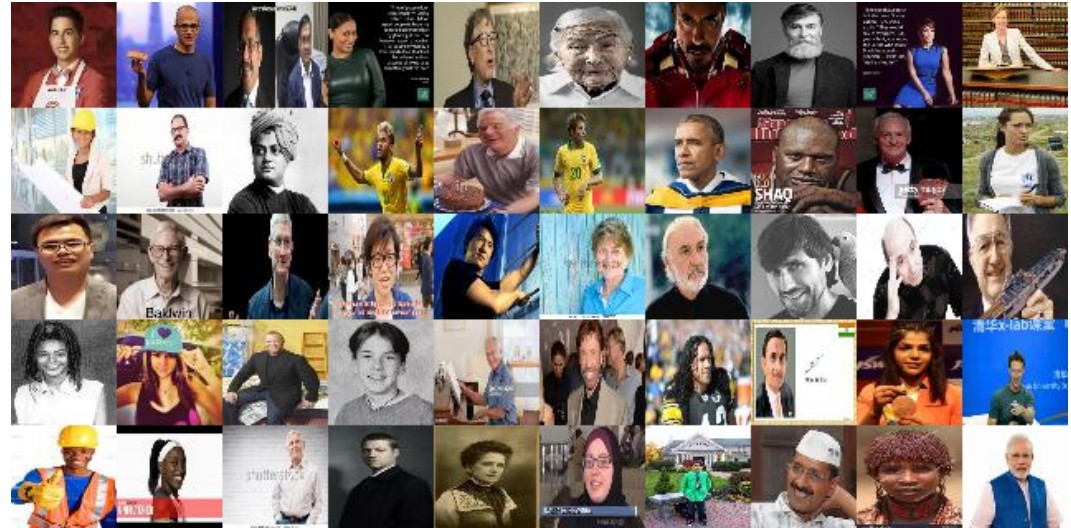

**Figure 30:** Retrievals for MMR with query "A photo of an architect"

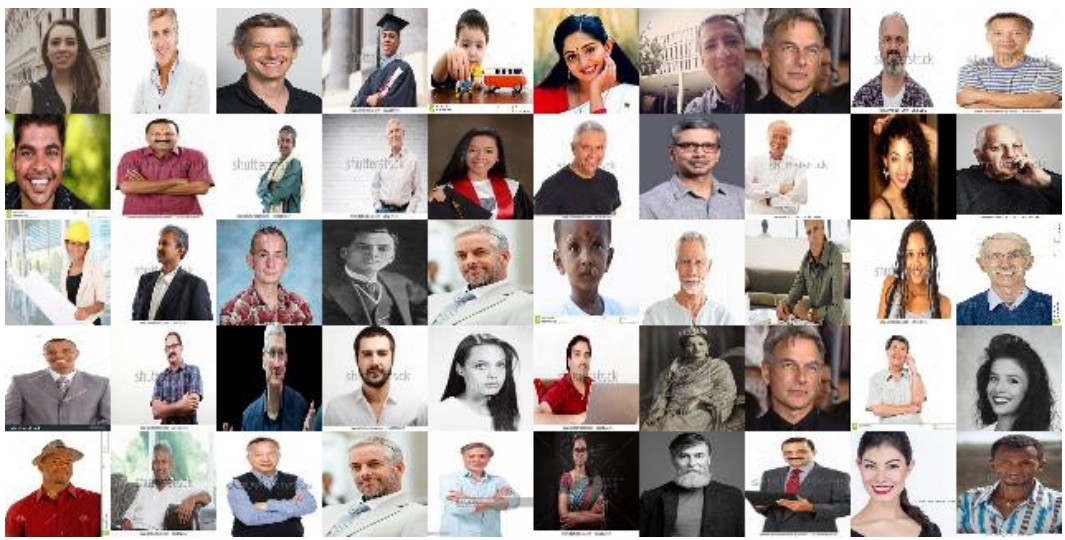

**Figure 31:** Retrievals for CLIP-Clip with query "A photo of an architect"

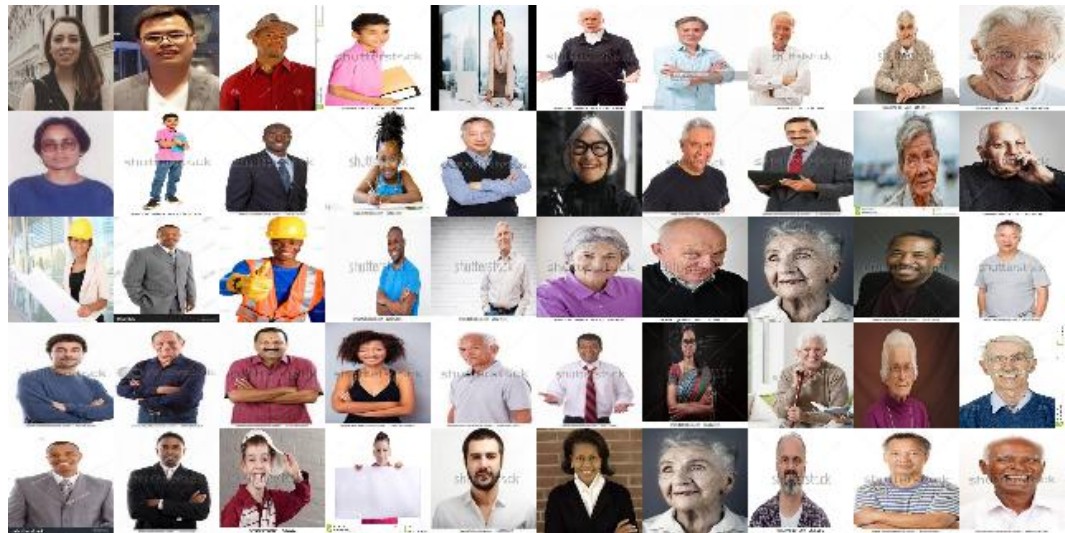

**Figure 32:** Retrievals for PBM with query "A photo of an architect"

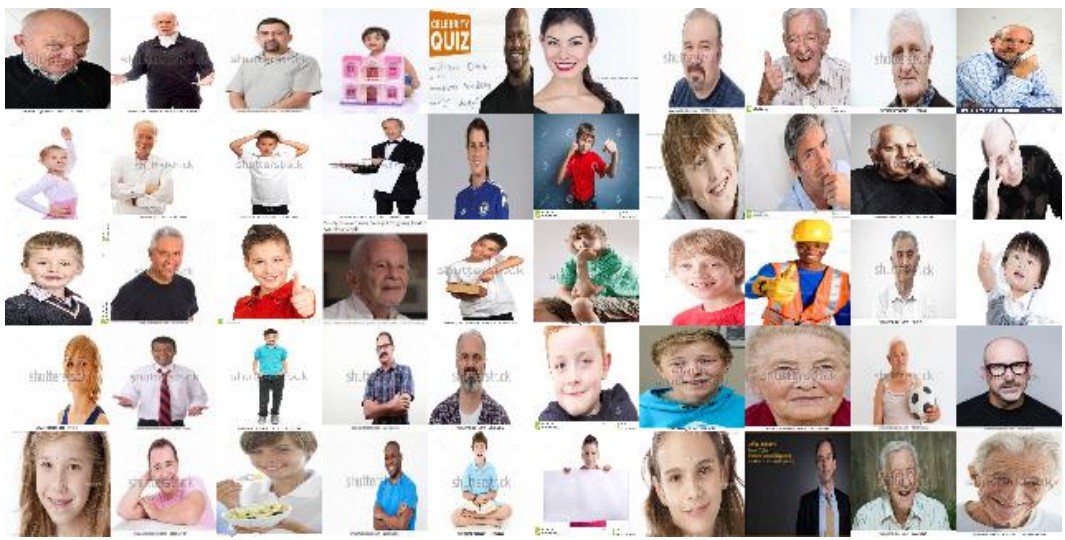

**Figure 33:** Retrievals for DebiasCLIP with query "A photo of an architect"

# H Experiment Notes

## H.1 GPT Generated Queries.

The 10 queries used in experiments and generated by GPT are as follows: [`programmer, nurse, architect, scientist, artist, chef, lawyer, teacher, engineer, doctor`].

## H.2 Experiment Setting and Details.

These experiments were conducted mainly on CPU after initially using a single A100 GPU to compute CLIP and DebiasCLIP embeddings and run FAISS to compute the top 10k candidates for each. Then, all embeddings are saved so that retrieval across all methods can be run on the CPU. When training linear probes, we conduct a cross-validated hyperparameter search from 0.01 to 100 over $\ell_2$ regularized logistic regression models from scikit-learn [69] to find the optimal probe.

To run `MOPR`, we use Gurobi [98] with an Academic License to solve Eqn. (9) each iteration of our cutting plane method and use CVXPY [99, 100] to solve our closed-form, quadratic program.

To train our oracle estimators, we call scikit-learn's implementations of Linear Regression, Decision Trees with depth 3, and MLPs with one hidden layer of size 64.

On average, the runtime for `MOPR` is approximately 1-5 minutes with 50 queries and 10000 elements in each of our retrieval and curation sets. However, with more features (such as using CLIP embeddings instead of group attributes), this runtime can increase. We note, however, that our most competitive baseline, MMR, is a greedy algorithm that requires 1-2 hours per query to compute. In live image retrieval settings, this runtime is too slow to be of any practical use.

