# OpenReview forum: "Multi-Group Proportional Representation in Retrieval"
_NeurIPS.cc/2024/Conference — NeurIPS 2024 poster_

### Official Review · Reviewer_4ELS · 2024-06-30

**Soundness:** 4
**Presentation:** 3
**Contribution:** 3
**Rating:** 7
**Confidence:** 4

**Summary:**

Authors proposed Multi-Group Proportional Representation, which is a metric to measure representation across multiple intersectional groups (could be many and overlapping) for retrieval. Given a set of representation statistics which quantify group membership in the target groups, MPR measures that group representation in the retrieval dataset approximates that in the wider target distribution. Of all the functions in the representation statistics class, it returns the function most violated. They provide generalization bounds for approximating the target distribution Q with a curated dataset, and provide guidelines for how large of a dataset to curate. They then refactor MPR to be suitable for calculation over a curated dataset. They also propose MAPR, a retrieval framework to relatively efficiently find a retrieval dataset that best performs retrieval while satisfying the MPR constraint.

The first experimental results compare the ability to achieve similarity while keeping MPR low, showing that their method consistently performs best compared to 4 other methods. They additionally show that their method can provide perfectly proportional representation when given an ideal dataset.

**Strengths:**

A) Authors effectively take a simple idea and support it with non-trivial error bounds / show evidence towards ideal dataset size

B) Proofs are clear

C) Explanations are clear and professionally presented

D) MAPR provides some algorithmic optimization

E) Problem (intersectional representation) is very important, and providing solutions for this in the retrieval setting shows novelty

**Weaknesses:**

A) This method is still extremely inefficient. Aside from making it difficult to use in practice, the results are only on 10 queries--this is very few and not great in terms of statistical significance

B) Despite acknowledging that the algorithm is very slow (line 351), there is no concrete analysis presented on runtime.

C) Check out the Proposition labeling in the Appendix--they extend the numbering from the paper rather than referring to which is being discussed

D) In Algorithm 1 line 3, do you mean to say "arg max a \in [0,1] n" rather than m?

E) Table 1 has several presentation / explanation problems. 1) Table 1 is presented first at the top of page 8, but I only see it discussed in lines 377 - 381 -- which is after the discussion from Figure 1, which is shown second. Am I missing something here? It would be clearer if they were presented in the same order as discussed. 2) Is Table 1 only referenced in the Results section starting at line 368? Why not in the Experimental Setup section starting at line 344? It makes it difficult to find where the explanation of the table is.

F) In Figure 1, the colors for Ours and MMR are way too close--it makes it very difficult to distinguish quickly between the fit lines (of course the shapes provide some indication, but more different colors would be much better)

G) in Appendix A line 697, please refer to the equation by its assigned number

Overall, my primary concern is the runtime practicality. As the idea itself is quite simple / straightforward, the contributions would be much stronger if the authors made it reasonable to do this in practice.

**Questions:**

A) In Figure 1, the axes are described as "fraction of", but I do not find it immediately obvious what this means. Please describe what this means or how it is calculated in more detail, either in the experimental description or figure caption

B) Table 1 shows proportional representation, but there is no mention here of similarity. What is to say that MAPR does not retrieve random samples in the correct group proportions?

**Limitations:**

This is mostly fine, although the runtime should be discussed / provided in more detail, as pointed out in Weakness B

---

> ### Author Rebuttal · Authors · 2024-08-07
>
> # Response to Reviewer 4ELS
>
> We thank the reviewer for their careful review and valuable feedback and for raising the issue of the computational efficiency of our method. We have addressed this issue in our response below, in the general remarks above, and in the attached pdf.
>
> ## WA/B. "This method is still extremely inefficient. Aside from making it difficult to use in practice, the results are only on 10 queries--this is very few and not great in terms of statistical significance". "Despite acknowledging that the algorithm is very slow (line 351), there is no concrete analysis presented on runtime."
>
> Please note that, on line 351, we are not describing our method but instead the SOTA and most competitive method to ours, the MMR algorithm proposed in [24, 28]. The runtimes reported in Table 1 of the rebuttal pdf demonstrate that MAPR (our method) runs up to 100x faster than MMR.
>
> Relative to other methods that achieve a worse representation-similarity trade-off relative to ours, we achieve competitive runtimes. The exception is DebiasCLIP, which is based on "debiasing" CLIP embeddings and thus uses the fast KNN algorithm for retrieval. However, DebiasCLIP was one of the worst-performing methods in our benchmarks. We kindly ask you to refer to our general response for more details on performance relative to the competitors.
>
> From a theoretical perspective, MAPR is actually very efficient when implemented with modern optimization techniques. MAPR minimizes a linear function subject to two linear constraints and iteratively adds MPR constraints, which amount to two linear constraints per iteration. The MPR constraint has a closed-form solution in the case of linear functions (Prop. 4, Sec. 3) or, more generally, an RKHS (Prop. 10, Appendix B.2), allowing us to directly solve a quadratic program with a conic structure, e.g., in the case of linear representation statistic functions. Such quadratic problems can be solved very efficiently with off-the-shelf conic solvers such as OSQP (available in Python).
>
> ## WC. "...Proposition labeling in the Appendix--they extend the numbering from the paper rather than referring to which is being discussed."
> Thank you. We will fix this in the final version of this paper.
>
> ## WD. "In Algo. 1 line 3, do you mean arg max $a \in [0,1]^n$ rather than m?"
>  Yes, thank you. Fixed!
>
> ## WE. "1) Table 1 is presented first at the top of page 8, but I only see it discussed in lines 377-381 -- which is after the discussion from Fig. 1, which is shown second. ... It would be clearer if they were presented in the same order as discussed. 2) Is Table 1 only referenced in the Results section starting at line 368? Why not in the Experimental Setup, starting at line 344? It makes it difficult to find the explanation for the table".
>
> 1) We wanted Table 1 to be first as an easily-digestible __teaser image__ for the paper. However, given that it comes so late in the paper, we will correct the order to follow the results.
> 2) We will elaborate on the setup for Table 1 in the experimental setup section of the final version of the paper by moving up and expanding on lines 377 and 378.
>
> ## WF. "In Fig. 1, the colors for Ours and MMR are way too close--it makes it very difficult to distinguish quickly between the fit lines."
>
> Thank you for the feedback! We will choose a more visually discernable color palette in the final version of our paper.
>
> ## WG. "in line 697, please refer to the equation by its assigned number"
> We will do so -- apologies for the confusion. We will use the correct label instead of saying "third equation" and "fourth equation" in Prop. 8.
>
> ## QA. "In Figure 1, the axes are described as "fraction of", but I do not find it immediately obvious what this means."
> Thank you for the question. The axes show normalized values that allow comparison across different queries. In this fig.:
>
> 1. $y$-axis (Similarity): The mean cosine similarity between the query embedding and retrieved image embeddings for $k$ retrieved images, divided by the maximum possible similarity given by the top-$k$ nearest neighbors of a query.
>
> 2. $x$-axis (Representation): The multi-group proportional representation (MPR) score, divided by the MPR score of the top-$k$ nearest neighbors.
>
> This normalization places the $k$-NN baseline (1,1) for all queries, allowing consistent comparison against the $k$-NN baseline where no fairness intervention is used. The ideal operation point of perfect representation and maximum similarity would be (0,1). We apply this normalization since, for each different query of the form "A photo of a [query]," the mean cosine similarity and the MPR values achieved by $k$-NN changes. This graph allows us to visualize how the MAPR algorithm and competing baselines trade off similarity and representation as it moves away from the vanilla $k$-NN retrieval. The unnormalized case would require different graphs for different queries since mean (cosine) similarity and MPR scores would be query-dependent. We will include this explanation in the experimental setup of the paper.
>
> ## QB. "Table 1 shows proportional representation, but there is no mention here of similarity. What is to say that MAPR does not retrieve random samples in the correct group proportions?"
>
> In these experiments, we pick the lowest-MPR values achieved by MAPR (e.g., the leftmost point on the curves traced in Fig 1.). Intuitively, the sets reported in the table are the most similar set of items possible that satisfy the minimum-achievable MPR by MAPR. Because MAPR optimizes for similarity subject to an MPR constraint, in Table 1 (see also Table 3 in Appendix D), MAPR isn't retrieving random samples but rather the most similar ones while still balancing group attributes. This is highlighted in Fig. 1, where MAPR Pareto-dominates competing methods in terms of MPR and preserved similarity. Nevertheless, **we report the similarity of the retrievals used to create Table 1 in Table R2 in the attached pdf.**

---

> > ### Comment · Reviewer_4ELS · 2024-08-08
> > **Response to Rebuttal**
> >
> > Thank you for your response to my concerns and questions. Many of your answers are satisfactory to me (once again, thank you for those), so for conversation clarity I will just address the few remaining discussion points I have here:
> >
> > WA/B-1) The additional runtime analysis is good, but I just want to make sure I am interpreting the relationship with PBM correctly: PBM is faster than MAPR and performs similarly in terms of similarity / MPR curve (figure 1). However, what we gain for the extra time is balance across groups, regardless of intersectionality (table 1).
> > WA/B-2) I am aware that 10 queries is what is used in previous literature, but I still have doubts that this is not very many samples for statistical significance. The additional experiments scale up the number of neighbors, but keep the number of queries fixed. Could you please comment on how this is enough samples to make conclusions? I apologize if I am missing your previous response to this point, but in the rebuttal I only see a response to the runtime concerns.

---

> ### Author Response · Authors · 2024-08-10
> **Response to Reviewer 4ELS - Part 1**
>
> Thank you for engaging with our response during the discussion period. We appreciate the very interesting questions.
>
> ## WA/B-1
>
> We will clarify the key similarities and differences between MAPR and PBM:
>
> - **Representation Goal:** You are correct that PBM ensures *equal representation* across a *single group*, while MAPR ensures *proportional representation* across *multiple intersectional groups*.
> - **Groups:** PBM focuses on a single binary attribute (though allowing for N/A values) and does not handle intersectional groups. MAPR considers multiple attributes simultaneously and considers the representation of intersecting groups (e.g., Women and Black Women).
> - **MPR Gap:** At high MPR levels, PBM and MAPR are both able to achieve comparable similarity scores in retrieval. However, it is incorrect to say PBM and MAPR perform similarly across the entirety of the similarity / MPR curve. PBM is unable to achieve low MPR values, as evidenced by it not extending all the way to the left on the graphs in Figure 1.
> - **Runtime:** You are indeed correct that MAPR has a higher runtime cost due to handling multiple attributes. Since PBM considers a single attribute, it is faster.
>
> In summary, PBM is only comparable to MAPR on the similarity/MPR curve for high MPR values due to its restriction to a single binary group. This can be seen in Fig. 1, with PBM not extending to the left-hand side of the graph (low MPR values). We explain why this is the case next.
>
> Recall that MPR -- our metric of interest -- quantifies the worst-case deviation in proportional representation across a collection of groups. In Fig. 1, these groups are determined by race, age, and gender attributes. PBM does an excellent job ensuring MPR **up to a point**, matching the performance of MAPR for high MPR values. However, it does not reach low MPR values (e.g., Fig. 1 center). This happens because PBM considers **equal representation** of a **single binary attribute**: we pick gender since this is a central focus of the PBM paper, and the attribute "race" in our experiments is not binary.
>
> PBM manages to close the gender representation gap in retrieved images in Fig. 1, thus lowering MPR up to a point. However, it stalls in achieving representation for other attributes such as race and age. Of course, this is not surprising: PBM is designed for a single attribute. MAPR (our method) manages to achieve proportional representation across groups, reaching lower MPR points on the left-hand side of the graph and thus significantly lowering the representation gap at a slightly higher runtime cost.
>
> Extending the single-attribute balancing algorithm behind PBM to multiple attributes is possible but very likely would incur a higher runtime due to requiring balancing over an *exponential* number of intersectional groups. We will add this discussion to the appendix of the paper, and summarize it in the main paper. We hope this clarifies your understanding!

---

> ### Author Response · Authors · 2024-08-10
> **Response to Reviewer 4ELS - Part 2**
>
> ## WA/B-2
>
> We apologize for not addressing this point. To verify whether 10 queries are statistically significant, we ran a second set of experiments with 45 queries on the Occupations dataset, using the ground-truth occupation categories in the dataset as our target query. Our goal is to verify if there is a statistically significant difference in our findings when we run the same experiments with more queries. To do so, we conduct a statistical hypothesis test between results with 10 queries and with 45 queries to check significant changes in the average performance of MAPR and competing methods.
>
> We measure *Max Reduction in MPR* with the formula `(MPR_min - MPR_topk)/MPR_topk`. This measures the best improvement in MPR achieved by each baseline as a percentage of the baseline top-k MPR. We report the mean and standard deviation of this max reduction over both 10 and 45 queries in Table 1.
>
> We conduct a 2 sample t-test (which can be easily done with the SciPy's Stats package) with 53 degrees of freedom (45 queries + 10 queries - 2), testing the hypothesis of a significant statistical difference between these two means. The idea is to test the null hypothesis that *there is no significant difference in the performance between using 10 queries and using 45 queries*. In this case, a small p-value would mean that we reject the null hypothesis in favor of the alternative hypothesis that there is a statistical difference between 10 and 45 queries. The t-statistic and p-value are also reported in the table below. For MAPR and most of our baselines, the t-test reports a p-value significantly greater than 0.05, indicating that we **accept** the null hypothesis that the samples across these two experiments are drawn from populations with the same population mean (or as noted in the SciPy documentation, "our observation is not so unlikely to have occurred by chance"). Note also that the max reduction in MPR is very close across both experiments. Thus, our findings based on 10 queries are statistically significant, and increasing the number of queries by a factor of 4 did not lead to different conclusions.
>
> |             | Mean (STD), 10 Queries | Mean (STD), 45 Queries | t-statistic | p-value |
> |-------------|------------------------|------------------------|-------------|---------|
> | MAPR (Ours) | -0.943 (0.025)         | -0.948 (0.029)         | -0.476      | 0.636   |
> | MMR         | -0.628 (0.127)         | -0.702 (0.094)         | -2.032      | 0.048   |
> | PBM         | -0.070 (0.049)         | -0.065 (0.066)         | 0.180       | 0.858   |
> | ClipCLIP    | -0.400 (0.067)         | -0.326 (0.180)         | 0.928       | 0.358   |
> | DebiasCLIP  | 0.013 (0.160)          | 0.110 (0.227)          | 0.825       | 0.414   |
>
>
> Table 1. Max reduction in MPR averaged across 10 and 45 queries, as well as results for a 2-sample t-test between sets of queries. We see that there is no statistical significance in our results when increasing to 45 queries. MPR estimated and MAPR optimized with a linear regression oracle for k=50 samples on the Occupations dataset.
>
> Again, we thank the reviewer, and we will be happy to provide further clarification about our contribution.

---

> > ### Comment · Reviewer_4ELS · 2024-08-12
> > **Response to Rebuttal part 2**
> >
> > I am unconvinced by this test for several reasons.
> > 1) In general, t-tests must be designed such that you reject the null hypothesis. Not rejecting the null hypothesis is not the same as accepting the null hypothesis.
> > 2) On a theoretical level, you have set this up such that such that my statement is what should be proved, and shown that it is not possible to prove my statement (aka that may be a difference between 10 and 45 queries). However, it is not me that needs to show that the experimental values may not be significant--it is you that needs to show that they are significant.
> > 3) MMR is actually below 0.05 for p-value, which is a commonly accepted threshold for acceptance.
> > 4) Looking at the differences in the means, I actually believe that several are rather significant (most strongly MMR, ClipCLIP)
> > 5) This has been shown on one dataset, but could be different across datasets (I'm not saying you should try it on more datasets for this rebuttal as I see this is unrealistic given the timeframe and appreciate the attempt to show results at all, but just that this is a weakness of the presented results).
> >
> > If you would like to make the case that 10 profession queries is enough, I would still be open to that. The issue I see is that I would expect high variance across different professions. If these were more similar to seeds I would say 10 is plenty, but they are measurements of how well the method can do given different queries, which I would interpret as more similar to different samples.

---

> ### Author Response · Authors · 2024-08-13
> **Further discussion of statistical significance.**
>
> First of all, thank you again for engaging with us during this discussion period. Your feedback will help us tune and improve our manuscript. We really appreciate your time, attention, and thorough assessment of our work.
>
> We acknowledge that the hypothesis test was rushed and could have been framed better -- we ran it during the weekend due to the time crunch surrounding the discussion period and wanted to answer your question promptly. We regret the error in our interpretation of the hypothesis test. Our results from the previous comment have not demonstrated that 10 queries are statistically indistinguishable from results over 45 queries, but rather that we cannot exclude this possibility. Again, we apologize for this misinterpretation.
>
> Upon further reflection (and if we are interpreting your comment correctly), the main issue here is generalization: How can we guarantee that, if a given method achieves better MPR-similarity trade-off over 10 or even 1000 queries, it will still perform favorably for an unseen query on a new dataset? This issue permeates (to the best of our knowledge) other published experimental results in the fairness in retrieval literature. For example, our baselines PBM [26, Figs. 2,3] and MMR [24, Figs 1,4,10] include results for a single query. The difficulty here (as you note regarding different queries as different samples) is that there is no simple prior distribution over datasets and queries over which we can frame randomization claims, making measuring statistical significance challenging.
>
> Our experiments and findings (and those of related literature) are deterministic: we select a query, retrieve items from a dataset using different "fair'' retrieval methods, and evaluate its similarity and representation. Across all datasets and for each **individual** query (including our new 45 queries from the Occupations benchmark [28]), MAPR Pareto-dominates competing benchmarks in its similarity-MPR curve. Note that Fig. 1 aggregates these results: we also observe the Pareto-dominance of MAPR at the individual query level. As the reviewer correctly points out, such consistent performance on a given set of queries does not preclude the existence of other queries and datasets where our method may perform less favorably, or even that on average over a "randomly chosen query" (however that may be defined) MAPR may have worse performance.
>
> One way to address this issue is to be as transparent, precise, and clear as possible about our findings and limitations. We suggest three additions:
>  *  We will include *individual* MPR-similarity trade-off curves in the appendix, for each query for each dataset (including the 45 additional Occupations queries). These curves do not include error bars, since they represent deterministic measurements of the performance of each method. This will demonstrate, **at the individual query level** for the datasets and queries we used (i.e. across 3 x 55 different "settings", or dataset-query pairs), that MAPR consistently Pareto-dominates competing methods listed in Fig. 1, a stronger statement than Pareto-dominance in aggregate.
>
>  * We will add the following text to the Limitations section and point to it in the numerical experiments: "The reader must interpret the aggregate results in Fig. 1 with caution. MPR and similarity are deterministic measurements, and the error bars represent confidence intervals over 10 pre-selected queries across each of the three datasets. Fig. 1 should **not** be interpreted as statistical evidence of performance for any of the benchmarked methods (including MAPR). In the appendix, we show MAPR Pareto-dominates competing methods not only in aggregate, but at the individual query level. This does not preclude the existence of a query where a competing method may achieve an MPR-similarity trade-off point not achieved by MAPR. However, we note that MAPR is specifically optimized to promote MPR while preserving similarity, explaining its favorable performance. In all instances observed by the authors, MAPR Pareto dominates competing benchmarks. Crucially, across all queries, MAPR was the most successful method in achieving low values of MPR while maintaining high levels of similarity with a given query."
>  * We will expand on the qualitative difference with competing methods, similar to Table 1 in the paper. In particular, we highlight that the methods most competitive to ours -- PBM and MMR -- have fundamental differences. As discussed above, PBM **does not** ensure representation across **multiple** groups, and MMR takes significantly longer to run (50x-100x).
>
> Finally, despite the limitations in existing benchmarks to make rigorous statistical conclusions as discussed above, we hope the reviewer can appreciate the technical contributions of our work, including the introduction and theoretical characterization of the MPR metric itself and our demonstration of practical methods to estimate and optimize for it via MAPR.

---

> > ### Comment · Reviewer_4ELS · 2024-08-13
> > **Change of Rating**
> >
> > Thank you for the thorough explanation of the interpretation of the queries, it helped me much better to understand what is being measured here. While I still believe on a high level that there is room for improvement in the evaluation of these types of methods, I will concede that it is extremely nontrivial, making it the task of future work rather than this particular paper. As you highlight, there are still significant contributions to this work. The proposals to present more detailed individual results in the appendix and highlight the interpretation in the text I believe also make the work much more transparent to the reader.
> >
> > I am raising my rating from 6 to 7, as my primary concerns of runtime and significance have been adequately addressed.
> >
> > While I feel this is valuable work that should be shared with the community, I limit my score to Accept because of the complications in the evaluation at the individual level. Once again, I see that there are complications in providing results beyond the individual query level that go beyond the scope of this work and I see that the individual queries provide a clear and valuable measurement given this complicated setup, but I still find it fundamentally difficult to reliably compare two methods based on single (or very few) samples. Specifically, I am unconfident in the similarity comparisons, which are core to retreival quality.
> >
> > However, I would like to thank the authors once again for the thoughtful discussion in "Further discussion of statistical significance". Before this comment I was leaning towards lowering my rating, but have instead decided to raise it.

---

> > > ### Author Response · Authors · 2024-08-13
> > > **Thank you!**
> > >
> > > We sincerely appreciate your decision to raise your score after discussing our paper and rebuttal with us. Your feedback and engagement throughout the review and discussion process has certainly improved our paper, and we will be sure to incorporate these changes and clarifications into our final version. If you have any further questions or concerns, please feel free to discuss with us.

---

### Official Review · Reviewer_XQou · 2024-07-03

**Soundness:** 4
**Presentation:** 3
**Contribution:** 4
**Rating:** 9
**Confidence:** 3

**Summary:**

This paper introduces the novel metric of Multi-Group Proportional Representation (MPR) designed to address the representational harms in image search and retrieval tasks, which can perpetuate harmful stereotypes, erase cultural identities, and amplify social disparities. Current methods to mitigate these issues often balance the number of retrieved items across population groups defined by a limited set of attributes, typically overlooking intersectional groups defined by combinations of attributes such as gender, race, and ethnicity.

Key contributions of the paper include:

Introduction of MPR Metric: The paper presents the MPR metric, which aims to ensure proportional representation of multiple groups in retrieval tasks, considering intersectionality and combinations of different group attributes.

Theoretical Guarantees: The authors provide theoretical foundations for the MPR metric, establishing its effectiveness and applicability in various contexts.

Practical Algorithm: A practical algorithm for promoting MPR in retrieval tasks is developed. This algorithm is evaluated through numerical experiments to demonstrate its superiority over existing methods that optimize for equal and proportional representation metrics but may fail to achieve MPR.

Experimental Results: The paper includes detailed numerical experiments showing that the proposed MPR-aware retrieval algorithm significantly closes the gap in MPR, outperforming baseline methods across different datasets and retrieval tasks.

Discussion of Limitations: The authors acknowledge several limitations of their work, including the computational complexity of the algorithm in high-dimensional feature spaces, the need to adapt MPR to other domains like ranking and recommendation systems, and the inherent limitations posed by biased datasets.

Overall, the paper makes significant strides in addressing the complex issue of representational harms in image retrieval tasks by introducing and validating the MPR metric and associated algorithms .

**Strengths:**

Originality:
Novel Metric: The paper introduces the Multi-Group Proportional Representation (MPR) metric, a novel approach to addressing representational harms in image retrieval tasks. This is a significant advancement over traditional methods that focus on equal or proportional representation without considering intersectionality.
Theoretical and Practical Contributions: The combination of theoretical guarantees with a practical algorithm for achieving MPR demonstrates a high level of originality. The approach is both innovative in its formulation and effective in its execution.

Quality:
Rigorous Methodology: The paper employs a thorough and well-documented experimental methodology. The numerical experiments are designed to rigorously test the effectiveness of the proposed algorithm against baseline methods.
Strong Theoretical Foundations: The theoretical underpinnings of the MPR metric are clearly articulated and well-supported, providing a robust foundation for the practical contributions of the paper.
Comprehensive Evaluation: The experimental results are comprehensive and convincingly demonstrate the superiority of the proposed approach, enhancing the overall quality of the research.

Clarity:
Clear Writing and Structure: The paper is well-written and logically structured, making it easy to follow the progression of ideas and understand the key contributions. The writing style is clear and concise, facilitating comprehension.
Detailed Explanations: The authors provide detailed explanations of both the theoretical and practical aspects of their work, ensuring that readers can fully grasp the significance and implementation of the MPR metric.
Contextualization: The paper effectively situates its contributions within the broader context of existing research, clearly delineating how it advances the state of the art in fair and diverse retrieval.

Significance:
Addressing a Critical Issue: The focus on representational harms in image retrieval tasks is highly significant, as it addresses a critical issue in AI/ML that has broad societal implications.
Impactful Results: The results of the study are valuable and impactful, offering a new approach that significantly improves over existing methods. This has the potential to influence future research and applications in the field.
Broad Applicability: While the paper focuses on image retrieval tasks, the principles and approaches developed could be extended to other domains, increasing the overall significance and potential impact of the work.

Overall, the paper makes substantial contributions across multiple dimensions, demonstrating a high degree of originality, quality, clarity, and significance.

**Weaknesses:**

Computational Complexity:
Algorithm Efficiency: One potential weakness of the paper is the computational complexity of the proposed algorithm. As noted by the authors, the algorithm can be computationally intensive in high-dimensional feature spaces, which could limit its practical applicability in large-scale or real-time systems.
Actionable Insight: Future work could focus on optimizing the algorithm to reduce computational overhead or developing approximation techniques that maintain high accuracy while improving efficiency.

Scope of Evaluation:
Limited Evaluation Domains: The evaluation of the MPR metric is primarily focused on image retrieval tasks. While the results are impressive within this domain, the generalizability of the approach to other domains such as ranking systems, recommendation systems, or text retrieval is not explored in depth.
Actionable Insight: Including preliminary experiments or discussions on how the MPR metric and algorithm could be adapted to other domains would strengthen the paper. Future research could expand the evaluation to demonstrate the broader applicability of the approach.

Dataset Bias:
Bias in Datasets: The paper acknowledges the limitations posed by biased datasets but does not delve deeply into how the proposed approach handles or mitigates these biases. The effectiveness of the MPR metric in the presence of inherently biased datasets is an important aspect that needs further exploration.
Actionable Insight: Providing a more detailed analysis of how dataset biases impact the performance of the MPR metric and discussing potential strategies for mitigating these effects would enhance the robustness of the findings. Future work could include experiments with datasets known to have specific biases to assess and address these challenges.

Reproducibility:
Details on Reproducibility: While the paper discusses the reproducibility of results, providing more explicit details on the implementation and data used could improve the ability of others to replicate the findings.
Actionable Insight: Including supplementary materials such as code, datasets, and detailed implementation steps would significantly enhance the reproducibility of the research. Clearer documentation of experimental settings and parameters would also be beneficial.

Ethical Considerations:
Ethical Analysis: The paper mentions that no ethical considerations remain unaddressed, but a more thorough discussion on the ethical implications of the MPR metric and its potential societal impact would add depth to the work.
Actionable Insight: Expanding the discussion on ethical considerations, including potential unintended consequences and mitigation strategies, would provide a more comprehensive view of the implications of the research.

By addressing these weaknesses, the paper could further improve its robustness, applicability, and overall impact on the field.

**Questions:**

Questions:

Computational Complexity:
Question: Can you provide more details on the computational requirements of the proposed algorithm? Specifically, how does its performance scale with increasing dimensionality and size of the dataset?
Suggestion: It would be helpful to include a discussion or additional experiments that benchmark the computational performance of your algorithm compared to baseline methods, particularly in high-dimensional scenarios.

Generality of MPR Metric:
Question: How well does the MPR metric generalize to other domains beyond image retrieval, such as text retrieval or recommendation systems? Have you considered or conducted any preliminary experiments in these areas?
Suggestion: Including a discussion on the potential adaptations of the MPR metric to other domains and any initial results from such experiments could strengthen your paper.

Handling Dataset Bias:
Question: How does your approach address or mitigate the effects of inherent biases in the datasets used? Have you evaluated the MPR metric's performance with datasets known to have specific biases?
Suggestion: Consider providing more detailed analyses or experiments that explicitly evaluate the impact of dataset biases on the performance of your method, and discuss any strategies for mitigating these biases.

Reproducibility:
Question: Can you elaborate on the steps taken to ensure the reproducibility of your results? Are there any supplementary materials (e.g., code, datasets, detailed experimental settings) available to assist other researchers in replicating your work?
Suggestion: Providing comprehensive supplementary materials, including code, datasets, and detailed documentation of experimental setups, would significantly enhance the reproducibility of your research.

Ethical Considerations:
Question: Could you provide a more detailed discussion on the ethical implications of the MPR metric? Specifically, what are the potential unintended consequences, and how might they be mitigated?
Suggestion: Expanding the discussion on ethical considerations, including potential risks and mitigation strategies, would offer a more thorough examination of the societal impact of your work.

By addressing these questions and incorporating the suggested improvements, the paper could provide a more comprehensive and robust contribution to the field.

**Limitations:**

Assessment:
The authors have acknowledged several limitations of their work, including the computational complexity of the algorithm in high-dimensional feature spaces and the need to adapt the MPR metric to other domains like ranking and recommendation systems. They have also recognized the limitations posed by biased datasets. However, there is room for a more detailed discussion on these limitations and their potential negative societal impacts.

Suggestions for Improvement:

Computational Complexity:
Current Acknowledgment: The authors note the computational complexity of the proposed algorithm in high-dimensional feature spaces.
Suggestion for Improvement: Providing more specific details on the computational requirements, such as time complexity analysis and potential optimizations, would give readers a clearer understanding of this limitation. Including practical suggestions for reducing computational overhead or discussing ongoing work to improve efficiency would be beneficial.

Generality Across Domains:
Current Acknowledgment: The need to adapt the MPR metric to other domains is mentioned.
Suggestion for Improvement: Elaborating on the challenges and potential approaches for applying the MPR metric to other domains, such as ranking systems and recommendation engines, would strengthen this acknowledgment. Preliminary experiments or theoretical discussions on these adaptations could provide valuable insights.

Dataset Bias:
Current Acknowledgment: The paper mentions the limitations posed by biased datasets.
Suggestion for Improvement: Conducting a more in-depth analysis of how dataset biases impact the performance of the MPR metric and discussing specific mitigation strategies would enhance this section. Experimental evaluations using biased datasets could provide empirical evidence and insights into this limitation.

Ethical Considerations and Societal Impact:
Current Acknowledgment: The paper claims that no ethical considerations remain unaddressed but does not provide a detailed discussion on potential negative societal impacts.
Suggestion for Improvement: Including a comprehensive discussion on the ethical implications of the MPR metric, such as potential unintended consequences and strategies for mitigation, would provide a more balanced view. Addressing questions like "Could the MPR metric inadvertently reinforce certain stereotypes?" or "What safeguards can be implemented to prevent misuse?" would be valuable.

By expanding on these areas, the authors can provide a more thorough examination of the limitations and potential societal impacts of their work. This transparency not only enhances the credibility of the research but also provides a solid foundation for future improvements and applications.

---

> ### Author Rebuttal · Authors · 2024-08-07
>
> # Response to Reviewer XQou
>
> We are grateful for your review and positive feedback. We will address each of the weaknesses and questions raised (in a combined way) in detail below. We will be happy to answer any further question.
>
> ## Q1. "Can you provide more details on the computational requirements of the proposed algorithm? Specifically, how does its performance scale with increasing dimensionality and size of the dataset?"
>
> Thanks for raising this --- we will address this point in the updated version of our paper. We provide benchmarks in the table attached to the rebuttal. Our method (MAPR) is competitive with existing fair retrieval methods in terms of runtime. Relative to methods with comparable diversity-similarity trade-offs, such as MMR (see Fig. 1 of the submitted paper), our method has 10x-100x lower computational overhead per query. The one-page additional material includes a table with runtime results for our method and competing baselines. We kindly ask that you refer to the general rebuttal above for more details surrounding the discussion on runtime.
>
> From a theoretical perspective, MAPR is actually very efficient when implemented with modern optimization techniques. MAPR minimizes a linear function subject to two linear constraints and iteratively adds MPR constraints, which amount to two additional linear constraints per iteration. The MPR constraint has a closed-form solution in the case of linear functions (Proposition 4, Section 3) or, more generally, an RKHS (Proposition 10, Appendix B.2), allowing us to directly solve a quadratic program with a conic structure, e.g., in the case of linear representation statistic functions. Such quadratic problems can be solved very efficiently with off-the-shelf conic solvers such as OSQP (available in Python).
>
> ## Q2. "How well does the MPR metric generalize to other domains beyond image retrieval, such as text retrieval or recommendation systems?"
>
> We believe our approach has significant potential for adaptation to other domains, such as recommender systems and text retrieval. Since we only made use of embeddings of images in all our numerical benchmarks, our method would directly apply to any other setting where a search is performed over vector embeddings, including text retrieval. In this case, we could use our method to ensure a proportional representation of different perspectives, sources, or demographics in retrieved text documents.
>
> ## Q3. "How does your approach address or mitigate the effects of inherent biases in the datasets used? Have you evaluated the MPR performance with datasets known to have specific biases?"
>
> The goal of the curated dataset is to use or construct a representative and diverse set *a priori* so that retrieval can be representative. As discussed in lines 162--163 (highlighted in bold) and lines 396--399, if this curated dataset is biased, such biases can propagate in retrieval.
>
> There may be cases where the retrieval dataset itself is so biased that proportional representation is impossible to achieve (e.g., all images in the dataset over which retrieval is performed are from one demographic group). In such cases, our metric MPR will flag gross deviations in representation relative to a curated dataset.
>
> ## Q4. "Can you elaborate on the steps taken to ensure the reproducibility of your results?"
>
> In the submission, we included an anonymized GitHub [code](https://anonymous.4open.science/r/representational-retrieval-86BB/). We will extend the readme file and create a tutorial for fairness practitioners to easily use our method. In the final version of the paper, we will include the codebase and a more extensive documentation of experimental setups. Moreover, all datasets used in this work are open-source and publicly available.
>
> ## Q5. "The paper mentions that no ethical considerations remain unaddressed, but a more thorough discussion on the ethical implications of the MPR metric and its potential societal impact would add depth to the work."
>
> Thanks for raising this point. We devote the introduction and much of Section 2 to discussing the ethical considerations of retrieval and MPR. We will expand our ethical discussion to include three critical concepts beyond the amplification of existing bias and misrepresentation of underrepresented groups already mentioned in the paper. First, we will address the potential for a __false sense of fairness__ that the MPR metric might create, emphasizing the importance of critical evaluation even when metrics suggest fairness. Second, we can discuss the __legal and regulatory risks__ associated with relying too heavily on such metrics, particularly in light of evolving anti-discrimination laws. Third, we will add comments about the implications of __skewed decision-making__ that could result from the uncritical application of the MPR metric, highlighting the need for human oversight and contextual understanding. Lastly, we will discuss safeguards such as comprehensive auditing, human oversight, and contextual application.

---

> > ### Comment · Reviewer_XQou · 2024-08-10
> > **Multi-Group Proportional Representation**
> >
> > Thank you for the detailed and thoughtful rebuttal. I appreciate the effort you have put into addressing the concerns raised in my review.
> >
> > Computational Complexity:
> > I appreciate the additional benchmarks provided in the rebuttal that demonstrate the competitiveness of your method in terms of runtime. The theoretical discussion on the efficiency of MAPR, particularly with modern optimization techniques, has also clarified the potential scalability of your approach. This helps address my concerns regarding the computational complexity, especially in high-dimensional feature spaces.
> >
> > Generality of the MPR Metric:
> > Your explanation of how the MPR metric could be adapted to other domains such as text retrieval and recommendation systems is convincing. The argument that your method is applicable to any setting involving vector embeddings broadens the potential impact of your work. The potential for adaptation to other domains strengthens the significance of your contribution.
> >
> > Handling Dataset Bias:
> > Your acknowledgment of the challenges posed by biased datasets, along with the explanation of how MPR flags deviations in representation when retrieval datasets are biased, provides a clearer understanding of how your approach handles these issues. This reassures me that you have considered the implications of biased datasets, even if further exploration in future work is necessary.
> >
> > Reproducibility:
> > I appreciate the commitment to enhancing the reproducibility of your results by extending the GitHub code, creating a tutorial, and documenting the experimental setups in more detail. These steps will undoubtedly aid other researchers in replicating and building upon your work.
> >
> > Ethical Considerations:
> > The expanded discussion on the ethical implications of the MPR metric, including the potential for a false sense of fairness, legal risks, and the importance of human oversight, adds significant depth to your work. This addition addresses the ethical concerns I mentioned and contributes to a more comprehensive understanding of the societal impact of your research.
> >
> > Given your thorough and thoughtful responses, I believe the paper has demonstrated robustness, applicability, and a high potential for impact. The clarifications and additional details provided in the rebuttal address many of the concerns I had, and I am inclined to improve my rating of the paper. I appreciate the effort you have put into making these improvements, and I believe your work makes a substantial contribution to the field.

---

> ### Author Response · Authors · 2024-08-11
> **Response to Reviewer XQou**
>
> We are glad you found our answers to your questions and concerns useful. If you have any more questions before the discussion period ends, please feel free to post them, and we will do our best to answer them. Finally, thank you for increasing your score. We are glad you feel our work substantially contributes to the field, and we look forward to developing connections between MPR and other facets of fairness as well as with diverse ML applications!

---

### Official Review · Reviewer_XWMb · 2024-07-12

**Soundness:** 3
**Presentation:** 2
**Contribution:** 2
**Rating:** 6
**Confidence:** 3

**Summary:**

This paper introduces a novel metric, Multi-Group Proportional Representation (MPR), to measure and ensure fair representation across intersectional groups in image retrieval. Current methods often ensure representation across individual groups, not intersectional groups. To address this, authors

**Strengths:**

1. Ensuring representation across intersectional groups by suggesting a new metric in image retrieval tasks.
2. The mathematical and theoretical foundations support the ideas.
3. A variety of experiments supports the methodology.

**Weaknesses:**

1. Fairness may include mixed-race individuals and various genders, not just White Male or Black Female. However, in this task, the 'White Male' as an intersectional group raises questions about whether it truly reflects the meaning of intersectionality.
2. FairFace is not a perfect fair dataset. However, the experimental results for individual groups in Table 1 show perfect fairness across gender and race. How is this possible?
3. In Table 1, there are significant differences between intersectional groups (e.g., Asian Male : Asian Female = 13 : 7).

**Questions:**

1. The concept of fairness is defined by humans and can vary across different countries and cultures. What are your thoughts on this aspect?
2. It is well-known that the CLIP model has significant bias issues, and many studies (e.g., https://arxiv.org/abs/2210.14562) have attempted to address this. Does your methodology incorporate such debiasing measures?
3. Is there any experiment or quantitative result about computational complexity?

**Limitations:**

The authors acknowledge the limitations regarding computational complexity and curated dataset.

---

> ### Author Rebuttal · Authors · 2024-08-06
>
> # Response to Reviewer XWMb
>
> We thank the reviewer for their careful reading of our paper and valuable feedback. We address each of the weaknesses and questions raised below. Please let us know if you have any additional questions.
>
> ## W1. "...the 'White Male' as an intersectional group raises questions about whether it truly reflects the meaning of intersectionality"
>
> This is an important point. Intersectionality, as a social science term, historically focuses on the intersection of marginalized identities (e.g., Black females) -- see [1] for a discussion.
>
> We also agree that intersectional groups are defined beyond simple binary combinations of siloed categories for race and gender and must account for individuals who, for example, identify with multiple racial identities. Our metric allows for measuring the proportional representation of such groups via the appropriate definition of the class $\mathcal{C}$ and its inputs. For instance, when encoding group membership $\mathbf{g}_i$ (line 134), the attribute race can be represented as a binary vector admitting multiple entries with 1 rather than as a one-hot vector, thus accounting for individuals with multiple racial identities. We will add this important point to the paper.
>
> We assume the White Male group mentioned by the reviewer may refer to the groups denoted in Table 1. Here, we select population groups based on the features available in the UTKFace dataset. The UTKFace dataset includes labels for individuals' identities in terms of non-overlapping race and gender attributes, thus not explicitly labeling individuals as members of multiple racial groups or as non-binary. Since we use the labels in the UTKFace dataset, the groups in Table 1 do not overlap. We will highlight this limitation in the main paper.
>
> ## W2. "results for individual groups in Table 1 show perfect fairness across gender and race. How is this possible?" and "In Table 1, there are significant differences between intersectional groups (e.g., Asian Male: Female = 13 : 7)"
>
> MAPR ensures proportionality with respect to a curated dataset. In Table 1, we create a perfectly balanced synthetic curated dataset that is used as reference (line 377). Moreover, MAPR aims to ensure proportional representation simultaneously across multiple groups, including groups defined by individual attributes (e.g., race or gender) and by intersectional attributes (e.g., race **and** gender). As we consider more combinations of attributes, achieving proportional representation becomes more difficult.
>
> In Table 1, MAPR successfully balances representation at the individual-attribute level and significantly closes the representation gap at the intersectional level. MAPR does not achieve exactly equal representation at the intersectional level due to the distribution of the underlying retrieval set not including enough samples for every group. However, Table 1 demonstrates that 1) MAPR still does better than vanilla k-NN and MMR (the state-of-the-art competing benchmark) for intersectional groups, and 2) MAPR can achieve perfectly balanced representation for groups defined by a single attribute.
>
> ## Q1 "fairness is defined by humans and can vary across countries and cultures. What are your thoughts?"
>
> You raise an excellent point about the cultural and contextual nature of fairness. The paper acknowledges this complexity, particularly in the context of representation in ML systems. Our flexible metric (MPR) can accommodate different notions of "fair representation" across countries and cultures by allowing users to define their own curated data set that is representative of a target population or desired representation goals. Moreover, the groups for which we aim to ensure representation can be encoded in the set of representation statistics $\mathcal{C}$ (see lines 174-204).
>
> The goal of this work is to create a multi-group representation metric that is flexible enough to accommodate varying notions of representation while still providing practical utility and mathematical tractability. Our "mathematization" of multi-group proportional representation allows us to understand the statistical limits of estimating representation and ensuring it algorithmically. However, like any "mathematization" of fairness, it is only one piece of the puzzle. We highlight the need to carefully consider cultural and societal factors when defining fairness and representation goals. Moreover, we emphasize the importance of involving stakeholders in defining representation goals and auditing retrieval systems. We include the discussion in the final version of our paper.
>
> ## Q2 "CLIP model has significant bias issues, and many studies have attempted to address this. Does your methodology incorporate such debiasing measures?"
>
> Our method does not directly rely on debiasing CLIP embeddings. Instead, we leave CLIP embeddings intact and rely on group-denoting attributes to ensure a proportional representation of computationally-identifiable groups, denoted by the set of representation statistics $\mathcal{C}$. Our experiments show that this approach achieves significantly better performance than methods that aim to "debias" CLIP embedding such as DebiasCLIP and clipCLIP (see Fig. 1 for benchmarks).
>
> ## Q3 "Is there any experiment or quantitative result about computational complexity?"
>
> That is a great question that will be addressed in the final version of our work. We provide benchmarks in the Table R1 attached. MAPR is competitive with existing fair retrieval methods in terms of runtime. Relative to methods with comparable diversity-similarity trade-offs such as MMR (Fig. 1 of the paper), our method has 10x-100x lower computational overhead per query. Please refer to the general rebuttal above for more details surrounding the discussion on runtime.
>
> [1] K. Crenshaw. Demarginalizing the intersection of race and sex: A black feminist critique of antidiscrimination doctrine, feminist theory and antiracist politics.

---

> > ### Comment · Reviewer_XWMb · 2024-08-12
> >
> > Thank you for the kind and detailed explanation. I have a few more questions I would like to ask:
> >
> > 1. Regarding your statement, "As we consider more combinations of attributes, achieving proportional representation becomes more difficult," I am curious if this means that in future work, achieving perfect balance will be impossible, or if it is something that you will continue to strive for.
> >
> > 2. Concerning "Moreover, the groups for which we aim to ensure representation can be encoded in the set of representation statistics," I wonder if these representative groups, which are ultimately based on subjective human judgment, can truly reflect fair representation.

---

> ### Author Response · Authors · 2024-08-13
> **Response to Reviewer Comment**
>
> We are glad you found our explanation useful and thank you for the additional questions! We will do our best to answer them below.
>
> **Achieving perfect balance.** As we consider more and more intersectional groups, there is a point at which there are more intersectional group identities than retrieved items, in which case we cannot achieve 0 MPR against a properly representative curated dataset. An extreme example of this is when the number of groups exceeds the number of retrieved items. Consider, for instance, a retrieval task where 100 items are retrieved, but representation is measured against **all** groups defined by $k$ binary attributes. For $k=7$, for example, we have $2^k=128$ potential groups -- certainly, some groups will not be represented since we only retrieve 100 items. However, even though MPR can't be made exactly equal to 0, there may still be an achievable lower bound where MPR is made small. In future work, understanding the fundamental limits of multi-group proportional representation and, in particular, deriving a lower (converse) bound on MPR is of significant theoretical interest.
>
> **Subjectivity of representative groups.** This is a very insightful point. At the end of the day, MPR requires a human to judge what groups are important to balance over. We acknowledge this limitation from a technical and social perspective and can discuss this in more depth in lines 174-184 as well as in our limitations section and conclusion. However, we note that the definition of groups and identities is a social and human construct, limiting our ability to define fairness outside the context of human subjectivity. To some extent, what is fair and what is unfair is defined by our society and the humans that exist within it. This underlines the importance of developing fairness methods such as MPR that can be adaptable to changing definitions of fairness (through the flexibility of the curation set) as our subjective human and social norms evolve. The only way to reduce subjectivity and converge on some "true" definition of representative groups and fairness is through constant discourse, criticism, and reevaluation of our existing notions. Having a method that can adapt to these changing notions, like MAPR, is critical to the alignment of our technical methods with social ideology.
>
> In our response to the ethics reviewer, we also highlighted that we will include in the paper the importance of participatory approaches in measuring representation. We reiterate the text that will be added to Section 2 here:
>
> Measuring representation requires defining what constitutes 'fair and proportional representation.' While equal representation may suffice for binary groups, proportional representation of more complex intersectional groups requires care. The choice of representation reference statistics (given by the distribution Q in the definition of MPR) should be application-dependent, context-aware, and culturally sensitive.
>
> We recommend that curated datasets used as reference statistics for MPR measurements be developed and verified through participatory design approaches [1,2] in collaboration with diverse stakeholders. This process could involve:
>
> 1. Identifying relevant stakeholder groups, especially those from marginalized communities.
> 2. Conducting focus groups and user studies to understand diverse perspectives on fair representation and evaluate if a curated dataset is indeed representative of a diverse population.
> 3. Iteratively refining the curated dataset based on stakeholder feedback.
> 4. Regularly auditing and updating the dataset to reflect evolving societal norms and demographics.
>
> By involving stakeholders in defining representation goals, we can help ensure that the proportional representation in information retrieval systems measured by MPR aligns with the values and needs of the user base these systems serve.
>
> [1] Delgado, F., Yang, S., Madaio, M., & Yang, Q. (2021). Stakeholder Participation in AI: Beyond" Add Diverse Stakeholders and Stir" NeurIPS 2021 Human Centered AI Workshop
>
> [2] Zytko, D., J. Wisniewski, P., Guha, S., PS Baumer, E., & Lee, M. K. (2022, April). Participatory design of AI systems: opportunities and challenges across diverse users, relationships, and application domains. In CHI Conference on Human Factors in Computing Systems Extended Abstracts

---

> > ### Comment · Reviewer_XWMb · 2024-08-14
> >
> > Thank you for your kind answers. I will positively review the feedback you provided.

---

### Author Rebuttal · Authors · 2024-08-06

# General rebuttal

We sincerely thank the reviewers for their thorough assessment of our paper. We appreciate their recognition of our work's contributions to the critical challenge of fair retrieval, particularly in addressing *intersectional representation* (reviewer 4ELS). We are pleased that the reviewers noted the solid theoretical foundations of MPR (reviewers XWMb and XQou), including our non-trivial generalization bounds and sample complexity results (reviewer 4ELS). The reviewers noted that our *proofs are clear* and our *explanations are clear and professionally presented* (all reviewers). It also is encouraging to see that they recognized how we *effectively take a simple idea and support it with non-trivial error bounds and show evidence towards ideal dataset size* (reviewer 4ELS). We respond to common reviewer comments below, as well as in our response to each reviewer.

**Computational complexity and runtime.** TL;DR: our method (MAPR) is competitive with existing fair retrieval methods in terms of runtime. Relative to the method with the most comparable diversity-similarity trade-offs (MMR, see Fig. 1 of the submitted paper), MAPR has a lower computational overhead per query (up to 100x smaller). A table with runtime results for our method and competing baselines is included in the 1-page additional material. We provide details next.

The sole technical question about our paper was the computational complexity and runtime of our algorithm (MAPR).  We provide runtime results for variations of MAPR, MMR [27], PBM [29], clipCLIP [18], and DebiasCLIP [20].

We benchmark three variants of MAPR:
1. The linear program (LP) solved with the cutting-plane method with 10 cuts (Alg. 1).
2. The same LP but solved with 50 cuts.
3. The quadratic program (QP) that comes from the setting when the family of functions $\mathcal{C}$ is given by the set of bounded-norm linear regressions (and for which we have a closed-form solution for MPR, e.g., Thm 4 and Sec. B.1).

Results for each variant of MAPR are included in Table R1 of the attached PDF. We analyze these results below.

First, we note that ClipCLIP and DebiasCLIP modify the CLIP embedding and then run a vanilla k-NN retrieval. DebiasCLIP has the lowest runtime of all methods we benchmarked since it uses a frozen "debiased" embedding model and does not require any modification of the retrieval algorithm. ClipCLIP has a slight computational overhead where CLIP embedding is modified at retrieval time. Despite these favorable runtimes, __neither of these methods is competitive to MAPR (ours) in terms of diversity-similarity trade-off__ (see results in Figure 1 in the paper). ClipCLIP and DebiasCLIP also do not offer any formal guarantee on the diversity of retrieved items.

PBM also achieves competitive runtime relative to MAPR.  However, unlike our method, PBM is limited to a single binary attribute and aims for equal representation and not for general proportional representation. While efficient for binary attributes (with runtime slightly better than the QP variant of MAPR), PBM also achieves a worse similarity-diversity trade-off relative to MAPR across all of our experiments (Table 1).

The method most competitive to ours in terms of diversity-similarity trade-off is MMR (see Fig. 1),  used in [24, 28] for achieving representation in retrieval. However, MMR is a greedy algorithm and scales poorly with the number of retrieved items. Consequently, MMR's runtime is up to 100x slower than ours (MAPR). In contrast, our method, MAPR, enjoys theoretical guarantees while achieving a significantly more favorable runtime, particularly when implemented using the closed-form quadratic program for linear models (see Prop. 4 and Appendix B.1).

From a theoretical perspective, MAPR is actually very efficient when implemented with modern optimization techniques. MAPR minimizes a linear function subject to two linear constraints and iteratively adds MPR constraints, which amount to two additional linear constraints per iteration. In case the MPR constraint has a closed-form solution, e.g., linear functions (Proposition 4, Section 3) or, more generally, an RKHS (Proposition 10, Appendix B.2), we solve a one-shot quadratic program with a conic structure. Such quadratic problems can be solved very efficiently with off-the-shelf quadratic or conic solvers such as OSQP or ECOS (available in Python). Here, the worst-case complexity for interior-point methods solving a quadratic program involved in retrieving $k$ items is typically $O(k^{3.5})$. However, solvers like OSQP work really well with the constraints that we have (such as box constraints), achieving $O(k^2)$ complexity per iteration with a relatively small number of iterations in practice. We observe that OSQP takes only a few iterations (around 30) to find the optimal solution described in Table 1.

---

### Decision · Program_Chairs · 2024-09-25

**Decision:**

Accept (poster)

**Comment:**

The paper proposes Multi-Group Proportional Representation (MPR) as a novel metric to ensure diverse and proportional representation across intersectional groups in image retrieval tasks. The authors demonstrate the limitations of current methods in addressing intersectional fairness and introduce an optimization algorithm, MAPR, to enforce MPR while minimizing retrieval error. They provide theoretical guarantees, extensive empirical validation, and address challenges related to biased datasets.

The paper addresses the problem of ensuring fairness across multiple, intersecting groups, which is often overlooked. Despite its computational challenges, the method is novel, theoretically sound, and backed by strong empirical results. The potential societal impact of the proposed solution makes it a valuable contribution to the field, particularly in applications involving image retrieval and other tasks requiring equitable representation. As such, the AC recommends acceptance to NeurIPS.